# Nuclear microRNA 9 mediates G-quadruplex formation and 3D genome organization during TGF-β-induced transcription

Julio Cordero[1,2,3] ✉, Guruprasadh Swaminathan [4], Diana G. Rogel-Ayala [3,4], Karla Rubio[3,4,5,6], Adel Elsherbiny[1,2], Samina Mahmood[7], Witold Szymanski [8], Johannes Graumann [8], Thomas Braun [9], Stefan Günther [7,9], Gergana Dobreva [1,2,10] & Guillermo Barreto [3,4] ✉

The dynamics of three-dimensional (3D) genome organization are essential to transcriptional regulation. While enhancers regulate spatiotemporal gene expression, chromatin looping is a means for enhancer-promoter interactions yielding cell-type-specific gene expression. Further, non-canonical DNA secondary structures, such as G-quadruplexes (G4s), are related to increased gene expression. However, the role of G4s in promoter-distal regulatory elements, such as super-enhancers (SE), and in chromatin looping has remained elusive. Here we show that mature microRNA 9 (*miR-9*) is enriched at promoters and SE of genes that are inducible by transforming growth factor beta 1 (TGFB1) signaling. Moreover, we find that *miR-9* is required for formation of G4s, promoter-super-enhancer looping and broad domains of the euchromatin histone mark H3K4me3 at TGFB1-responsive genes. Our study places *miR-9* in the same functional context with G4s and promoter-enhancer interactions during 3D genome organization and transcriptional activation induced by TGFB1 signaling, a critical signaling pathway in cancer and fibrosis.

The nuclear genome in eukaryotic cells consists of DNA molecules packaged into thread-like structures known as chromosomes, which are built of chromatin. Thus, chromatin is the physiological template for biological processes in the nucleus of eukaryotic cells. Studying how chromatin is folded inside the cell nucleus and its dynamic three-dimensional (3D) structure is essential to understanding these biological processes comprising transcription, RNA-splicing, -processing, -editing, DNA-replication, -recombination, and -repair. The chromatin is hierarchically organized at different levels including chromosomal territories, compartments, and self-interacting topologically associating domains, altogether giving rise to a highly dynamic 3D genome organization[1]. Remarkably, the structure of the genome is intrinsically linked to its function as shown by extensive correlations between chromatin condensation and related gene transcription. For example,

[1]Department of Cardiovascular Genomics and Epigenomics, European Center for Angioscience (ECAS), Medical Faculty Mannheim, Heidelberg University, 68167 Mannheim, Germany. [2]German Centre for Cardiovascular Research (DZHK), 68167 Mannheim, Germany. [3]Lung Cancer Epigenetics, Max-Planck-Institute for Heart and Lung Research, 61231 Bad Nauheim, Germany. [4]Université de Lorraine, CNRS, Laboratoire IMoPA, UMR 7365, F-54000 Nancy, France. [5]Massachusetts General Hospital and Harvard Medical School, Charlestown, MA 02129, USA. [6]International Laboratory EPIGEN, Consejo de Ciencia y Tecnología del Estado de Puebla (CONCYTEP), Instituto de Ciencias, EcoCampus, Benemérita Universidad Autónoma de Puebla, 72570 Puebla, Mexico. [7]ECCPS Bioinformatics and Deep Sequencing Platform, Max-Planck-Institute for Heart and Lung Research, 61231 Bad Nauheim, Germany. [8]Department of Medicine, Institute of Translational Proteomics & Core Facility Translational Proteomics, Philipps-University Marburg, 35043 Marburg, Germany. [9]Department of Cardiac Development, Max-Planck-Institute for Heart and Lung Research, 61231 Bad Nauheim, Germany. [10]Helmholtz-Institute for Translational Angio-CardioScience (HI-TAC) of the Max Delbrück Center for Molecular Medicine in the Helmholtz Association (MDC) at Heidelberg University, 69117 Heidelberg, Germany. ✉e-mail: Julio.Cordero@medma.uni-heidelberg.de; Guillermo.Barreto@univ-lorraine.fr

chromatin shows condensed regions, referred to as heterochromatin (by convention, transcriptionally "inactive"), and less condensed regions, referred to as euchromatin (transcriptionally "active"). Transcriptional regulation directly corresponds to the mechanisms of how chromatin may be structurally arranged rendering it accessible to the transcription machinery[2]. These mechanisms regulating chromatin structure and transcription involve histone modifications, histone deposition, nucleosome remodeling, DNA methylation, non-coding RNAs (ncRNA), and secondary structures of nucleic acids, among others[3–7]. In addition, an increasing number of recent publications based on integrative analysis of multi-omics studies implementing next-generation sequencing (NGS) technologies, chromosome conformation capture-based methods, and super-resolution microscopy have provided comprehensive and multilevel insights into 3D genome organization emphasizing its role during transcriptional regulation[8].

Chromatin structure alone does not determine the functional status of a gene, but it effectively enables RNA polymerase II (Poll II) recruitment to the promoters, as well as binding of transcription factors, co-activators, co-repressors to DNA sequences that function as regulatory elements controlling gene expression[9]. A promoter is a sequence of DNA to which proteins bind to initiate transcription of RNA molecules that are usually complementary to the DNA sequence that is located 3′ of the promoter. On the other hand, enhancers are relatively short (-100–1000 bp) DNA sequences that are bound by transcription factors and regulate gene transcription independent of their distance, location, or orientation relative to their cognate promoter[10]. Super-enhancers (SE) have been proposed to be long genomic domains consisting of clusters of transcriptional enhancers enriched with histone modification markers (such as histone 3 monomethylated at lysine 4 or acetylated at lysine 27, H3K4me1 and H3K27ac respectively), cofactors (such as mediator of RNA polymerase II transcription subunit 1, MED1, and components of the multimeric protein complex Cohesin), chromatin modifying proteins (such as E1A Binding Protein P300, EP300) and cell-type-specific transcription factors[11–13]. One versatile feature of chromatin is its ability to form loops, mediating long-range interactions in which two distant sequences of DNA come into close physical proximity. Chromatin looping has been broadly accepted as a means for enhancer-promoter interactions[14].

In addition to the predominant DNA double-helix structure, there are different non-canonical DNA secondary structures, including G-quadruplex (G4), R-loop, H-DNA, Z-DNA, etc. [4]. A G4 represents a stable nucleic acid secondary structure formed by square planes, in which four guanines located in the same plane are stabilized by a monovalent cation[15]. While the early work on G4s mainly focused on their roles in telomeres[16], recent studies demonstrated that G4s are enriched at promoters[17,18] and related to increased gene expression[19–22]. On the other hand, G4s were also located in gene bodies and related to reduced gene expression by inhibiting elongation of RNA polymerase[23,24]. All these previous studies characterized the biological function of G4s in promoter or promoter-proximal regions enhancing or reducing gene expression depending on the relative position of G4s. Nevertheless, the role of G4s in promoter-distal regulatory elements, such as SE, as well as in chromatin looping mediating long-range enhancer-promoter interactions remains unclear.

The majority of the eukaryotic genome is transcribed into ncRNAs including microRNAs (miRNAs, 21−25 nucleotides long) and long non-coding RNAs (lncRNAs, >200 nucleotides long)[25]. LncRNAs are important regulators of different biological processes in the nucleus[26]. Together with other factors, lncRNAs provide a framework for the assembly of defined chromatin structures at specific loci, thereby modulating gene expression, centromere function, and silencing of repetitive DNA elements[26,27]. Although miRNAs are assumed to act primarily in the cytosol by inhibiting translation[28], mature miRNAs have also been reported in the nuclei of different cells[6,29–32]. While a hexanucleotide element has even been reported to direct miRNA nuclear import[33], the function of miRNAs in the cell nucleus has been sparsely studied. Here we report on microRNA-9 (miR-9), which even though its nucleotide sequence is highly conserved across species, also shows high diversity in expression patterns and biological functions depending on the cellular context[34,35]. For example, miR-9 has been reported to target the lncRNA MALAT1 for degradation in the cell nucleus[32]. However, it has not been linked to transcription regulation, chromatin structure nor 3D genome organization. Here we propose a mechanism of transcriptional regulation of transforming growth factor beta 1 (TGFB1) responsive genes that requires nuclear miR-9 and involves G4s and promoter-SE looping.

## Results

### Mature *miR-9* is detected in the cell nucleus and is enriched at promoters and introns

A phylogenetic tree generated from sequences of mature mouse miRNAs and a heat map comparing their sequence similarity showed miR-9 to cluster with miRNAs that have been functionally characterized in the cell nucleus, such as miR-29b-3p[33], miR-126-5p[36] and let-7d-5p[6] (Fig. 1a, top). Sequence alignment between mouse miR-126-5p, miR-9 and miR-29b-3p showed various nucleotides as conserved in a sequence stretch reported as nuclear shuttling motif from miR-29b-3p[33] (Fig. 1a, bottom). Accordingly, we refer to the partially conserved sequence 5′-AKYACCWUUUGRUWA-3′ as an expanded miRNA nuclear shuttling motif. In addition, we found that the human orthologs of these mature miRNAs, hsa-miR-126-5p, hsa-miR-29B-3p and hsa-miR-9-5p, also contain the expanded miRNA nuclear shuttling motif (Supplementary Fig. 1a), demonstrating its conservation across species. To confirm the nuclear localization of mature miR-9, we performed expression analysis after cell fractionation using TaqMan assays specific for mature miR-9 and total RNA isolated from the cytosolic and the nuclear fractions of different cells (Fig. 1b and Supplementary Fig. 1b), including mouse lung fibroblasts (MLg and MFML4), mouse lung epithelial cells (MLE-12), mouse mammary gland epithelial cells (NMuMG), and primary human lung fibroblasts (hLF). We were interested in lung cells since miR-9 levels are increased in hyperproliferative lung diseases. We detected mature miR-9 in the cytosolic fraction and the nuclear fraction of all cells analyzed. Interestingly, the relative levels of nuclear miR-9 were higher in mouse fibroblasts as compared to epithelial cells. Further, the nuclear localization of miR-9 was confirmed by RNA fluorescence in situ hybridization (FISH) in MLg cells (Fig. 1c and Supplementary Fig. 1c) and hLF from control donors (Ctrl hLF) or patients with idiopathic pulmonary fibrosis (IPF hLF; Supplementary Fig. 1d), a lethal interstitial lung disease involving TGFB1 signaling[37]. In MLg cells, we detected miR-9 in specific regions of the nuclei, whereas the levels of miR-9 were reduced after loss-of-function (LOF) experiments using unlabeled miR-9-specific antagomiR probes. In Ctrl hLF, the intensity of miR-9 FISH was higher in the cytosol than in the nucleus, whereas in IPF hLF the majority of miR-9 was detected in the cell nucleus, pointing to a translocation mechanism of miR-9 into the cell nucleus potentially related to IPF. Further, miR-9-LOF in IPF hLF reduced the levels of miR-9. All these results suggest a function of miR-9 in the cell nucleus. To investigate the role of miR-9 in the cell nucleus we performed a sequencing experiment after chromatin isolation by miRNA purification (ChIRP-seq) using chromatin from MLg cells and control (Ctrl) or miR-9-specific biotinylated antisense oligonucleotides for the precipitation of endogenous mature miR-9 along with the chromatin bound to it (Fig. 1d, e and Supplementary Fig. 2a−f). To demonstrate the specificity of our ChIRP-seq experiment, we also used a probe specific for another miRNA characterized in the cell nucleus (miRNA lethal 7 d, Mirlet7d, also known as let-7d)[8,38], and chromatin from MLE-12 cells. We detected specific enrichment of miR-9 at loci without

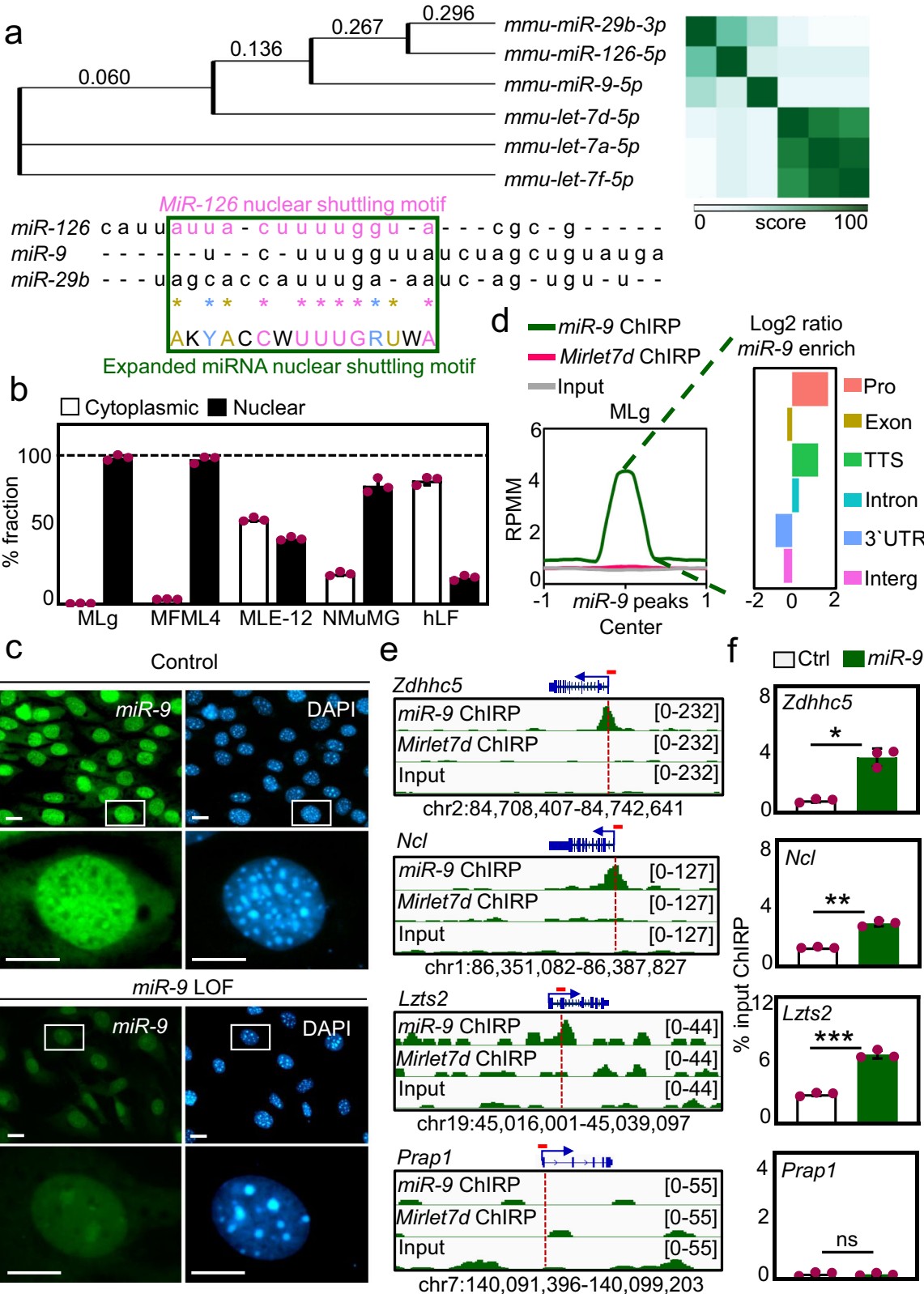

*Mirlet7d* in MLg and MLE-12 cells (Fig. 1d, left, and Supplementary Fig. 2b, left). Further, genome-wide binding profile analysis of *miR-9* in MLg cells revealed an increase in the number of *miR-9* peaks at promoters, transcription termination sides (TTS) and intronic regions compared to the negative control (Fig. 1d, right), whereas in MLE-12 cells *miR-9* was enriched at TTS and intronic regions (Supplementary Fig. 2b, right). Interestingly, the loci with *miR-9* enrichment were

different in MLg and MLE-12 cells (Supplementary Fig. 2c), suggesting that *miR-9* regulates different genes in different cells. The loci of the putative *miR-9* target genes in MLg cells were distributed on all chromosomes (Supplementary Fig. 2e, f). From the *miR-9* ChIRP-seq results we selected putative *miR-9* target genes (*Zdhhc5, Ncl, Lzts2* and *Hdac7*) for further analysis. Visualization of the loci of the putative *miR-9* target genes using the integrative genomic viewer (IGV) (Fig. 1e and

**Fig. 1 | Mature *miR-9* is detected in the cell nucleus enriched at promoters and introns. a** Phylogenetic tree (left) and heat map (right) generated using sequences of indicated mature mouse miRNAs. Numbers, distance score. Bottom, sequence alignment of indicated mature mouse miRNAs highlighting the published *miR-126* nuclear shuttling motif (magenta) and the expanded miRNA nuclear shuttling motif (green square) using IUPAC nucleotide code. Pink letters, conserved among all sequences; golden, conserved in at least 2 sequences; blue, conserved type of base (either purine or pyrimidine). **b** Mature *miR-9*-specific TaqMan assay following cellular fractionation of indicated cell lines. **c** Fluorescence microscopy of MLg cells after RNA FISH confirmed nuclear localization of endogenous *miR-9*. Cells were transiently transfected with control (top) or *miR-9*-specific antagomiR probes (bottom) to induce a *miR-9* loss-of-function (LOF). Representative images from three independent experiments. Squares are shown at higher magnification. DAPI, nucleus. Scale bars, 10 μm. **d** Left, enrichment plot after *miR-9*-, or *Mirlet7d*-specific ChIRP-seq in MLg cells. *Mirlet7d*-specific probe was used as negative control.

RPMM, read count per million mapped reads. Right, genome-wide distribution of *miR-9* peaks by ChIRP-seq in MLg cells in different genomic regions and represented as Log2 ratios. Pro, promoters; TTS, transcription termination sites; Intron, intronic regions; 3´UTR, 3´untraslated regions; Interg, intergenic regions. **e** Visualization of selected *miR-9* target genes using IGV genome browser showing *miR-9* or *Mirlet7d* enrichment in MLg cells. ChIRP-seq reads were normalized using RPKM (reads per kilobase of transcript per million mapped reads) and are represented as log2 enrichment over inputs. Images show the indicated gene loci with genomic coordinates. Arrows, direction of genes; blue boxes, exons; red line, regions selected for single gene analysis in (**f**). **f** Analysis of selected putative *miR-9* target genes by ChIRP using chromatin from MLg cells and control (Ctrl) or *miR-9*-specific biotinylated antisense oligonucleotides. In all bar plots data are presented as means; error bars, s.e.m ($n = 3$ biologically independent experiments); asterisks, $P$-values after two-tailed t-test, ***$P \leq 0.001$; **$P \leq 0.01$; *$P \leq 0.05$; ns, non-significant. See also Supplementary Figs. 1 and 2. Source data are provided as a Source Data file.

Supplementary Fig. 2d) showed specific *miR-9* enrichment at the promoters, whereas no *miR-9* enrichment was detected at the promoter of the negative control *Prap1*. These results were confirmed by quantitative PCR (qPCR) after ChIRP using promoter-specific primers, chromatin from MLg cells, and Ctrl or *miR-9*-specific biotinylated antisense oligonucleotides (Fig. 1f). Taken together, our results demonstrate that mature *miR-9* is present in the cell nucleus and directly binds to promoters of putative *miR-9*-target genes, suggesting a potential role in transcription regulation.

## MiR-9 is required for H3K4me3 broad domains at promoters, basal transcriptional activity, and G-quadruplex formation

To further investigate a potential role of nuclear *miR-9* in transcription regulation, we performed a sequencing experiment following Cleavage Under Targets and Tagmentation (CUT&Tag) for high-resolution, genome-wide profiling of tri-methylated lysine 4 of histone 3 (H3K4me3) in MLg and MLE-12 cells that were transiently transfected with Ctrl or *miR-9*-specific antagomiR to induce a *miR-9*-LOF (Fig. 2a–e, Supplementary Fig. 3a–d). Peak distribution analysis of the H3K4me3 CUT&Tag showed that 60.8% ($P = 0.01$) of the H3K4me3 broad domains in Ctrl transfected MLg cells were enriched with *miR-9* (Fig. 2a), whereas 63% ($P < 0.01$) of the H3K4me3 broad domains were enriched with *miR-9* in Ctrl transfected MLE-12 cells (Supplementary Fig. 3b). Interestingly, H3K4me3 levels at broad domains were significantly reduced in MLg cells from a median of 1.6 RPKM (IQR = 3.3) in Ctrl transfected cells to a median of 0.8 RPKM (IQR = 1.8; $P = 0.002$) following *miR-9*-LOF (Fig. 2b, top), whereas the effects of *miR-9*-LOF in MLE-12 cells were not significant (Fig. 2b, bottom). In addition, we observed that the loci of the H3K4me3 broad domains with *miR-9* enrichment were different in MLg and MLE-12 cells (Supplementary Fig. 3c, d), confirming that *miR-9* regulates different genes in these two cell lines. Due to these results and the higher levels of nuclear *miR-9* (Fig. 1b), we focused on MLg cells. Further peak distribution analysis showed that H3K4me3 broad domains were reduced from 27.6% in Ctrl transfected MLg cells to 22.1% after *miR-9*-LOF, whereas medium and narrow H3K4me3 domains increased (Fig. 2c). Interestingly, the shift from H3K4me3 broad domains to medium and narrow domains following *miR-9*-LOF was significant at promoters but not at gene body and intergenic regions (Fig. 2d). However, enrichment plots showed that H3K4me3 levels were reduced following *miR-9*-LOF in H3K4me3 broad and medium domains at promoter, gene body and intergenic regions (Fig. 2e). The reduction of H3K4me3 levels after *miR-9*-LOF was confirmed by confocal microscopy after H3K4me3-specific immunostaining in Ctrl- and *miR-9*-antagomiR transfected MLg cells (Supplementary Fig. 3e).

Since broad domains of H3K4me3 have been associated with increased transcription elongation[39], we analyzed the transcriptome of MLg cells after *miR-9*-LOF by total RNA sequencing (RNA-seq, Fig. 2f–h, Supplementary Fig. 3f). Remarkably, from the transcripts that were significantly affected after *miR-9*-LOF ($n = 3320$), only a minority

($n = 881$; 26.5%) showed increased expression after *miR-9*-LOF, whereas 73.5% ($n = 2439$) showed reduced expression with a median of 1.03 log2 RPKM and an interquartile range (IQR) of 1.51 log2 RPKM ($P = 5.34\text{E-36}$), when compared to 1.50 log2 RPKM (IQR = 1.81 log2 RPKM) in Ctrl antagomiR transfected cells (Fig. 2g, top). The genes coding for the transcripts significantly affected by *miR-9*-LOF will be further referred to as *miR-9* target genes. We also observed genes coding for transcripts that were not significantly affected by *miR-9*-LOF (Fig. 2g, bottom, non-targets). Remarkably, the most significant transcription reducing effect after *miR-9*-LOF was observed in those transcripts whose promoter or gene body were embedded within H3K4me3 broad domains (Fig. 2h).

Our results indicate that *miR-9* is required for the basal transcriptional activity of its target genes. This interpretation was supported by sequencing subsequent to chromatin immunoprecipitation (ChIP-seq) in mouse embryonic fibroblasts (MEF) using antibodies specific to total RNA polymerase II (Pol II) and serine 5 phosphorylated Pol II (Pol II S5p[5],) showing transcription initiation. Heat maps representing the results of the *miR-9* ChIRP-seq (Fig. 3a), Pol II and Pol II S5p ChIP-seq (Fig. 3b) revealed that Pol II and Poll II S5p were enriched at the transcription start sites (TSS) of the *miR-9* target genes. Moreover, genome-wide precision nuclear run-on assay (PRO-seq)[40] and global run-on sequencing (GRO-seq)[41], both in MEF, showed nascent RNAs at the TSS of the *miR-9* target genes demonstrating their basal transcriptional activity (Fig. 3c). Correlating with these results, we also observed at the TSS of the *miR-9* target genes increased chromatin accessibility and increased H3K4me3 levels by assay for transposase-accessible chromatin with sequencing (ATAC-seq) and ChIP-seq, respectively (Fig. 3d). To gain further insight into these results, we performed a motif search analysis of the *miR-9* target genes and identified significant enrichment of nucleotide motifs with high G content (Fig. 3e), known to favor the formation of G4[4,15]. Remarkably, we found similar motifs significantly enriched in loci that form G4 as determined by G4 CUT&Tag. Further, as G4 has been shown to cooperate with transcription factors at gene promoters[20,21], we analyzed publicly available NGS data generated using different methods for the assessment of G4 formation (Fig. 3f). On one hand, we analyzed ChIP-seq data generated using an artificial 6.7 kDa G4 probe (G4P) protein, which binds G4s with high affinity and specificity[18]. On the other hand, we analyzed NGS data generated by G4access, which is an antibody-independent method relying on moderate nuclease digestion of chromatinized DNA[21]. Remarkably, we found by both approaches enrichment of G4 at the TSS of *miR-9* target genes (Fig. 3f). Our results demonstrate that the TSS of *miR-9* target genes show (1) reduced nucleosome density, (2) increased levels of the euchromatin histone mark H3K4me3, (3) enrichment of G4, *miR-9* and transcription initiating Poll II S5p, and (4) nascent RNAs.

To demonstrate that *miR-9* is required for G4 formation, we analyzed by CUT&Tag using G4-specific antibodies chromatin from MLg

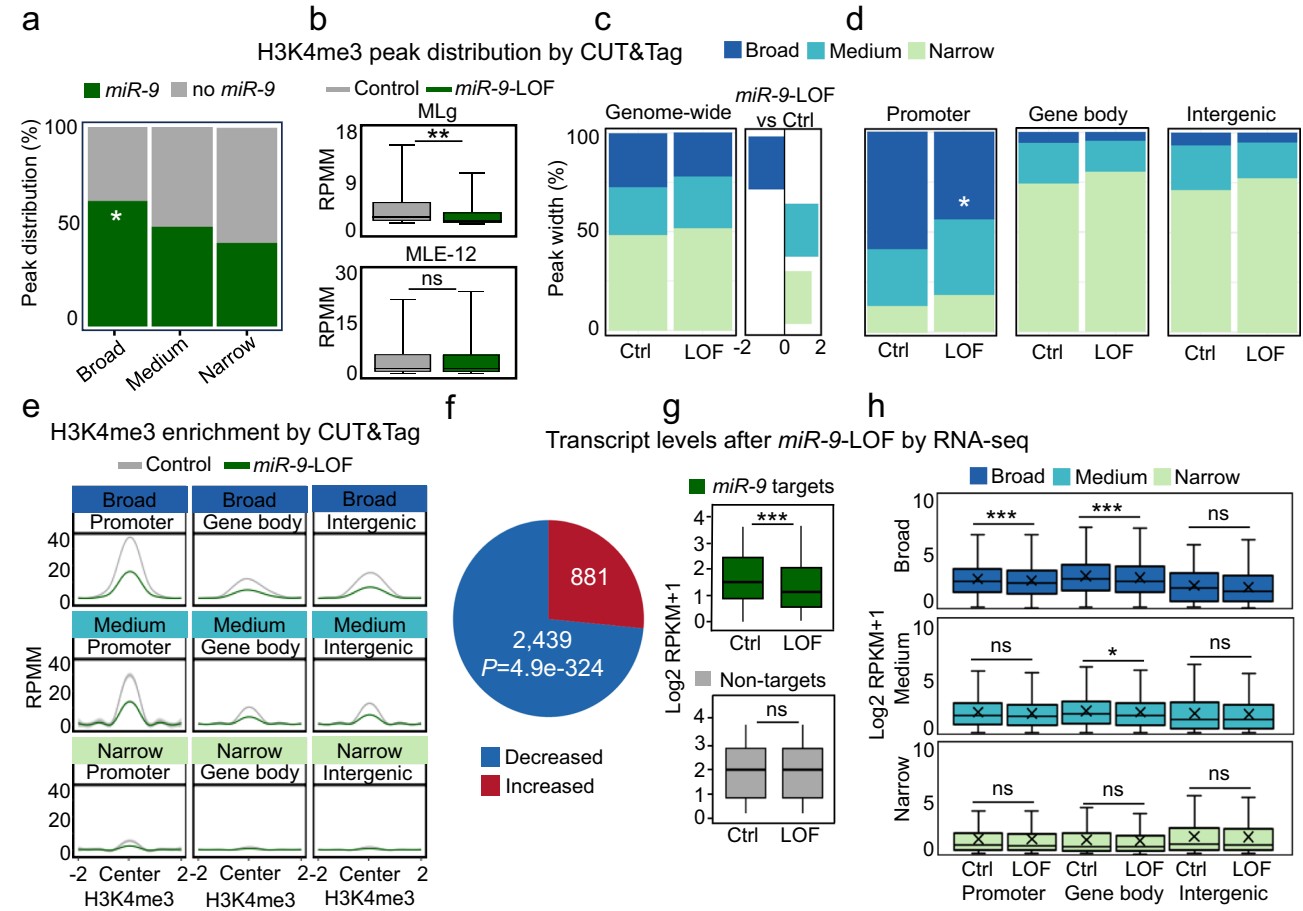

**Fig. 2 | *MiR-9* is required for H3K4me3 broad domains, and basal transcriptional activity. a** Genome-wide distribution of H3K4me3 peaks by CUT&Tag in MLg cells, relative to broad (≥2.7 kb), medium (≥2 kb and <2.7 kb) and narrow (<2 kb) H3K4me3 domains that are also enriched with *miR-9* or not (no *miR-9*). **b** Box plots showing the levels of H3K4me3 in MLg and MLE-12 cells that were transiently transfected with control (Ctrl) or *miR-9*-specific antagomiR probes to induce *miR-9* loss-of-function (LOF). RPMM, read count per million mapped reads. Bar plots displaying the broadness of H3K4me3 domains genome-wide (**c**) or in different genomic regions (**d**) in MLg cells that were transfected as in (**b**). Square in (**c**) shows H3K4me3 enrichment in different domains as Log2 ratios of MLg cells after *miR-9*-LOF *versus* Ctrl transfected cells. Promoter (Peaks -/+ 2 kb from TSS), Gene body (exon and intron regions outside the −/+2 kb TSS) and Intergenic (peaks not located in previous regions). **e** Aggregate plots showing H3K4me3 enrichment at the indicated genomic regions and relative to indicated H3K4me3 domains in

MLg cells transfected as in (**b**). Data were normalized using RPMM. **f** Total RNA-seq in MLg cells transfected as in (**b**). Pie chart shows distribution of significantly, differentially expressed transcripts (*n* = 3320) in decreased (*n* = 2439) and increased transcripts (*n* = 881) after *miR-9*-LOF. **g, h** Box plots of RNA-seq-based expression analysis of transcripts with non-significantly changed levels (non-targets; *n* = 324) and significantly decreased levels after *miR-9*-LOF (*miR-9* targets; *n* = 2439). In (**h**), data of significantly decreased transcripts (*n* = 2439) were separated into the indicated H3K4me3 domains, and into the indicated genomic regions. In all box plots, values were normalized using RPKM; represented as log2 RPKM + 1; and showed as median (middle line); 25th, 75th percentile (box) and 5th and 95th percentile (whiskers). In all plots asterisks represent *P*-values, ***$P \leq 0.001$; *$P \leq 0.05$; ns, non-significant. *P*-values were calculated after two-tailed t-test (box plots) or two-tailed Fisher exact test (bar plots). See also Supplementary Fig. 3. Source data are provided as a Source Data file.

and MLE-12 cells transiently transfected with Ctrl or *miR-9*-specific antagomiR (Fig. 3g and Supplementary Fig. 4a−c). Analysis of the G4 CUT&Tag data without or with filtering based on G4Hunter scores[42] showed genome-wide reduction of G4s after *miR-9*-LOF in MLg and MLE-12 cells (Fig. 3g and Supplementary Fig. 4b, both left), thereby demonstrating the requirement of *miR-9* for G4 formation in both cell lines. However, correlating with the levels of nuclear *miR-9* in both cell lines (Fig. 1b), the reducing effect of *miR-9*-LOF on G4 levels was more pronounced in MLg cells at TSS of *miR-9* target genes (Fig. 3g and Supplementary Fig. 4b, both right). Interestingly, the majority of the loci with G4s were different in MLg and MLE-12 cells (Supplementary Fig. 4c), supporting that G4s are involved in the regulation of different genes in both cell lines. To verify the interaction between mature *miR-9* and G4s, we performed TaqMan-based miRNA enrichment analysis following chromatin-RNA immunoprecipitation (Ch-RIP) using G4 specific antibodies (Fig. 3h). G4s significantly bound mature *miR-9* and *miR-9*-LOF abolished this interaction, whereas G4s did not bind mature

*Mirlet7f*, a miRNA used as negative control, thereby showing the specificity of the interaction between mature *miR-9* and G4s. To further investigate this interaction and identify protein-binding partners of *miR-9* in the nucleus, we performed a high-resolution mass spectrometry based proteomic approach after miRNA pulldown (miR-Pd) using the nuclear fraction of MLg or MLE-12 cells and biotinylated control miRNA (*mirctrl*) or *miR-9* as baits (Fig. 3i, Supplemenatary Fig. 4d−f and Source Data file). Results from three independent experiments identified 169 proteins in the nuclear fraction of MLg cells and 233 proteins in the nuclear fraction of MLE-12 cells that were significantly enriched after *miR-9*-Pd. From the *miR-9* binding proteins, 20 proteins in the nuclear fraction of MLg cells and 58 proteins in the nuclear fraction of MLE-12 cells have been reported to interact with G4s[43]. Interestingly, *miR-9* did not significantly bind AGO1, AGO2, and MEX3D. However, Gene Set Enrichment Analysis (GSEA)[44] of the *miR-9* binding proteins (Fig. 3j) showed that nuclear *miR-9* interacted with proteins involved in G4s (*P* = 1.23E-4), Chromatin (*P* = 1.5E-3), RNA

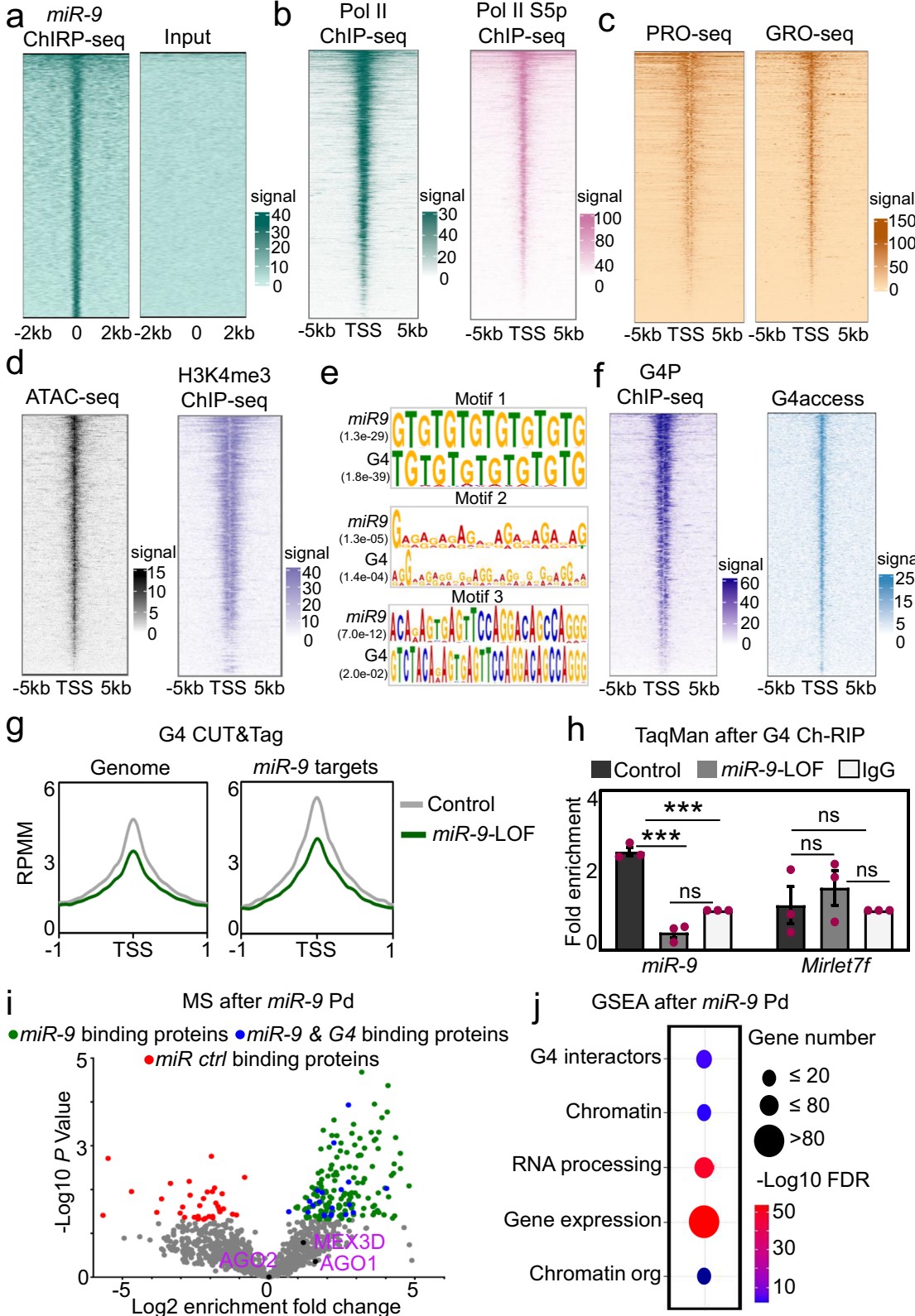

processing ($P = 1.65\text{E-}49$), Gene expression ($P = 1.56\text{E-}54$) and Chromatin organization ($P = 9.3\text{E-}3$). All these results support an important role of *miR-9* in G4 formation and transcription regulation. Furthermore, loci visualization of selected *miR-9* target genes (*Zdhhc5*, *Ncl*, *Lzts2*, *Hdac7* and *Ep300*) using the IGV genome browser (Fig. 4a top and Supplementary Fig. 5a, b) showed enrichment of the euchromatin histone mark H3K4me3 at the promoters, which was reduced after

*miR-9*-LOF. We also observed nascent RNA at the same loci, supporting basal transcriptional activity, as well as G4 enrichment. Notably, G4 enrichment was reduced by *miR-9*-LOF, confirming the requirement of *miR-9* for G4 formation. Zooming into the loci revealed G-rich sequences that favor the formation of G4[4,15] (Fig. 4a, bottom, and Supplementary Fig. 5a, bottom). These results were confirmed by promoter analysis of *Zdhhc5*, *Ncl* and *Lzts2* by ChIP using H3K4me3- or

**Fig. 3 | *MiR-9* is required for G-quadruplex formation at promoters. a** Heat map for *miR-9* enrichment at the TSS ± 2 kb of *miR-9* target genes as determined by RNA-seq in Fig. 2f, g. Heat maps for enrichment of total Pol II and Pol II S5p (**b**), nascent RNA by precision nuclear run-on assay (PRO-seq) and global run-on sequencing (GRO-seq) (**c**), chromatin accessibility by ATAC-seq and H3K4me3 by ChIP-seq (**d**), at the TSS ± 5 kb of *miR-9* target genes. **e** Motif analysis of *miR-9* target genes showed significant enrichment of nucleotide motifs that are similar to motifs found in loci form G4 as determined by G4 CUT&Tag. **f** Heat maps for G4 enrichment at the TSS ± 5 kb of the *miR-9* target genes by G4P ChIP-seq (left) or G4access (right). **g** Enrichment plots after G4-specific CUT&Tag in MLg cells transiently transfected with control (Ctrl) or *miR-9*-specific antagomiR probes to induce *miR-9* loss-of-function (LOF). Data were normalized using RPMM. **h** *Mir-9-* or *Mirlet7f*-specific TaqMan assays after chromatin-RNA immunoprecipitation (Ch-RIP) in MLg cells transfected as in (**g**) and using G4-specific antibodies or IgG. Bar plot shows fold

enrichment over IgG as means; error bars, s.e.m ($n = 3$ biologically independent experiments); asterisks, *P*-values after two-tailed t-test, ***$P ≤ 0.001$; ns, non-significant. **i** Mass spectrometry-based analysis (MS) of proteins precipitated by miRNA pulldown (miR-Pd) from the nuclear fraction of MLg cells using biotinylated control miRNA (*mirctrl*) or *miR-9* as baits. Volcano plot representing the significance (-log10 *P*-values after limma two-tailed t-test) vs. enrichment fold change (log2 enrichment ratios) between *miR-9*-Pd and *mirctrl*-Pd. Each dot represents a protein; green, proteins significantly enriched by *miR-9*; blue, proteins significantly enriched by *miR-9* and interacting with G4s[43]; red, proteins significantly enriched by *mirctrl*; gray and black, non-significantly bound proteins. Black dots show AGO1, AGO2 and MEX3D. **j** Gene set enrichment analysis (GSEA) of the proteins that were significantly binding *miR-9* as identified in (**i**). G4 interactors, proteins interacting with G4s[43]; Chromatin org, Chromatin organization; FDR, false discovery rate. See also Supplementary Fig. 4. Source data are provided as a Source Data file.

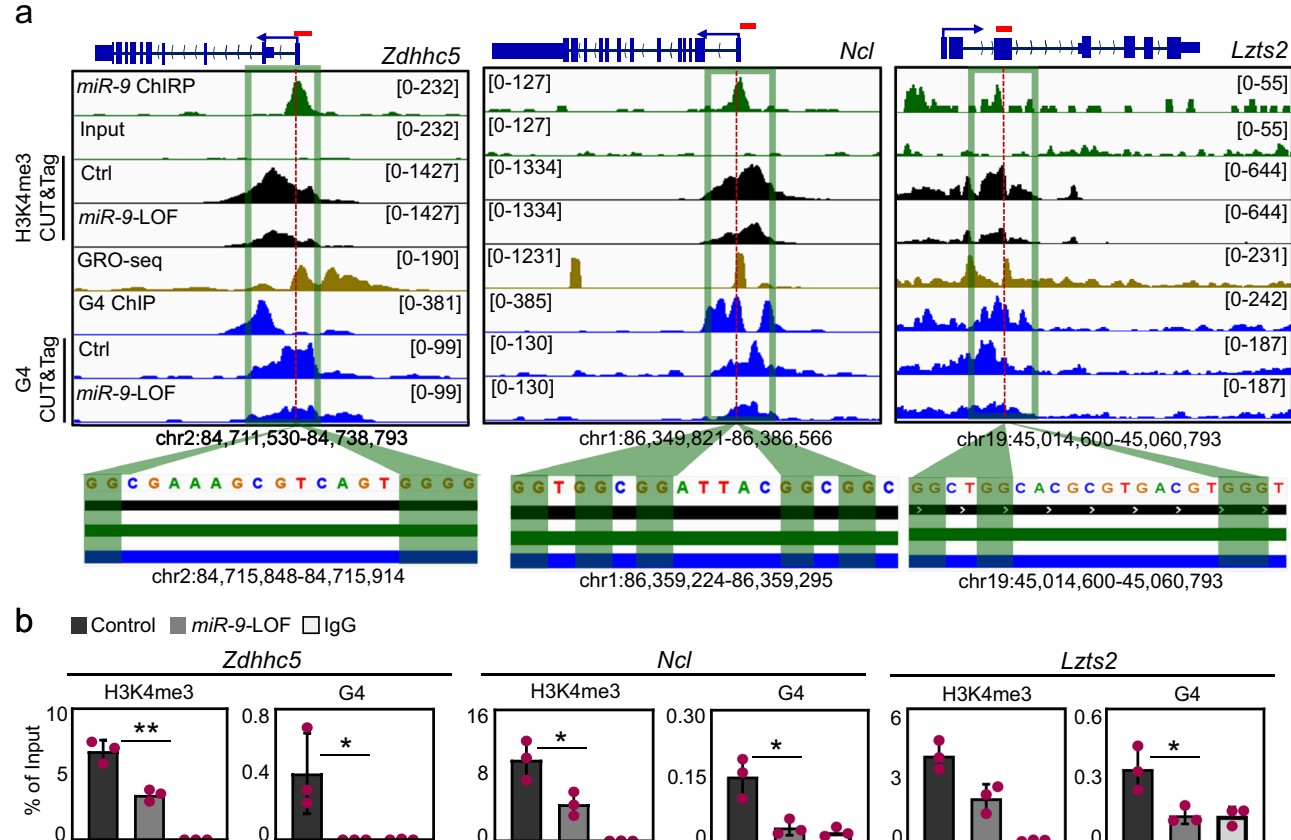

**Fig. 4 | H3K4me3, nascent RNA and G4 are enriched at promoters of selected *miR-9* target genes. a** Visualization of selected *miR-9* target genes using IGV genome browser showing enrichment of *miR-9* by ChIRP-seq (green), H3K4me3 by CUT&Tag in Ctrl and *miR-9*-specifc antagomiR transfected MLg cells (black), nascent RNA by GRO-seq (brown), G4 by G4P ChIP-seq in NIH/3T3 cells (blue), G4 by CUT&Tag in Ctrl and *miR-9*-specifc antagomiR transfected MLg cells (blue). Reads were normalized using reads per kilobase per million (RPKM) after bamCoverage. Images show the indicated gene *loci* with their genomic coordinates. Arrows, direction of the genes; blue boxes, exons; red lines, regions selected for single gene

analysis in (**b**); green squares, regions with enrichment of *miR-9*, H3K4me3, nascent RNA and G4; dotted lines, regions shown at the bottom with high G content. Bottom, black line, H3K4me3 enrichment; green line, *miR-9* enrichment; blue line, G4 enrichment. **b** Analysis of the promoter of selected *miR-9* target genes by ChIP using chromatin from MLg cells transfected with control (Ctrl) or *miR-9*-specific antagomiR to induce *miR-9* loss-of-function (LOF). Bar plots presenting data as means; error bars, s.e.m ($n = 3$ biologically independent experiments); asterisks, *P*-values after two-tailed t-test, **$P ≤ 0.01$; *$P ≤ 0.05$. See also Supplementary Fig. 5. Source data are provided as a Source Data file.

G4- specific antibodies and chromatin from MLg cells that were transiently transfected with Ctrl or *miR-9*-specific antagomiR probes (Fig. 4b). We detected H3K4me3 and G4 enrichment at the promoters of all analyzed *miR-9* target genes in Ctrl antagomiR-transfected cells, which was significantly reduced after *miR-9*-LOF. Our results demonstrate that the promoters of *miR-9* target genes are enriched with H3K4me3 and G4, correlating with the basal transcriptional activity detected by RNA-seq (Fig. 2f–h), in a *miR-9*-dependent manner.

## Nuclear *miR-9* is enriched at super-enhancers and is required for G-quadruplexes

An alluvial plot using the data from *miR-9* ChIRP-seq and G4P ChIP-seq[18] showed enrichment of *miR-9* and G4 at loci that are also enriched for markers of SE, such as MED1 and H3K27ac[45,46] (Fig. 5a). Furthermore, a Venn diagram using the same data sets together with the data from a GRO-seq experiment[41] showed 3583 common loci (Fig. 5b) suggesting transcriptional activity from these loci. Remarkably, 95.5%

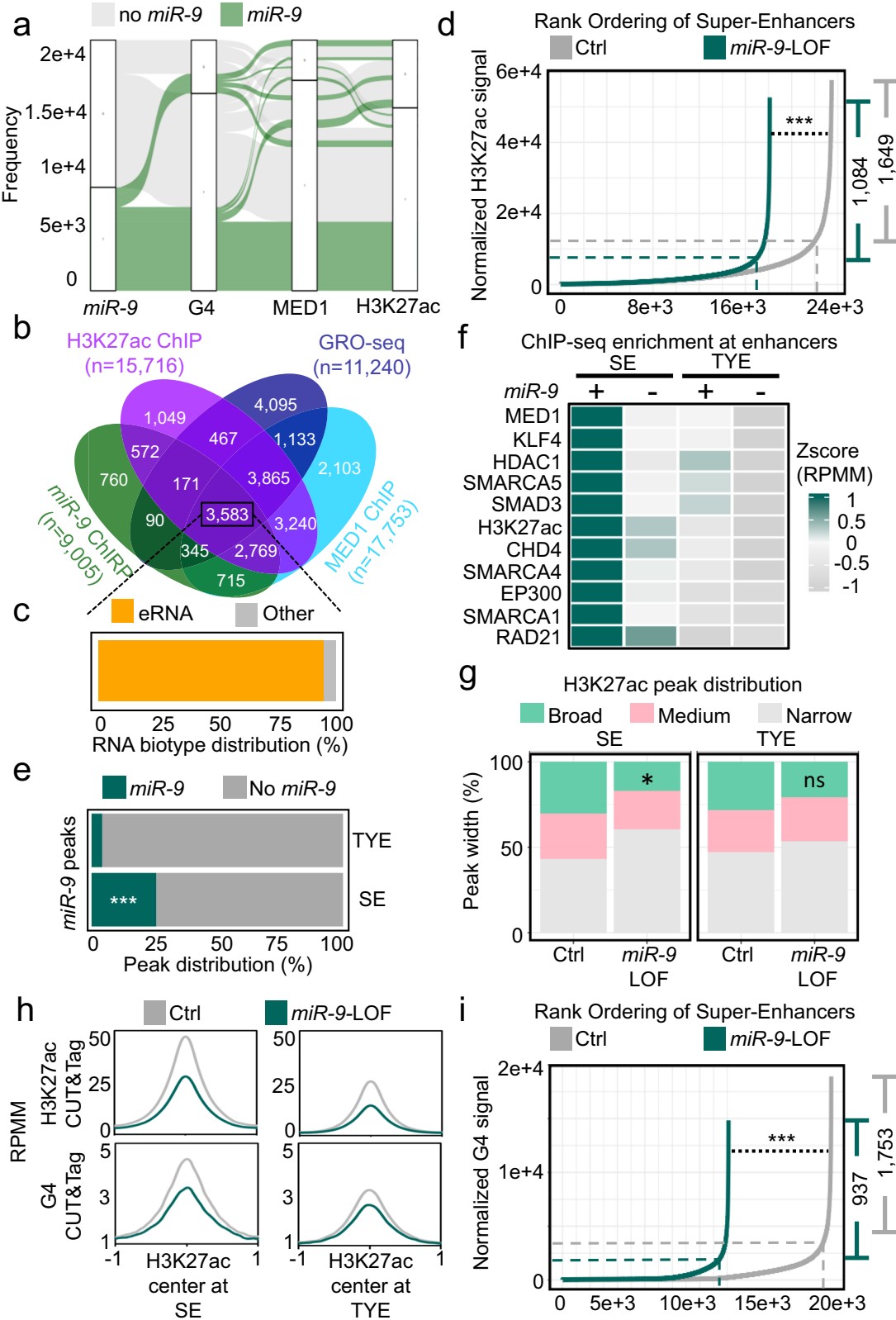

(n = 3423) of these transcripts were found in the animal eRNA database[47] as enhancer RNAs (eRNA) (Fig. 5c). To investigate a potential role of *miR-9* in enhancers, we performed H3K27ac CUT&Tag in Ctrl or *miR-9* antagomiR-transfected MLg cells (Supplemenatary Fig. 6a) and analyzed the data using the rank-ordering of super-enhancers (ROSE) algorithm[11] to separate SE from typical enhancer (TYE) (Fig. 5d). We detected 1649 SE in Ctrl transfected MLg cells that

were significantly reduced to 1084 (*P* = 9.9E-142) after *miR-9*-LOF supporting the requirement of *miR-9* for SE formation. Moreover, crossing the results obtained by applying the ROSE algorithm to the H3K27ac CUT&Tag data with our *miR-9* ChIRP-seq data (Fig. 1d) showed that 25.5% of the SE were enriched with *miR-9* (*P* = 1.5E-5), whereas only 4% of TYE were enriched with *miR-9* (Fig. 5e). These results were confirmed by further analysis of the *miR-9* ChIRP-seq data

**Fig. 5 | Nuclear *miR-9* is enriched at super-enhancers. a** Alluvial plot showing loci with *miR-9* enrichment (ChIRP-seq, green) also with enrichment of G4 (G4P ChIP-seq) and SE markers MED1 and H3K27ac (ChIP-seq). No-*miR-9*, loci without *miR-9*. **b** Venn diagram showing common loci (*n* = 3583) with *miR-9* enrichment (ChIRP-seq, green), H3K27ac (ChIP-seq, purple) and MED1 (ChIP-seq, turquoise) and nascent RNA (GRO-seq, blue). **c** RNA biotype distribution of transcripts related to the common loci in (**b**) showing that 3423 transcripts (95.5%) were found in a database as enhancer RNAs. **d** Hockey stick plot after analysis using the ROSE algorithm and showing distribution of normalized H3K27ac CUT&Tag signal across typical enhancers (TYE) and super-enhancers (SE) in Ctrl (gray line) or *miR-9* antagomir (green line) transfected MLg cells. **e** Bar plot showing percentage of TYE and SE with or without *miR-9* in *Ctrl*-transfected MLg cells after cross analysis of the results obtained in (**d**) with the results obtained by *miR-9* ChIRP-seq from Fig. 1d. **f** Heat map showing significant enrichment of the indicated proteins by ChIP-seq at SE and

TYE that are also enriched with *miR-9* (green) or not (No *miR-9*, gray). Values, z-Score of the normalized reads counts from annotatePeaks.pl from HOMER. Indicated proteins were previously related to SE in murine cells. All selected proteins showed a significant *P*-value (*P* < 0.001) after two-tailed Willcox test. **g** Bar plots showing H3K27ac peak distribution after ROSE analysis in MLg cells transfected as in (**d**) at SE and TYE divided by the size of the enhancers (SE-broad >3.2 kb, SE-medium >2.3 kb <3.2 kb, SE-narrow <2.3 kb, TYE-broad >0.9 kb, TYE-medium >0.6 kb <0.9 kb and TYE-narrow <0.6 kb). **h** Aggregate plots after CUT&Tag in MLg cells transfected as in (**d**) showing the enrichment of H3K27ac and G4 at SE and TYE. Data were normalized using RPMM. **i** Hockey stick plot after analysis of G4 CUT&Tag as in (**d**). In all plots, asterisks, *P*-values after two-tailed Willcox test (Hockey stick plots) or two-tailed Fisher´s exact test (bar plots), ***$P \leq 0.001$; *$P \leq 0.05$; ns, non-significant. See also Supplementary Fig. 6 and Supplementary Data 1. Source data are provided as a Source Data file.

together with publicly available ChIP-seq data of proteins that have been related to enhancers. Aggregate plots (Supplementary Fig. 6b, c) and a heat map (Fig. 5f) showed significant enrichment of *miR-9* at SE together with H3K27ac, MED1, KLF4, HDAC1, SMAD3, SMARCA4, EP300, SMARCA1, RAD21, MYC, MEIS1, BRG1, HDAC2, and EST1. Interestingly, we also detected significant enrichment of *miR-9* and all the analyzed proteins at TYE. However, the enrichment of all these proteins was significantly higher at SE that also contained *miR-9* as compared to SE without *miR-9* or TYE with/without *miR-9*. In addition, we observed enrichment of CHD4 and SMARCA5 at the same SE as *miR-9*. Supporting these observations, we found that *miR-9* pulled down endogenous CHD4 and SMARCA5 by miR-Pd followed by Western Blot (Supplementary Fig. 6d), whereas both proteins precipitated endogenous *miR-9* by Ch-RIP followed by *miR-9*-specific TaqMan assays (Supplementary Fig. 6e). Our results suggest CHD4 and SMARCA5 as components of SE. Further analysis of the H3K27ac CUT&Tag showed that the breadth of H3K27ac domains significantly decreased after *miR-9*-LOF at SE of MLg cells (Fig. 5g). In addition, the levels of H3K27ac and G4 were higher at SE than at TYE (Fig. 5h). However, *miR-9*-LOF reduced the levels of H3K27ac and G4 at both enhancer types. Remarkably, analysis of the G4 CUT&Tag using the ROSE algorithm showed that the number of SE containing G4s (1753) was significantly reduced (937, *P* = 2.5E-35) after *miR-9*-LOF (Fig. 5i) suggesting a role of G4s in SE formation in a *miR-9*-dependent manner. Supporting these results, aggregate plots using G4-specific ChIP-seq data[18] showed G4 enrichment at the same SE as *miR-9* in mouse lung fibroblasts (MLg cells) and adenocarcinoma human alveolar basal epithelial cells (A549 cells) (Supplementary Fig. 6f).

Loci visualization of selected SE with *miR-9* enrichment using the IGV genome browser (Fig. 6a top and Supplementary Fig. 7, top) confirmed the enrichment of G4 at the same loci, as well as of MED1, KLF4 and H3K27ac[45,46], which are markers of SE. Remarkably, CUT&Tag experiments in MLg cells revealed that enrichment of G4 and H3K27ac was reduced a these loci after *miR-9*-LOF, supporting the requirement of *miR-9* for the correct levels of these chromatin features at SE. Interestingly, zooming into the loci revealed G-rich sequences that favor the formation of G4[4,15] (Fig. 6a, bottom, and Supplementary Fig. 7, bottom). These results were confirmed by analysis of the loci of these SE by ChIP followed by qPCR using H3K4me3- or G4-specific antibodies and chromatin from MLg cells that were transiently transfected with Ctrl or *miR-9*-specific antagomiR probes (Fig. 6b). We detected H3K4me3 and G4 enrichment at the loci of all analyzed SE in Ctrl antagomiR-transfected cells, which was significantly reduced after *miR-9*-LOF. Our results demonstrate that SE with *miR-9* enrichment are also enriched with H3K4me3 and G4 in a *miR-9*-dependent manner.

### Promoter-super-enhancer looping of TGFB1-responsive genes requires *miR-9*

To further elucidate the biological relevance of our findings we performed GSEA[44] on the loci with *miR-9* enrichment and nascent RNA as

determined by *miR-9* ChIRP-seq and GRO-seq, respectively (Fig. 7a, b). We found significant enrichment of genes related to the categories "TGFB cell response" (*P* = 7.55E-09), "TGFB" (*P* = 2.61E-08), "Cell proliferation" (*P* = 4.66E-08), and "Fibroblasts proliferation" (*P* = 1.61E-07), suggesting an involvement of the loci with *miR-9* enrichment and nascent RNA in these biological processes. Supporting these observations, RNA FISH and immunostaining in MLg cells showed that TGFB1 treatment increased the levels of *miR-9*, G4 and H3K4me3 in *miR-9*-dependent manner (Fig. 7c, d). Similarly as in IPF hLF (Supplementary Fig. 1d), the majority of *miR-9* was detected in the cell nucleus of MLg cells after TGFB1 treatment. Interestingly, we detected significantly increased enrichment of *miR-9* at promoters of *miR-9* target genes after TGFB1 treatment in MLg cells that were analyzed by qPCR after ChIRP using *miR-9*-specific biotinylated antisense oligonucleotides (Fig. 7e). These results were complemented by H3K4me3 ChIP-seq in MLg cells that were transiently transfected with Ctrl or *miR-9*-specific antagomiR probes, and non-treated or treated with TGFB1 (Fig. 7f–h and Supplementary Fig. 8a–c). H3K4me3 levels significantly increased after TGFB1 treatment at promoters of TGFB-responsive genes in *miR-9*-dependent manner (Fig. 7f, left). These effects were not observed at the promoters of genes that did not respond to TGFB1 treatment (Fig. 7f, right) supporting the specificity of the effects observed. By checking on H3K4me3 levels at loci, in which we detected G4s by CUT&Tag, we observed that TGFB1 did not significantly affect H3K4me3 levels, whereas the combination of TGFB1 and *miR-9*-LOF reduced H3K4me3 levels at these loci (Supplementary Fig. 8b). Interestingly, the breadth of H3K4me3 peaks increased after TGFB1 treatment also in *miR-9*-dependent manner (Fig. 7g). Further, loci visualization of the selected *miR-9* target genes (*Zdhhc5*, *Ncl*, *Lzts2*, *Hdac7* and *Ep3O0*) using the IGV genome browser showed H3K4me3 enrichment at the promoters in non-treated, and Ctrl antagomiR transfected MLg cells that increased after TGFB1 treatment (Fig. 7h and Supplementary Fig. 8c). However, a combination of TGFB1 treatment and *miR-9*-specific antagomiR transfection showed that *miR-9*-LOF counteracted the effect caused by TGFB1 demonstrating the requirement of *miR-9* for the chromatin changes induced by TGFB1 and suggesting its requirement for TGFB1-inducibility of the analyzed genes, as shown below.

Since we observed enrichment of *miR-9*, H3K4me3 and G4 not only at promoters of *miR-9* target genes (Figs. 1–4), but also in specific SE (Figs. 5–6), we decided to analyze the genome-wide effect of TGFB1 on chromatin conformation by a technique that combines an in situ Hi-C library preparation with a chromatin immunoprecipitation (HiChIP, Fig. 8a). For this HiChIP-seq we used H3K4me3-specific antibodies to precipitate SE that physically interact with active promoters and chromatin from MLg cells that were transiently transfected with Ctrl or *miR-9*-specific antagomiR probes, and non-treated or treated with TGFB1 (Fig. 8b–e and Supplementary Fig. 9a–d). We detected a significantly increased number of chromatin interaction hubs after TGFB1 treatment (Fig. 8b). Furthermore, up to 65.1% (*P* = 0.02) of these

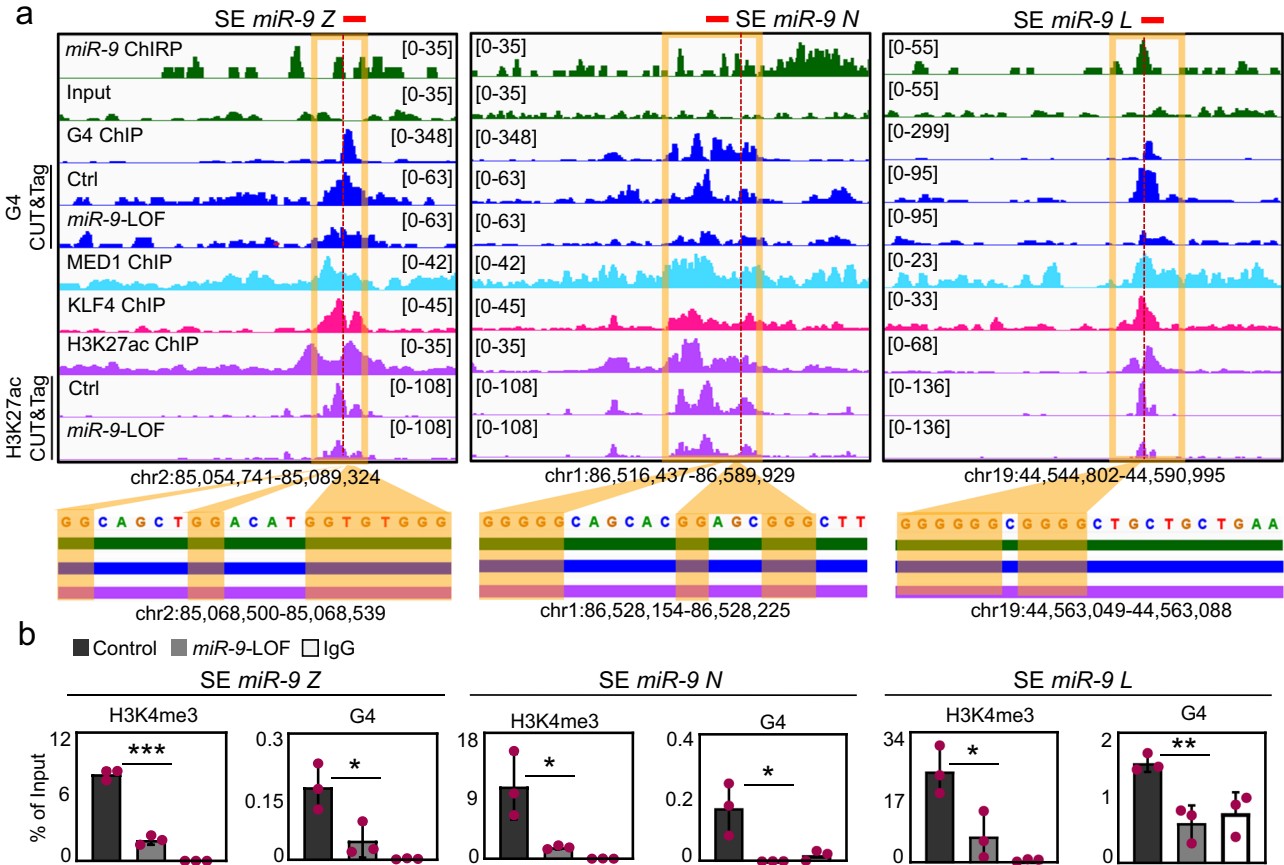

**Fig. 6 | Nuclear *miR-9* is enriched at super-enhancers and is required for G-quadruplexes. a** Visualization of selected SE with *miR-9* enrichment using IGV genome browser showing enrichment *miR-9* by ChIRP-seq (green), G4 by G4P ChIP-seq in NIH/3T3 cells (blue), G4 by CUT&Tag in Ctrl and *miR-9*-specifc antagomiR transfected MLg cells (blue), MED1 (turquoise), KLF4 (magenta) and H3K27ac (purple) in by ChIP-seq in mouse embryonic fibroblasts, H3K27ac by CUT&Tag in Ctrl and *miR-9*-specifc antagomiR transfected MLg cells (purple). Reads were normalized using reads per kilobase per million (RPKM). Images show the indicated gene loci with their genomic coordinates. Orange squares, regions with enrichment of *miR-9*, G4 and SE markers; red lines, regions selected for single gene analysis in (**b**); dotted lines, regions shown at the bottom with high G content. Bottom, green line, *miR-9* enrichment; blue line, G4 enrichment; purple line, H3K27ac enrichment. **b** Analysis of selected SE with *miR-9* enrichment by ChIP using chromatin from MLg cells transfected with control (Ctrl) or *miR-9*-specific antagomiR to induce *miR-9* loss-of-function (LOF). Bar plots presenting data as means; error bars, s.e.m (*n* = 3 biologically independent experiments); asterisks, *P*-values after two-tailed t-test, ***$P \leq 0.001$; **$P \leq 0.01$; *$P \leq 0.05$. See also Supplementary Figs. 6, 7. Source data are provided as a Source Data file.

chromatin interaction hubs were also enriched with miR-9 (Fig. 8c). Interestingly, *miR-9*-LOF counteracted the effect induced by TGFB1 treatment. Further analysis of the H3K4me3-specific HiChIP-seq data by k-means clustering revealed four clusters (Fig. 8c). We focused on clusters 1 and 4 since we observed an increase of chromatin interactions in response to TGFB1 treatment in a *miR-9*-dependent manner in these two clusters. Interestingly, these two clusters comprise the loci of the *miR-9* target genes (*Zdhhc5*, *Ncl*, *Lzts2* and *Hdac7*). Further, we generated IGV genome browser snapshots to visualize the enrichment of *miR-9*, G4, MED1, KLF4 and H3K27ac at the loci of promoters of *miR-9* target genes and SE with *miR-9* enrichment (Fig. 8d, top). In the same snapshots, we present the results of the H3K4me3-specific HiChIP-seq (Fig. 8d, bottom, and Supplementary Fig. 9d) showing chromatin loops between the promoters of *miR-9* target genes and SE with *miR-9* enrichment in Ctrl antagomiR-transfected cells. Strikingly, these promoter-SE-loops increased in TGFB1 treated cells in a *miR-9*-dependent manner, since *miR-9*-LOF counteracted the effect caused by TGFB1. To correlate the results from H3K4me3-specific HiChIP-seq with changes in chromatin, we analyzed the promoters and SE with *miR-9* enrichment by ChIP qPCR using H3K4me3- and G4-specific antibodies and chromatin from MLg cells that were transfected with Ctrl or *miR-9*-specific antagomiRs, and non-treated or treated with TGFB1 (Fig. 9a). TGFB1 treatment significantly increased H3K4me3 and G4 levels at the analyzed promoters and SE. Further, *miR-9*-LOF significantly reduced the effect caused by TGFB1 treatment, whereas *miR-9*-LOF alone significantly reduced H3K4me3 and G4 levels when compared to Ctrl antagomiR transfected cells. To correlate these changes in chromatin structure with gene expression, we analyzed the expression of *miR-9* target genes by qRT-PCR in MLg cells under the same conditions as specified above (Fig. 9b). The expression of all analyzed *miR-9* target genes significantly increased after TGFB1 treatment in *miR-9*-dependent manner, correlating with our chromatin structure analysis (Figs. 7f-g and 9a). Further, *miR-9*-LOF alone significantly reduced the basal transcription levels of the analyzed *miR-9* target genes as compared to Ctrl antagomiR transfected cells confirming our RNA-seq results (Fig. 2f-h). These qRT-PCR-based results were confirmed by RNA-seq using total RNA from in MLg cells (Fig. 9c). In summary, our results support a model (Fig. 9d), in which G4s are formed in a *miR-9*-dependent manner at both, promoters of TGFB1-responsive genes, as well as SE with which these promoters form loops (left). Further, H3K4me3, G4 and promoter-SE looping increased after TGFB1 treatment allowing these two regulatory elements to come into close physical proximity and enhance transcription of the corresponding genes (middle) also in *miR-9*-dependent manner (right).

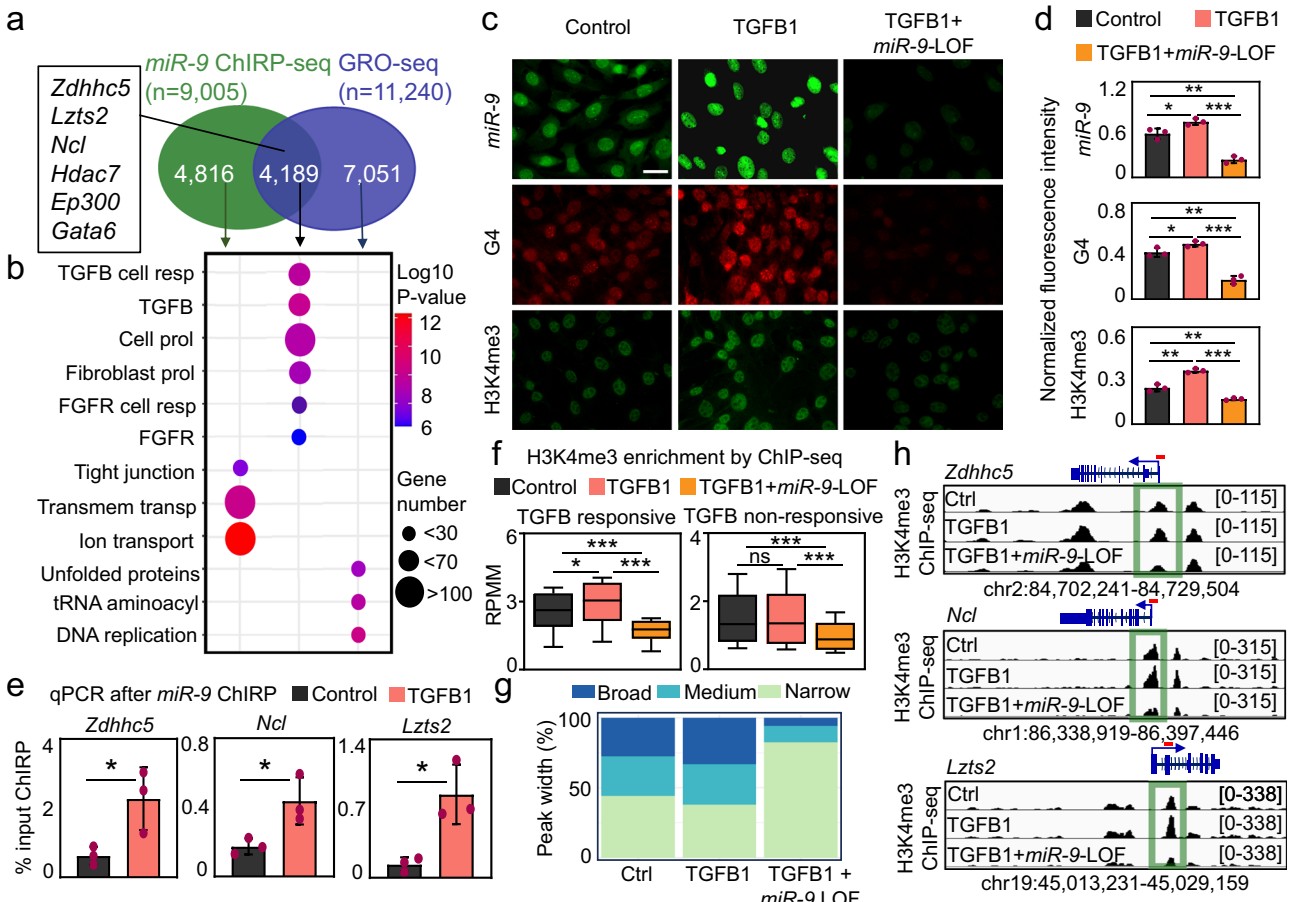

**Fig. 7 | Nuclear *miR-9* is required for H3K4me3 enrichment at promoters of TGFB1-responsive genes. a** Venn diagram after cross analysis of *miR-9* ChIRP-seq and GRO-seq[41] showing loci with *miR-9* enrichment and nascent RNAs (*n* = 4189), in which the selected *miR-9* target genes are included (square). **b** Gene set enrichment analysis (GSEA) of the three loci groups identified in (**a**). Resp, response; prol, proliferation; trans, transport. *P* values after two-tailed Fisher´s exact test and represented as log10. **c**, **d** Fluorescence microscopy of MLg cells after *miR-9*-specific FISH (top), G4- (middle) or H3K4me3-specific (bottom) immunostaining. Cells were transfected with control (Ctrl) or *miR-9*-specific antagomiR to induce a loss-of-function (LOF), and non-treated or treated with TGFB1, as indicated. Representative images from three independent experiments (**c**) and quantification of them (**d**). Scale bars, 10 μm. **e** Promoter analysis of the indicated *miR-9* target genes in non-treated or TGFB1-treated MLg cells by qPCR after *miR-9*-specific ChIRP. **f, g** H3K4me3 CUT&Tag in MLg cells that were treated as in (**c**). **f** Box plots showing

H3K4me3 enrichment at promoters of genes that are responsive (left) or non-responsive (right) to TGFB1. Data were normalized using RPMM. **g** Genome-wide distribution of H3K4me3 peaks relative to broad, medium and narrow H3K4me3 domains. **h** Visualization of selected *miR-9* target genes using IGV genome browser showing enrichment H3K4me3 by ChIP-seq in MLg cells that were treated as in (**c**). Images show the indicated loci with their genomic coordinates. Arrows, transcription direction; green squares, promoter regions; dotted lines, regions selected for single gene analysis in Fig. 4b. Bar plots show data as means; error bars, s.e.m (*n* = 3 biologically independent experiments). Box plots indicate median (middle line), 25th, 75th percentile (box) and 5th and 95th percentile (whiskers). In all plots, asterisks or *P*-values after two-tailed t-test, ***$P \le 0.001$; **$P \le 0.01$; *$P \le 0.05$; ns, non-significant. See also Supplementary Fig. 8. Source data are provided as a Source Data file.

## Discussion

Our study placed a nuclear microRNA in the same structural and functional context with non-canonical DNA secondary structures and 3D genome organization during transcription activation. We uncovered a mechanism of transcriptional regulation of TFGB1-responsive genes that requires nuclear *miR-9* and involves G4s and promoter-SE looping. Various aspects of the model proposed here are interesting. For example, nuclear *miR-9* was neither related to transcription regulation nor chromatin structure prior to our study, even though *miR-9* participates in a wide spectrum of biological functions including AGO2-dependent degradation of the lncRNA *MALAT1* in the cell nucleus[32,34,35]. Previously, we have shown that other nuclear miRNA, *Mirlet7d*, is part of the ncRNA-protein complex MiCEE that mediates epigenetic silencing of bidirectionally transcribed genes and nucleolar organization[6,37]. In this context, nuclear *Mirlet7d* binds ncRNAs expressed from these genes, and mediates their degradation by the RNA exosome complex. It will be the scope of future work to

determine whether nuclear *miR-9* also binds to ncRNA expressed from the *miR-9* target loci and mediates their degradation by a similar mechanism. Following this line of thought, it has been reported that G4s are found in genomic regions containing R-loops[48,49], which are three-stranded nucleic acid structures consisting of a DNA-RNA hybrid and the associated non-template single-stranded DNA[50]. Moreover, we have previously reported that R-loops regulate transcription in response to TGFB1 signaling[5]. Interestingly, when a G4 is formed opposite to the R-loop on the associated single-stranded DNA, a so called G-loop structure is generated[51]. Sato and colleagues recently reported a mechanism involving G-loop structures, in which the transcripts stabilizing the R-loops are relevant for the controlled resolution of the G4s, thereby preventing mutagenic G4s and supporting genomic stability[51]. We will investigate in a future project whether nuclear *miR-9* is involved in a similar mechanism targeting the ncRNAs in R-loops and promoting controlled G4 resolution during transcription initiation.

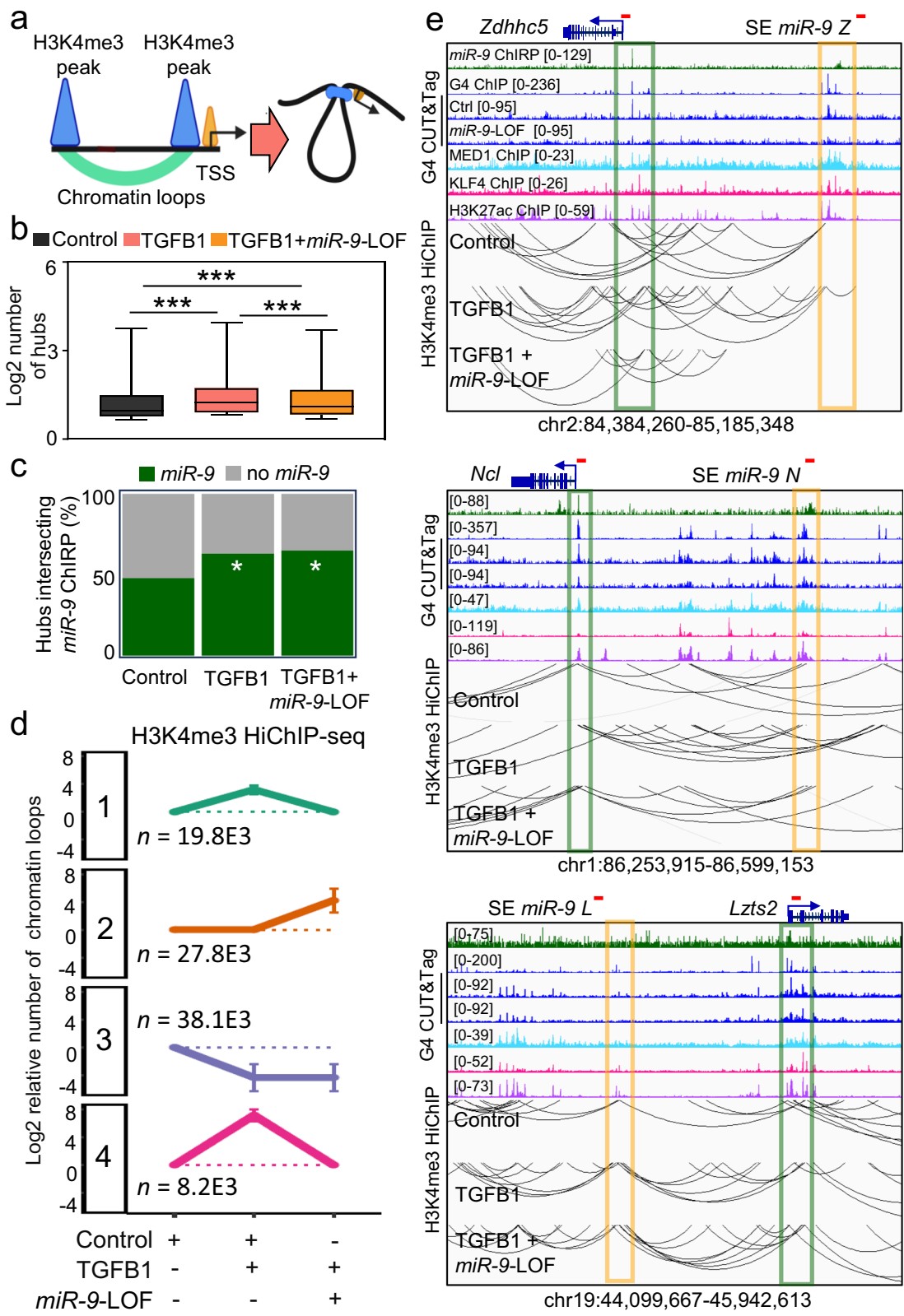

Another interesting aspect of the model proposed here is the participation of G4s in 3D genome organization during transcription activation. DNA G4s are stable four-stranded non-canonical structures that are highly related to promoters and transcription activation[17–19]. Further, a recent publication based on integrative analysis of multi-omics studies have provided comprehensive mechanistic insights into the function of G4s as promoter elements that reduce nucleosome density, increase the levels of active histone marks (H3K4me3 and H3K27ac), generate nucleosome arrays by positioning nucleosomes at a periodic distance to each other and facilitate pause release of Poll II into effective RNA production resulting in enhanced transcriptional activity[22]. Even though similar chromatin features were described to be induced by G4s at promoter-distal regulatory elements, such as SE[52], no mechanistic insight elucidating the role of G4s in chromatin looping

**Fig. 8 | Nuclear *miR-9* is required for chromatin loops at promoters of TGFB1-responsive genes. a** Schematic representation of chromatin loops that are enriched with H3K4me3 and enhance transcription. TSS, transcription start site. Created in BioRender. Rogel, D. (2022) BioRender.com/q42a090. **b** Box plot showing the number of chromatin interaction hubs by H3K4me3-specific HiChIP-seq in MLg cells that were transfected with control (Ctrl) or *miR-9*-specific antagomir (*miR-9*-LOF, loss-of-function), and non-treated or treated with TGFB1, as indicated. Box plot indicates median (middle line), 25th, 75th percentile (box) and 5th and 95th percentile (whiskers). Number of hubs in Ctrl, $n = 34,638$; in TGFB1, $n = 16,361$ and TGFB1+*miR-9*-LOF, $n = 16,668$. Asterisks, *P*-values after two-tailed t-test, ***$P \leq 0.001$. **c** Bar plot showing the percentage of significant chromatin interactions at loci with or without *miR-9* after cross analysis of the results obtained in (**b**) with the results obtained by *miR-9* ChIRP-seq from Fig. 1d. Asterisk, *P*-values after Fisher´s exact test, *$P \leq 0.05$. **d** Line charts showing the number of significant

chromatin interactions at loci with *miR-9* and H3K4me3 enrichment in MLg cells treated as in (**b**). Four clusters were generated using k-means algorithm. Data are represented as log2 of the ratio relative to Ctrl-transfected, non-treated cells. Numbers indicate number of significant chromatin interactions in each cluster; error bars, SD. **e** Visualization of promoters of selected *miR-9* target genes (green squares) and SE with *miR-9* enrichment (orange squares) using IGV genome browser showing enrichment *miR-9* by ChIRP-seq (green), G4 by G4P ChIP-seq (blue), MED1 (turquoise), KLF4 (magenta) and H3K27ac (purple) by ChIP-seq. Reads were normalized using reads per kilobase per million (RPKM) measure and are represented as log2 enrichment over their corresponding inputs. Bottom, chromatin loops by HiChIP-seq in MLg cells treated as in (**b**). Images show the indicated loci with their genomic coordinates. Arrows, transcription direction; red lines, regions selected for single gene analysis in Fig. 9a. See also Supplementary Fig. 9. Source data are provided as a Source Data file.

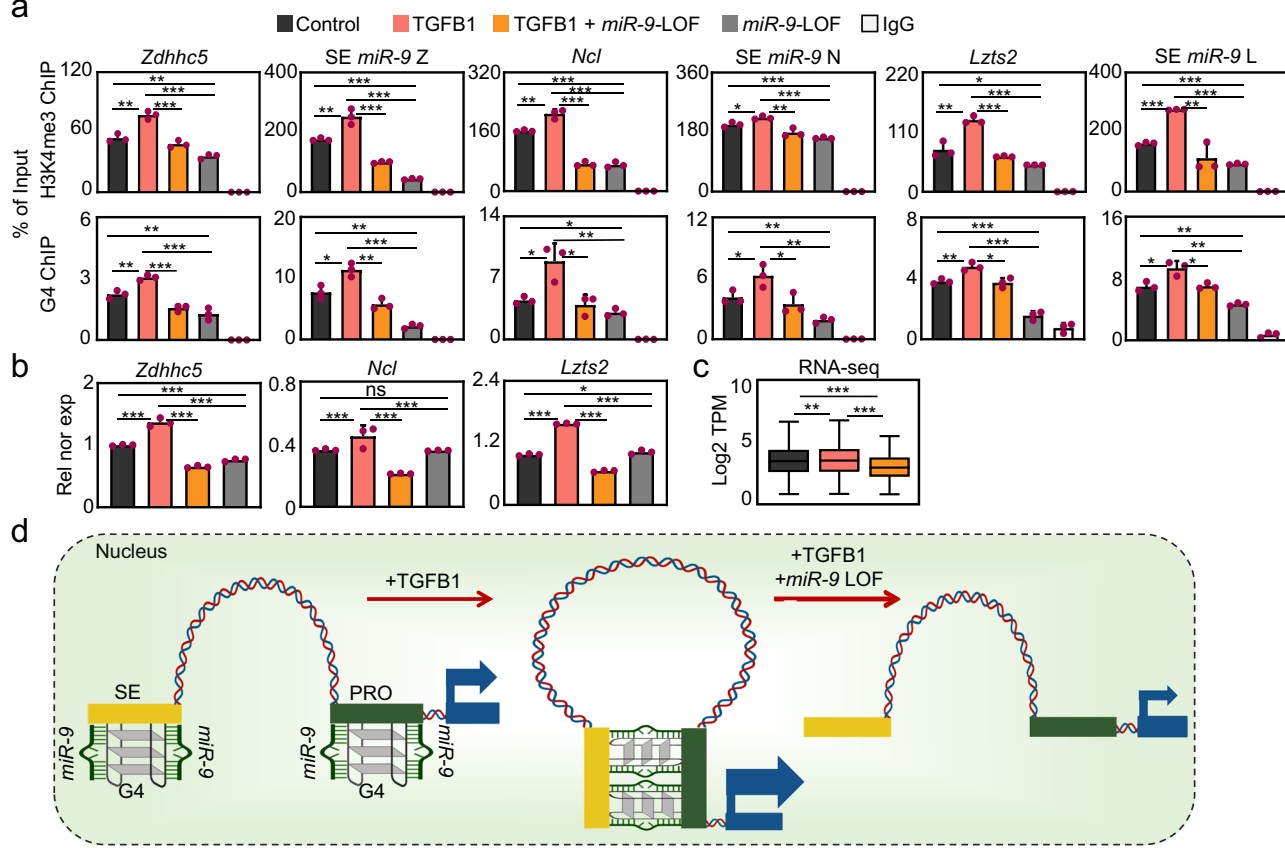

**Fig. 9 | Promoter-super-enhancer looping of TGFB1-responsive genes requires *miR-9*. a** Analysis of the promoters and SE highlighted in Fig. 8b by ChIP using the indicated antibodies and chromatin of MLg cells that were transfected with control (Ctrl) or *miR-9*-specific antagomir (*miR-9*-LOF, loss-of-function), and non-treated or treated with TGFB1, as indicated. **b** Expression analysis of the selected *miR-9* target genes by qRT-PCR in MLg cells treated as in (**a**). All bar plots present data as means; error bars, s.e.m ($n = 3$ biologically independent experiments). **c** RNA-seq using total RNA from in MLg cells treated as in (**a**). Data are presented as Log2 of transcript per million (TPM). Box plot shows median (middle line), 25th, 75th percentile (box) and 5th and 95th percentile (whiskers). Number of genes in all three

conditions is $n = 24,957$. I all plots, asterisks represent *P*-values after two-tailed t-test, ***$P \leq 0.001$; **$P \leq 0.01$; *$P \leq 0.05$; non-significant. See also Supplementary Fig. 9. Source data are provided as a Source Data file. **d** Model summarizing the results presented in the manuscript. Left, G4 are formed in *miR-9* (red lines)-dependent manner at SE (orange box) and promoters (green box) of TGFB1-responsive genes (blue box, coding region). Middle, TGFB1treament increases euchromatin histone mark H3K4me3, G4 and chromatin loops bringing SE and promoter to close physical proximity, thereby enhancing transcription (arrow) of the corresponding gene. Right, *miR-9*-LOF antagonizes the effects induced by TGFB1. Created in BioRender. Rogel, D. (2023) BioRender.com/c64w876.

mediating long-range enhancer-promoter interactions was available prior to our study. Our results demonstrate physical interaction of *miR-9* with G4s, as well as with proteins that are known to interact with G4s (Fig. 3h, i)[43]. In addition, we found that G4s are formed in a *miR-9*-dependent manner (Figs. 3g, 4, 5h, i and 6) at promoters of TGFB1-responsive genes, as well as SE with which these promoters form loops

(Fig. 8d). Interestingly, TGFB1 treatment of MLg cells (1) increased *miR-9* levels in the cell nucleus and more specifically at promoters of *miR-9* target genes (Fig. 7c–e), (2) augmented H3K4me3 levels at promoters of TGFB1-responsive genes and the breadth of H3K4me3 domains (Fig. 7f–h), (3) increased chromatin interaction hubs and promoter-SE looping (Fig. 8) allowing these two regulatory elements to come into

close physical proximity and enhance transcription of the corresponding genes (Fig. 9b). All these effects in the chromatin structure after TGFB1 treatment occurred in a *miR-9*-dependent manner. One can hypothesize that G4 may stabilize the interaction between these two regulatory elements, allowing for increased transcription. We will investigate this hypothesis and whether the G4 are formed with DNA strands from the promoter and from the SE in a future project.

Our *miR-9*-LOF experiments showed that *miR-9* is required for H3K4me3 broad domains (Fig. 2a–e), basal transcriptional activity (Fig. 2e–g) and TGFB1-inducibility (Fig. 9b) of *miR-9* target genes. H3K4me3 is a well-characterized euchromatin histone mark related to genes with high transcriptional activity, probably contributing to release of Poll II pausing into elongation[39]. Furthermore, it has been shown that genes with broad domains of H3K4me3 are transcriptionally more active than genes with narrow domains[39,53], which is consistent with our findings. Interestingly, H3K4me3 broad domains have been linked to genes that are critical to cellular identity and differentiation[39,38,53,54]. Moreover, H3K4me3 broad domains are associated with increased transcription elongation and enhancer activity, which together lead to exceptionally high expression of tumor suppressor genes, including TP53 and PTEN[39]. On the other hand, TGFB signaling is one of the prominent pathways implicated in hyperproliferative disorders, including cancer[27,37,55–57]. Our results in primary, patient-derived hLF showing *miR-9* in the cell nucleus of IPF hLF, whereas in Ctrl hLF the majority of *miR-9* was in the cytosol, suggest a translocation mechanism of *miR-9* into the cell nucleus that is related to IPF, which in turn is linked to TGFB1 signaling[37]. It will be the scope of our future work to determine the clinical relevance of the model of transcription regulation proposed here within the context of hyperproliferative disorders with special attention on IPF.

## Methods

### Cell culture

Mouse lung fibroblast cells MLg (ATCC CCL-206) and MFML4[58] were cultured in complete DMEM (4.5 g/L glucose, 10% FCS, 1% Pen-Strep, 2 mM L-glutamine) at 37 °C in 5% $CO_2$. Mouse lung epithelial cells MLE-12 (ATCC CRL-2110) were cultured in complete DMEM/F12 (5% FCS, 1% Pen-Strep) at 37 °C in 5% $CO_2$. Mouse mammary gland epithelial cells NMuMG (ATCC CRL-1636) were cultured in complete DMEM (4.5 g/L glucose, 10 µg/mL insulin (90%), 10% FCS, 1% Pen-Strep) at 37 °C in 5% $CO_2$. Human primary lung fibroblasts from control donors were cultured in complete MCDB131 medium (8% fetal calf serum (FCS), 1% L-glutamine, penicillin 100 U/ml, streptomycin 0.1 mg/ml, epidermal growth factor 0.5 ng/ml, basic fibroblast growth factor 2 ng/ml, and insulin 5 µg/ml)) at 37 °C in 5% $CO_2$. During subculturing, cells were 1x PBS washed, trypsinized with 0.25% (w/v) Trypsin and split at the ratio of 1:5 to 1:10. The cell lines used in this paper were mycoplasma free. They were regularly tested for mycoplasma contamination. In addition, they are not listed in the database of commonly misidentified cell lines maintained by ICLAC.

### Cell transfection, treatment and antagomiR-mediated *miR-9* loss-of-function

Cells were transfected with antagomiR probes (Ambion) using Lipofectamine 2000 (Invitrogen) following the manufacturer's instructions, and harvested 24 h later for further analysis. Anti-hsa-miR-*Mir-9-5p* (Ambion, #17000), and Anti-miR negative control (Ambion, #17010) were transfected at 60 nM final concentrations. Following 20 h after transfection, TGFB1 signaling was induced with 5 ng/ml final concentration of human recombinant TGFB1 (Sigma-Aldrich) for 4 h.

### Bacterial culture and cloning

For cloning experiments, chemically competent *E. coli* TOP10 (ThermoFisher Scientific) were used for plasmid transformation. TOP10 strains were grown in Luria broth (LB) at 37 °C with shaking at 180 rpm for 16 h or on LB agar at 37 °C overnight.

### RNA isolation, reverse transcription, quantitative PCR and TaqMan assay

Expression analysis by qRT-PCR were performed as previously described[59]. Briefly, total RNA from cell lines was isolated using the RNeasy Mini kit (Qiagen) and quantified using a Nanodrop Spectrophotometer (ThermoFisher Scientific). Synthesis of complementary DNA was performed using 1–2 µg total RNA and the High Capacity cDNA Reverse Transcription kit (Applied Biosystems). Quantitative real-time PCR reactions were performed using SYBR® Green on the Step One plus Real-time PCR system (Applied Biosystems). Housekeeping gene *Gapdh* was used to normalize gene expression. Primer pairs used for gene expression analysis are described in the Supplementary Table 1.

For *miR-9* expression analysis, total RNA was isolated with Trizol (Invitrogen), quantified using a Nanodrop Spectrophotometer (ThermoFisher Scientific), 0.5–2 µg total RNA was used for reverse transcription (High-Capacity cDNA Reverse Transcription Kit, ThermoFisher Scientific) and subsequently *miR-9*-specific TaqMan assay (Applied Biosystems) in the Step One plus Real-time PCR system (Applied Biosystems). All bar plots with individual dots were generated using GraphPad Prism 8 software.

### MiRNA fluorescence in situ hybridization

MiRNA Fluorescence in situ hybridization (miRNA-FISH) was performed as described earlier[6] with minor adaptations. Briefly, cells were fixed with 4% PFA, dehydrated with 70% ethanol and incubated with pre-hybridization buffer (50% formamide, 5X SSC, 5X Denhardt's solution, 200 µg/ml yeast RNA, 500 µg/ml salmon sperm DNA and 2% Roche blocking reagent in DEPC treated water). Incubation with pre-hybridization buffer was carried out for 4 h at room temperature. Pre-hybridization buffer was replaced with denaturing hybridization buffer (10% CHAPS, 20% Tween, 50% formamide, 5X SSC, 5X Denhardt's, 200 µg/ml yeast RNA, 500 µg/ml salmon sperm DNA and 2% Roche blocking reagent in DEPC treated water) containing biotin and Locked Nucleic Acid (LNA™) probes (Exiqon) specific to mature *miR-9* to a final concentration of 20 pM and incubated at 55 °C overnight. Next day, cells were briefly washed with 5X SSC buffer pre-warmed to 60 °C and then incubated with 0.2X SSC at 60 °C for 1 h. Later, cells were incubated with B1 solution (0.1 M Tris pH 7.5, 0.15 M NaCl) at room temperature for 10 min. B1 solution was then replaced with blocking solution (10% FCS, 0.1 M Tris pH 7.5, 0.15 M NaCl) and incubated for 1 h at RT. Cells were then incubated with FITC labeled rabbit anti-Biotin (Abcam) antibody overnight at 4 °C. DAPI was used as nuclear dye. Cells were examined with a fluorescence microscope (Leica DMI300 B) and/or confocal microscope (Leica SP5-X).

### Immunofluorescence and confocal microscopy

Immunostaining was performed as previously described[37]. Briefly, cells were grown on coverslips, fixed with 4% PFA for 10 min at RT and permeabilized with 0.4% Triton X-100 in 1x PBS for 10 min at RT. During immunostaining procedure, all incubations and washes were performed with histobuffer containing 3% bovine serum albumin (BSA) and 0.2% Triton X-100 in 1x PBS, pH 7.4. Non-specific binding was blocked by incubating with 5% BSA in 1x PBS, pH 7.4. Cells were then incubated with primary antibodies overnight at 4 °C. After 3 washes with histobuffer (15 min each), secondary antibody was incubated at RT for 1 h followed by DAPI nuclear staining (Sigma, Germany). Immunostainings were examined with an immunofluorescence microscope (Leica DMI300 B) and/or confocal microscope (Leica

SP5-X). Antibodies used were specific for H3K4me3 (Abcam, ab8580). Alexa 488 or Alexa 594 tagged secondary antibodies (Invitrogen, Germany, dilution 1:1000) were used. DAPI (Sigma, Germany) was used as nuclear dye.

## Chromatin isolation by miRNA purification and sequencing

Chromatin isolation by miRNA purification (ChIRP) was performed as described[60], with slight modifications. Briefly, MLg and MLE-12 cells were cross-linked by 1% formaldehyde for 10 min, lysed, and sonicated with a Diagenode Bioruptor to disrupt and fragment genomic DNA. After centrifugation, chromatin was incubated with 100 pmol of biotin-labeled anti-sense LNA™ probes (Exiqon) specific to mature Mirlet7d, or control, at 37 °C for 4 h. Streptavidin-magnetic C1 beads (Invitrogen) were blocked with 500 ng/μL yeast total RNA and 1 mg/mL BSA for 1 h at room temperature, and washed three times in nuclear lysis buffer (2 mM Tris-HCl, pH 7.0, 250 mM NaCl, 2 mM EDTA, 2 mM EGTA, 1% Triton X-100, 0.2 mM DTT, 20 mM NaF, 20 mM Na3VO4, 40 μg/mL phenylmethylsulfonyl fluoride, protease inhibitor, and RNase inhibitor in DEPC-treated water) and resuspended in its original volume. We added 100 μL of washed/blocked C1 beads per 100 pmol of probes, and the whole reaction was mixed for another 30 min at 37 °C. Beads:biotinprobes:RNA:chromatin adducts were captured by magnets (Invitrogen) and washed five times with 40×bead volume of wash buffer. DNA was eluted with a cocktail of 100 μg/mL RNase A (ThermoFisher Scientific). Chromatin was reverse cross-linked at 65 °C overnight. Later, DNA was purified using the QIAquick PCR Purification Kit (Qiagen) according to manufacturer's instructions and used for single gene promoter analysis by qPCR and for sequencing. Primer pairs used for qPCR analysis are described in in the sub-section "Chromatin immunoprecipitation" below. ChIRP-seq was performed by single-end sequencing on an Illumina HiSeq2500 machine at the Max Planck-Genome-Centre Cologne. Raw reads were trimmed using Trimmomatic-0.36 with the parameters (ILLUMINACLIP:${ADAPTERS}:2:30:10 LEADING:3 TRAILING:3 SLIDINGWINDOW:4:15 MINLEN:20 CROP:70 HEADCROP:10 (https://doi.org/10.1093/bioinformatics/btu170). Trimmed reads were mapped to mouse genome mm10 using Bowtie2 (default settings)[61]. Next, PCR duplicates were removed from the BAM files using the MarkDuplicates.jar tool from Picard (version 1.119). MCAS14[62] was used (macs14 -t ChIRP -c Inp -f BAM -p 1e-3 -g 1.87e9 --nomodel --shiftsize 100 and 30 -n). Peak lists were merged with bedtools merge (default settings). Peaks were annotated using annotatePeaks.pl from HOMER[63]. BAM files were converted to bigwig files by the help of bamCoverage from deeptools (-bs 20 --smoothLength 40 -p max --normalizeUsing RPKM -e 150)[64]. The cis-regulatory element annotation system (CEAS)[65] was used to determine the distribution of the peaks from *miR-9* ChIRP-seq in different genomic areas (intron, exon, 5′ UTR, 3′ UTR, promoter, intergenic, and downstream).

## Motif analysis of ChIRP and G4-ChIP-seq data

MEME Suite (Motif-based sequence analysis tools)[66] was used for de-novo DNA motif search analysis. The *miR-9* ChIRP-seq data file containing peaks annotated near the promoter of the genes was used for the motif search. The G4 ChIP-seq[18] file containing peaks annotated near the promoter of the genes was used for the motif search was used for the motif search analysis. The settings in the de-novo DNA motifs search were: (1) a normal enrichment mode to search the motif, (2) length of the motifs allow between 10 and 25 bp widths, (3) motif site distribution of repetition on the fasta file and (4) three maximum motifs to report.

## Super-Enhancer and related enhancer areas enriched with *miR-9*

To find enhancer and Super-Enhancer, the Program Rank Ordering of Super-Enhancer (ROSE) was used (default settings). To determine the potential super-enhancer marked or not marked by *miR-9*, we crossed the *miR-9* ChIRP-seq, with the output results from ROSE of H3K27ac peaks from MLg Ctrl cells using bedtools intersect. If at least a peak of *miR-9* overlap with an H3K27ac peak that type of enhancer will be considered as marked by *miR-9*. overlap with H3K27ac.

All ChIP-seq and *miR-9* ChIRP-seq were quantified from the center of both Peak list by the help of annotatePeaks from HOMER with the settings: annotatePeaks.pl list.bed mm10 -size 4000 -norm 1,000,000 -hist 10 -d maketagLib_Samples > output_quanty.txt. The results of this command were used as input in a custom R-script to produce the aggregate plots to normalized by Z-score of the enrichment of the protein on the type of enhancer area by the help of a custom Script in R. (https://github.com/jcorderJC12/001nuMir9_G4_3D).

## Chromatin immunoprecipitation (ChIP)

ChIP analysis was performed as described earlier[5,67] with minor adaptations. Briefly, cells were cross-linked with 1% methanol-free formaldehyde (ThermoFisher Scientific) lysed, and sonicated with Diagenode Bioruptor to an average DNA length of 300–600 bp. After centrifugation, the soluble chromatin was immunoprecipitated with 3 μg of antibodies specific for H3K4me3 (Abcam, # ab8580), DNA G-quadruplex structures, clone BG4 (Millipore, # MABE917), and IgG (Santa Cruz, #sc-2027). Reverse crosslinked immunoprecipitated chromatin was purified using the QIAquick PCR purification kit (Qiagen) and subjected to ChIP-quantitative PCR. The primer pairs used for gene promoter and super-enhancer regions are described in the Supplementary Table 2.

## Chromatin RNA immunoprecipitation

Chromatin RNA immunoprecipitation (Ch-RIP) analysis was performed as described 47 with minor adaptations. Briefly, cells were cross-linked by 1% formaldehyde for 10 min, lysed, and sonicated with Diagenode Bioruptor to disrupt and fragment genomic DNA. After centrifugation, the soluble chromatin was immunoprecipitated using antibodies. Precipitated chromatin complexes were removed from the beads by incubating with 50 μl of 1% SDS with 0.1 M NaHCO3 for 15 min, vortexing every 5 min and followed by treatment with DNase I. Reverse cross-linked immunoprecipitated materials were used for RNA isolation by Trizol (Invitrogen). Isolated RNA was subjected to cDNA synthesis and further for TaqMan assay or qPCR. Primer pairs used for Ch-RIP analysis are described in Supplementary Table 2.

## Cleavage under targets and tagmentation (CUT&Tag)

CUT&Tag experiments were performed as described previously[49,68] using antibodies specific for DNA G-quadruplex structures, clone BG4 (Millipore, # MABE917), H3K4me3 (Abcam, # ab8580), and H3K27ac (Abcam, # ab4729). Briefly, $1 \times 10^5$ cells were harvested, washed with wash buffer (20 mM HEPES pH 7.5, 150 mM NaCl, 0.5 mM spermidine), and immobilized to concanavalin A coated beads with incubation at room temperature for 10 min. The bead-bound cells were incubated in 200 μl of primary antibody buffer (wash buffer with 1% BSA, 2 mM EDTA, and 0.05% digitonin for gentle permeabilization of the plasma and nuclear membrane) with 1:100 primary antibody dilution at 4 °C by rotating overnight. The next day, the primary antibody buffer was removed and cells were washed with 800 μl of dig-wash buffer (wash buffer with 1% BSA and 0.05% digitonin) three times. After washing, BG4 antibody-incubated cells were resuspended in 200 μl of dig-wash buffer with 1:100 dilution of mouse anti-FLAG antibody (Sigma, F1804) and incubated at room temperature for 1 h with slow rotation. Cells were washed with 800 μl of dig-wash buffer briefly three times to remove unbound antibodies. Anti-FLAG treated cells were incubated with 1:100 dilution of rabbit anti-mouse antibody (Sigma, M7023) in 200 μl of dig-wash buffer at room temperature for 1 h with slow rotation. H3K4me3-treated or H3K27ac-treated cells were incubated with guinea pig anti-rabbit antibody (Novus Biologicals, NBP1-72763) in 200 μl of dig-wash buffer at room temperature for 1 h with slow

rotation. After a brief wash with dig-wash buffer as above, cells were resuspended in 200 μl of dig-300 buffer (20 mM HEPES pH 7.5, 300 mM NaCl and 0.5 mM spermidine, 1% BSA and 0.01% digitonin) with 1:200 dilution of pA-Tn5 adapter complex and incubated at room temperature for 1 h with slow rotation. pA-Tn5-bound cells were washed with 800 μl of dig-300 buffer three times, followed by tagmentation in 200 μl of tagmentation buffer (dig-300 buffer with 10 mM MgCl2) at 37 °C for 1 h. After tagmentation, 15 mM EDTA, 500 g/ml proteinase K and 0.1% SDS were added and further incubated at 63 °C for another 1 h to stop tagmentation and digest protein. Genomic DNA was extracted and purified QIAquick PCR purification kit (Qiagen) and subjected for library preparation and paired-end sequencing.

## CUT&Tag sequencing
The TruSeq ChIP Library Preparation Kit (Illumina) was used to perform the CUT&Tag libraries. Instead of gel-based size selection before the final PCR step, libraries were size selected by SPRI-bead-based approach after the final PCR with 18 cycles. In detail, samples were 1st cleaned up by 1x bead: DNA ratio to eliminate residuals from PCR reaction, followed by a 2-sided-bead cleanup step with an initial 0.6x bead:DNA ratio to exclude larger fragments. The supernatant was transferred to a new tube and incubated with additional beads in 0.2x bead:DNA ratio to eliminate smaller fragments, like adapter and primer dimers. Bound DNA samples were washed with 80% ethanol, dried, and resuspended in TE buffer. Library integrity was verified with 2100 Bioanalyzer system (Agilent Technologies).

Trimmed reads were mapped with bowtie2 settings (--local --very-sensitive --no-mixed --no-discordant --phred33 -I 10 -X 700). After mapping SAM files were converted to BAM format by the help of samtools view -Sb. Further PCR duplicated were removed using MarkDuplicates.jar from Picard-tools (v.1.119) and mitochondrial Chr was removed from BAM mapped files using (awk '{if($3!= "chrM" && $3!= "chrUn"){print $0}}'). Peaks calling for H3K4me3 marker MACS3 callpeak, with settings (--broad -g mmu -q 0.001 --keep-dup 1 --fix-bimodal --nomodel --extsize 1000). Peaks for H3K27ac were called with MACS3 callpeak, settings (--broad -g mmu -q 0.01 --keep-dup 1 --fix-bimodal --nomodel --extsize 200) and G4 was called with MCAS14 setting (p1e-3 for the p-value). After call peaks, high signal areas from the mm10 blacklist were removed using bedtools intersect (v2.30.0), settings (-wa -v). Where -a is the peak file and -b is the edited blacklist from mm10 genome from (https://github.com/Boyle-Lab/Blacklist/blob/master/lists/mm10-blacklist.v2.bed.gz).

## Chromatin conformation analysis by in situ Hi-C library preparation followed by chromatin immunoprecipitation (HiChIP)
HiChIP experiments were performed as previously described[69] using antibodies specific for H3K4me3 (Abcam, # ab8580) with the following optimizations: 5−10 million cells were crosslinked with 1% formaldehyde for 10 min at room temperature; prior to restriction digestion, sodium dodecyl sulfate treatment at 62 °C for 10 min; restriction digestion with MboI (New England Biolabs France, R0147M) for 2 h at 37 °C; prior to fill-in reaction, heat inactivation of MboI at 62 °C for 10 min followed by 2 washing steps of pelleted nuclei with 1x fill-in reaction buffer; after fill-in reaction, ligation at 4 °C for 16 h.

## HiChIP sequencing and data analysis
HiChIP-seq paired-end reads were aligned to the mm10 genome, duplicate reads were removed, reads were assigned to MboI restriction fragments, filtered into valid interactions, and the interaction matrices were generated using the HiC-Pro pipeline default settings[70]. The config file of the HiC-Pro was set to allow validPairs at any distance from each other. HiC-Pro valid interaction reads were then used to detect significant interactions using: (1) Maketag libraries done as follows (makeTagDirectory output_PCA_ucsc smaple_R1_mm10.

bwt2merged.bam,sample_R2_mm10.bwt2merged.bam -tbp 1 -genome mm10 -checkGC -restrictionSite GATC. (2) We used (runHiCpca.pl sample_mer25_50 sample_PCA_ucsc -res 25,000 -superRes 50000 -genome mm10 -cpu 16) to find the Principal component analysis (PCA) of the data. (3) We used analyzeHiC as followed (analyzeHiC sample_PCA_ucsc -res 1000000 -interactions sample_significantInteractions.txt -nomatrix). Only Interaction looping between the H3K4me3 peaks and the TSS (+/-2 kb) from miR-9 candidate genes were considered. The settings were (annotateInteractions.pl Sample_significan mm10 Sample_output_filterk4m9 -filter H3K4me3_HICHIP-seq_peaks -filter2 TSS2kb_mouse transcripts -cpu 16 -washu -pvalue 0.01). (4) The mapped merged BAM files ouput from HiC-pro was processed as ChIP-seq to perform Peak calling, using (peak_call -i bam_file -o output_peak -r MboI_mm10.txt -f 0.01 -a mm10_chr_size -w 8) from (https://github.com/ChenfuShi/HiChIP_peaks).

HiChIP interaction hubs were generated by analyzeHiC and the interactions were annotated to the mm10 genome using annotateInteractions.pl as mentioned above. The hubs with frequency of interactions were filtered for a frequency greater than or equal to 10, and these frequencies were normalized to the total mean of the hubs. The output file generated from this was used to quantify and plot the number of hubs.

HiChIP interactions were annotated to the miR-9 ChIRP peaks using analyzeHiC. The interactions annotated to these loci were normalized by sampling down the number of interactions per condition and crossed with the H3K4me3 peaks specific for each condition (control, TGFB1 and TGFB1 + miR-9-LOF) using intersectBed from bedtools. The specific interactions were represented as loops visualized in IGV.

## Meta-analysis of Next-generation Sequencing data (ChIP-seq, PRO-seq, GRO-seq, ATAC-seq, CUT&Tag)
All published data from ChIP-seq, PRO-seq, GRO-seq and ATAC-seq used in this manuscript were listed in the Supplementary Table 1. All these NGS data were downloaded and processed according to the description in their respective publications. Briefly, PRO-seq, GRO-seq, ChIP-seq and CUT&Tag datasets were processed according to (https://doi.org/10.1038/s41588-018-0139-3). Trimmed high quality reads from CUT&Tag of H3K4me3, H3K27ac and G4 were mapped bowtie2 settings (--local --very-sensitive --no-mixed --no-discordant --phred33 -I 10 -X 700). After mapping, SAM files were converted to BAM format by the help of samtools view -Sb. Further, PCR duplicates were removed using MarkDuplicates.jar from Picard-tools (v.1.119) and mitochondrial Chr was removed from BAM mapped files using (awk '{if($3!= "chrM" && $3!= "chrUn"){print $0}}'). Peaks calling was performed for H3K4me3 marker MACS3 callpeak, with settings (--broad -g mmu -q 0.001 --keep-dup 1 --fix-bimodal --nomodel --extsize 1000). Peaks for H3K27ac were called with MACS3 callpeak, settings (--broad -g mmu -q 0.01 --keep-dup 1 --fix-bimodal --nomodel --extsize 200) and G4 was called with MCAS14 setting (p1e-3 for the p-value). The ideal settings for the peak calling parameters were selected after a custom comparison of the peak caller results with the visual representation in the IGV. After peak calling, high signal areas from the mm10 blacklist were removed using bedtools intersect (v2.30.0), settings (-wa -v). Where -a is the peak file and -b is the edited blacklist from mm10 genome from (https://github.com/Boyle-Lab/Blacklist/blob/master/lists/mm10-blacklist.v2.bed.gz).

The enrichment profiles of the factors and nascent RNAs (PRO-seq and GRO-seq) on the TSS of the Down-regulated genes after miR-9-LOF were performed using R package profileplyr (https://www.bioconductor.org/packages/devel/bioc/vignettes/profileplyr.html) adapting to a custom R-script (https://github.com/jcorderJC12/001nuMir9_G4_3D) or using HOMER: annotatePeaks.pl <peak/BED file> <genome> -d <tag directories> -hist <bin size> output file.txt. The line plots were produced using R or Microsoft Excel.

To calculate the broadness of H3K4me3 peaks, we first performed summary statistics of the peak size from the merged peak list. If the size of a peak was equal to or higher than the top 75% of peaks quantile 3 (Q3) was consider wide (≥2.7 kb). If the peak size was between Q3 and Q2 it was considered as medium size peak (≥2 kb and <2.7 kb). The narrow H3K4me3 peaks were in the bottom 25% peaks, Q1 or less (<2 kb).

### G4 Hunter analysis of G4 CUT&Tag data

The G4Hunter coordinates were generated for the mm10 genome using default settings as described previously[42]. The BAM files of G4 CUT&Tag was filtered with the G4Hunter coordinates of mm10 genome using bedtools with options: bedtools intersect -abam <BAM file > -b <G4Hunter BED> output.bam. After generating the G4Hunter filtered BAM files, HOMER was using with options -makeTagDirectory and the enrichment was quantified using -annotatePeaks.pl <peak/BED file > <genome > -d <tag directories > -hist <bin size> output file.txt.

### MicroRNA pulldown

For microRNA pulldown (miR-Pd), MLg cells were transfected with Biotinylated *mmu-miR-9-5p*-RNA (Exiqon) or Biotinylated *mirctrl* (Exiqon), at a final concentration of 20 nM. After 48 h, cells were fixed with 1% glutaraldehyde (Sigma-Aldrich) to preserve RNA-Chromatin interactions. After washing three times with PBS, cell pellets were resuspended in 2 ml hypotonic cell lysis buffer (10 mM Tris-HCl (pH 7.4), 1.5 mM MgCl2, 10 mM KCl, 1 mM DTT, 25 mM NaF, 0.5 mM Na3VO4, 40 µg/ml phenylmethylsulfonyl fluoride, protease inhibitor (Calbiochem) and RNase inhibitor (Promega) in DEPC treated water) on ice for 10 min and then spun down at 700 g for 10 min at 4 °C. Nuclear pellets were resuspended in 300 µL of nuclear lysis buffer (50 mM Tris-HCl (pH 7.4), 170 mM NaCl, 20% glycerol, 15 mM EDTA, 0.1% (v/v) Triton X-100, 0.2 mM DTT, 20 mM NaF, 20 mM Na3VO4, 40 µg/ml phenylmethylsulfonyl fluoride, protease inhibitor, RNase inhibitor in DEPC treated water). Nuclear fraction was sonicated (Bandelin Sonoplus GM70HD) at high amplitude for 10 cycles of 1 s (on/off) pulses. M280 streptavidin magnetic beads (ThermoFisher Scientific) were blocked for 1 h at 4 °C in blocking buffer (10 mM Tris-HCl pH 6.5, 1 mM EDTA, and 1 mg/ml BSA) and washed twice with 1 ml washing buffer (10 mM Tris-HCl (pH 7.0), 1 mM EDTA, 0.5 M NaCl, 0.1% (v/v) Triton X-100, RNase inhibitor and protease inhibitor). Beads were resuspended in 0.5 ml washing buffer. The nuclear extract was then added to the beads and incubated for 1 h at 4 °C with slow rotation. The beads were then washed five times with 1 ml washing buffer. RNA bound to the beads (pulled down RNA) and from 10% of the extract (input RNA) was isolated using Trizol reagent LS (Invitrogen) after DNase I (NEB) treatment.

For miRNA-protein interaction analysis, nuclear protein extract after DNase I treatment incubated with 500 pMol Biotinylated *miR-9* or Biotinylated *mirctrl* overnight at 4 °C. 30 µL of pre-blocked M280 streptavidin magnetic beads (ThermoFisher Scientific) were added and the nuclear protein extract was incubated 1 h in rotation at 4 °C. Protein bound to the beads (miR-Pd), were incubated with 30 µl 2x SDS samples loading buffer, boiled at 95 °C for 5 min, spun down and loaded on SDS-PAGE for western blot analysis and/or processed for proteomic analysis by mass spectrometry.

### Chromatin RNA immunoprecipitation

Chromatin RNA immunoprecipitation (Ch-RIP) analysis was performed as described[6] with minor adaptations. Briefly, cells were cross-linked by 1% formaldehyde for 10 min, lysed, and sonicated with Diagenode Bioruptor to disrupt and fragment genomic DNA. After centrifugation, the soluble chromatin was immunoprecipitated using antibodies. Precipitated chromatin complexes were removed from the beads by incubating with 50 µl of 1% SDS with 0.1 M NaHCO3 for 15 min, vortexing every 5 min and followed by treatment with DNase I. The collected material was immunoprecipitated using IgG or an HA-specific antibody (Santa Cruz, Cat No. 805). Reverse cross-linked immunoprecipitated materials were used for RNA isolation by Trizol (Invitrogen). Isolated RNA was subjected to cDNA synthesis and further for TaqMan assay specific for mature *miR-9* (Applied Biosystems, Cat. No. 4427975).

### Western blot

Nuclear protein extracts from MLg cells were prepared by cell lysis and nuclei isolation. Briefly, MLg cells were spooled down and washed with PBS. cell pellets were resuspended in 2 ml hypotonic cell lysis buffer (10 mM Tris-HCl (pH 7.4), 1.5 mM MgCl2, 10 mM KCl, 1 mM DTT, 25 mM NaF, 0.5 mM Na3VO4, 40 µg/ml phenylmethylsulfonyl fluoride, protease inhibitor (Calbiochem) and RNase inhibitor (Promega) in DEPC treated water) on ice for 10 min and then spun down at 700 g for 10 min at 4 °C. Nuclear pellets were resuspended in 300 µL of nuclear lysis buffer (50 mM Tris-HCl (pH 7.4), 170 mM NaCl, 20% glycerol, 15 mM EDTA, 0.1% (v/v) Triton X-100, 0.2 mM DTT, 20 mM NaF, 20 mM Na3VO4, 40 µg/ml phenylmethylsulfonyl fluoride, protease inhibitor, RNase inhibitor in DEPC treated water). Nuclear fraction was sonicated (Bioruptor NextGen, Diagenode) at high amplitude for 10 cycles of 30 sec (on/off) pulses. Detergent-insoluble material was precipitated by centrifugation at 20,800 g for 30 min at 4 °C. The supernatant was transferred to a fresh tube and stored at −20 °C. Protein concentration was estimated using Bradford assay, using serum albumin as standard. 5 µl of serial dilutions of standard protein and samples were mixed with 250 µl of Bradford reagent (500-0205, BIO-RAD Quick Start™).

Western blotting was performed using standard methods and antibodies specific for SMARCA5 (Invitrogen, MA5-35378), LMNB1 (Santa Cruz, sc-374015), LMNA (Santa Cruz, sc-20681), CHD4 (Abcam, 70469), RAD21 (Abcam, ab992), GAPDH (Sigma, MFCD01322099). Immunoreactive proteins were visualized with the corresponding HRP-conjugated secondary antibodies (Santa Cruz) using the Super Signal West Femto detection solutions (ThermoFisher Scientific). Signals were detected and analyzed with Luminescent Image Analyzer (Las 4000, Fujifilm).

### RNA sequencing and data analysis

RNA sequencing data for this paper were generated as previously described[6,37]. Briefly, total RNA from MLg cells that were transfected with control or *miR-9*-specific antagomiR was isolated using the Trizol method. RNA was treated with DNase I (RNase-Free DNase Set, Qiagen) and repurified using the RNeasy micro plus Kit (Qiagen). Total RNA and library integrity were verified on LabChip Gx Touch 24 (Perkin Elmer). Sequencing was performed on the NextSeq500 instrument (Illumina) using v2 chemistry with 1 × 75 bp single end setup. Raw reads were visualized by FastQC to determine the quality of the sequencing. Trimming was performed using trimmomatic with the following parameters: LEADING:3 TRAILING:3 SLIDINGWINDOW:4:15 MINLEN:15 CROP:60 HEADCROP:15. High quality reads were mapped to mm10 genome using with bowtie2. Tag libraries were obtained with Make-Taglibrary from HOMER (default setting). Samples were quantified by using analyzeRepeats.pl with the parameters: mm10 −count exons −strand both −noad. Gene expression was quantified in reads per kilo base million (RPKM) by the help of rpkm.default from EdgeR. Down-regulated genes after *miR-9*-LOF were those genes with a Log2fc (LOF/Ctr) <= −0.58 and Up-regulated those genes with a log2FC >= 0.58.

### Mass spectrometry-based proteomics

A total of 12 RNA interactomes from 2 different mouse cell lines using *miR-9*, as well as control miRNA probes in triplicate were captured on DynaBeads were washed and proteins were eluted by incubation in 50 µl of 4% sodium lauroyl sarcosinate (SLS) in 50 mM TEAB solution at 95 °C for 10 min. All DNA and RNA remnants were removed by addition of 0.5 µl of Turbonuclease and incubation for 15 min in room

temperature. Then, samples were reduced and alkylated through addition of DTT to a final concentration of 10 mM and incubation at 95 °C for 10 min, as well as iodoacetamide to a final concentration of 13 mM and incubation for 30 min at RT in darkness. A modified version of the SP3 method[71] as used for further sample preparation on an in-house made magnetic rack. Protein binding was performed in a final concentration of 70% anhydrous acetonitrile (ACN) solution at neutral pH with subsequent washes with 70% ethanol and 100% anhydrous acetonitrile. After acetonitrile removal, beads were resuspended in 50 μl of 50 mM TEAB buffer and 1 ug of trypsin (Promega, Madison, Wisconsin, USA) was added. Protein digestion was performed overnight, at 37 °C with shaking. Next, sample volumes were reduced to approximately 5 μl in a SpeedVac concentrator. Peptide binding to beads was initialized by addition of 100% ACN to its' final concentration above 95 %. Beads were washed twice using the same solvent. Peptides were eluted by addition of 40 μl of 0.1 % formic acid and transferred to MS-vials. Peptide concentration was estimated using the fluorimetric Pierce Quantitative Peptide Assays and sample volumes adjusted to achieve equal concentrations.

Purified peptides were analyzed by liquid chromatography–tandem mass spectrometry (MS) carried out on a Bruker Daltonics timsTOF Ultra instrument connected to a Bruker Daltonics nanoElute instrument. Approximately 50 ng of peptides were loaded onto a C18 precolumn (Thermo Trap Cartridge 5 mm, μ-Precolum TM Cartridge / PepMap TM C18, Thermo Scientific) and then eluted in the backflush mode with a gradient from 98% solvent A (0.15% formic acid) and 2% solvent B (99.85% acetonitrile and 0.15% formic acid) to 17% solvent B over 36 min, continued from 17 to 25% of solvent B for another 18, then from 25 to 35% of solvent B for another 6 min over a reverse-phase high-performance liquid chromatography (HPLC) separation column (PepSep Ultra, C18, 1.5 μm, 75 μm × 25 cm, Bruker Daltonics) with a flow rate of 300 nL/min. The outlet of the analytical column was coupled to the MS instrument by CaptiveSpray 20 μm Emitter. Data were acquired using a data-independent acquisition (DIA) paradigm using a default method provided by Bruker. In short, spectra were acquired with fixed resolution of 45,000 and mass range from 100 to 1700 m/z for the precursor ion spectra and a1/k0 range from 0.64 to 1.45 V s/cm2 with 100 ms ramp time for ion mobility, followed by DIA scans with 24 fixed DIA windows of 25 m/z width, ranging from 400 to 1000 m/z. Detailed method settings can be extracted from deposited.d measurement files.

Peptide spectrum matching and label-free quantitation were subsequently performed using DIA-NN[72] and a library-free search against the Mouse Uniprot.org database (17191 reviewed Swiss-Prot entries; January 2024). In brief, output was filtered to a 1% false discovery rate on precursor level. Deep learning was used to generate an in silico spectral library for library-free search. Fragment m/z was set to a minimum of 100 and a maximum of 1700. In silico peptide generation allowed for N-terminal methionine excision, tryptic cleavage following K*,R*, a maximum of one missed cleavage, as well as a peptide length requirement of seven amino acid minimum and a maximum of 30. Cysteine carbamidomethylation was included as a fixed modification and methionine oxidation (maximum of two) as a variable modification. Precursor masses from 100 to 1700 m/z and charge states one to four were considered. DIA-NN was instructed to optimize mass accuracy separately for each acquisition analysed and protein sample matrices were filtered using a run-specific protein q-value ("--matrix-spec-q" option). Downstream data processing and statistical analysis were carried out by the Autonomics package developed in-house (version 1.11.81). Proteins with a q-value of <0.01 were included for further analysis. MaxLFQ[73] values were used for quantitation and missing values imputed. DIA-NN spectral identification software initially identified 4904 protein groups. All intensities and maxLFQ values, that contain only 1 precursor per sample, were exchanged by NA for that particular sample. After dropping 421 without replication (within subgroup), and filtering out 1907 proteins with less than 2 peptides identified, 2576 protein groups were retained for further analysis. Differential abundance of protein groups was evaluated by Autonomics employing Bayesian moderated t-test as implemented by limma[74].

## Statistical analysis

Depending on the data, different tests were performed to determine the statistical significance of the results. The values of the statistical tests used in the different experiments can be found in Source Data file. Further details of statistical analysis in different experiments are included in the figures and figure legends. Briefly, one set of ChIRP, RNA, ChIP and HiChIP samples were analyzed by deep sequencing. For the rest of the experiments presented here, samples were analyzed at least in triplicates and experiments were performed three times. Statistical analysis was performed using Excel Solver and Prism9. Data in bar plots are represented as mean ± standard error (mean ± s.e.m.). Two-tailed t-tests were used to determine the levels of difference between the groups and $P$-values for significance. $P$-values after two-tailed t-test, $*P \leq 0.05$; $**P < 0.01$; and $***P < 0.001$.

## Reporting summary

Further information on research design is available in the Nature Portfolio Reporting Summary linked to this article.

# Data availability

The data supporting the findings of this study are available from the corresponding authors upon request. Source data are provided with this paper as a Source Data file. The sequencing data generated in this study have been deposited in NCBI's Gene Expression Omnibus database[75] under accession number GSE244952. Furthermore, we retrieved and used publicly available datasets to aid analysis of our data. Supplementary Data 1 contains all data sets used in this study. The model in Fig. 9d was Created with BioRender.com. Regarding the mass spectrometry-based proteomic, the full list of settings can be found in the "report.log.txt", full code for data processing and statistical analysis can be found in "stat-mir9-001.zip" uploaded along with the mass spectrometric raw data to the ProteomeXchange Consortium with dataset identifier: PXD054375, via the MassIVE partner repository (https://massive.ucsd.edu/, MassIVE-ID: MSV000095480; https://doi.org/10.25345/C5639KH31;). Source data are provided with this paper.

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

## Acknowledgements

We thank Roswitha Bender and Anne Robert for technical support; Virgine Marchand, Iouri Motorine and the EpiRNA-Seq Core Facility for support with NGS-based methods; Malgorzata Wygrecka, Kerstin Richter, Alessandro Ianni, Sylvain Maenner and Bruno Charpentier for reagents; Dominique Dumas for support with confocal microscopy; Amine Armich, Jean-Baptiste Vincourt, Sylvie Fournel-Gigleux, Mohamed Ouzzine, Sandrine Gulberti, Catherine Bui, Lydia Barré and Nick Ramalanjaona for comments. GB was funded by the "Centre National de la Recherche Scientifique" (CNRS, France), "Délégation Centre-Est" (CNRS-DR6) and the "Lorraine Université" (LU, France) through the initiative "Lorraine Université d'Excellence" (LUE) and the dispositive "Future Leader", the Max-Planck-Society (MPG, Munich, Germany) and the "Deutsche Forschungsgemeinschaft" (DFG, Bonn, Germany) (BA 4036/4-1). G.D. and J.C. are supported by the CRC 1366 (Projects A03, A06), the CRC 873 (Project A16), the CRC1550 (Project A03) funded by the DFG, the DZHK (81Z0500202) funded by BMBF, the Medical Faculty Mannheim of University of Heidelberg (90703207) and the Baden-Württemberg foundation special program "Angioformatics single cell platform". G.S. receives a doctoral fellowship through the initiative "Lorraine Université d'Excellence" (LUE). DGRA receives a doctoral fellowship from the DAAD (57552340). K.R. was funded by the "Consejo de Ciencia y Tecnología del Estado de Puebla" (CONCYTEP, Puebla, Mexico) through the initiative International Laboratory EPIGEN. T.B. is supported by the Deutsche Forschungsgemeinschaft, Excellence Cluster Cardio-Pulmonary Institute (CPI), Transregional Collaborative Research Center TRR81, TP A02, SFB1213 TP B02, TRR 267 TP A05 and the German Center for Cardiovascular Research. We acknowledge financial support by Heidelberg University for covering article publication charges (APCs).

## Author contributions

J.C., G.S., D.G.R.A., K.R,. A.E., S.M., W.S., J.G., S.G., and G.B. designed and performed the experiments; T.B. and G.D. were involved in study design; G.B. and J.C. designed the study; J.C., G.B., G.S., D.G.R.A., K.R., S.G., W.S., and J.G. analyzed the data; G.B., J.C., G.S., and D.G.R.A. wrote the manuscript. All authors discussed the results and commented on the manuscript.

## Funding

## Competing interests

The authors declare no competing interests.
