## [Transparent Peer Review file · Nature Communications]

Nuclear microRNA 9 is required for G-quadruplexes and 3D genome organization during TGF β -induced transcription

Corresponding Author: Dr Guillermo Barreto

Version 0:

Reviewer comments:

Reviewer #1

(Remarks to the Author)

The study by Cordero et al. investigated the role of miR-9 in transcription regulation. Following the finding that miR-9 is preferentially localized in the nucleus, the authors mapped the genomic binding of miR-9 by ChIRP-seq and assessed the impact of miR-9 depletion on gene expression and chromatin structure. They found that miR-9 binds to a subset of regulatory regions associated with expressed genes and its binding is required for H3K4me3 broad domains, high basal transcriptional activity, and, to a lesser extent G-quadruplex formation at promoters. The authors further investigated the role of miR-9 on distal interaction between promoters and super-enhancers (SE) upon TGF β stimulation. Interestingly, they found a striking interplay between miR-9 binding, 3D interactions and G4 formation.

Overall, the study is well performed and combines state-of-the-art approaches to unravel the nuclear function of miR-9. However, as detailed below, some conclusions are overstated and will need either further experimental validation or changes in the text.

Major remarks

1. Figure 1b shows the preferential localization of miR-9 in the nucleus. This figure should be complemented by control transcripts preferentially localized in the nucleus or the cytoplasm.
2. How the authors assessed the specificity of the miR-9 ChIRP? If feasible, the a control ChIRP should be performed after depletion of miR-9 using the specific antagomiR.
3. The association of miR-9 with Broad H3K4me3 domains is very interesting but needs further investigation. 1) Is miR-9 preferentially targeting (binding) Broad-associated genes? 2) To better appreciate the impact of miR-9 on Broad H3K4me3 domains, it will be interesting to compare the size distribution of H3K4me3 peaks in wild-type and miR-9 depleted conditions. In addition, the comparison of H3K4me3 signals (e.g., average profiles) for the three H3K4me3 categories (broad, medium and narrow) will be useful. 3) Finally, the impact of miR-9 on TGF β -dependent broad domains should be investigated.
4. To claim that miR-9 is required for G4 formation will require a genome-wide assessment of G4 formation in wild-type and miR-9 depleted cells.
5. Figure 4d: to claim that miR-9 preferentially associates with SE requires a statistical assessment of the overlap with SE and typical enhancers (TE), taking into account the relative genomic size of these elements.
6. Figures 4e and 4f compare the average profiles between SE bound by miR-9 and TE not bound by miR-9. Therefore, it is not possible to assess whether the observed differences are due to the binding of miR-9 or the type of regulatory element. A fairer comparison should show the average profiles of the same type of regulatory element (SE or TE) with or without miR-9 binding.
7. Figures 3 and 5: To demonstrate the specificity of miR-9 impact on G4 formation, a control G4-region not affected by miR-9 depletion should be shown.

Minor remarks

8. Some rationale should be provided to justify the study of miR-9 in lung cells. Is miR-9 expressed at a high level in lung tissues or deregulated in lung cancers?
9. The subtitle "Mature miR-9 is detected in the cell nucleus enriched at promoters and introns" could be replaced by "Mature miR-9 is detected in the cell nucleus and is enriched at promoters and introns"
10. Figure 2b: the legend "non target" is not useful here as this category is not presented in this panel.
11. Figure 2c: the blue color is misleading as the same color is used in panel 2b to represent downregulated genes only.
12. Figure 2h: while motifs 1 and 3 show some level of similarities between mir-9 targets and G4 peaks, motifs 2 are not similar. This should be better explained in the text.
13. The authors claim to use HiChIP-seq with H3K4me3-specific antibodies to precipitate active SE (lines 291-293). However, the use of H3K4me3 will primarily precipitate promoters. Maybe the authors could modify the sentence to indicate that HiChIP-seq will allow to precipitate SEs interacting with active promoters.
14. Figure 6d: the number of interactions should be shown for each cluster.
15. Line 296: clusters 1 and 3 should be replaced by 1 and 4.
16. Typo line 575: H3K4me3.
17. Line 348: "a so called a G-loop structure is generated" should be "a so called G-loop structure is generated"

Reviewer #2

(Remarks to the Author)

The manuscript delineates a new regulatory axis mediated by the nuclear-localized microRNA, miR-9, and G-quadruplex formation that facilitates three-dimensional chromatin looping between the promoter regions and super-enhancers (SEs) of genes pertinent to TGF- β 1 signaling. The insights presented by the authors could potentially enrich our understanding of microRNA-mediated regulatory mechanisms in the nucleus. Nonetheless, the conclusions drawn require more thorough validation prior to consideration for publication in Nature Communications.

Major Comments:

1. The investigation is limited to a single fibroblast cell type, within which the enrichment of miR-9 targets associated with TGF- β 1 response and cell proliferation has been identified. To thoroughly comprehend the nuclear functions of miR-9, it is imperative to extend the analysis of miR-9 targets and G-quadruplex formations across different cell types. The specificity of miR-9/G-quadruplex target interactions raises several questions: Is miR-9 ubiquitously associated with all transcriptionally active genes across cell types? Does it regulate a distinct subset of genes involved in specific cellular pathways? Or does it selectively modulate genes requiring high expression levels?
2. The association between miR-9 and super enhancers, as depicted in Figure 4a, is not immediately evident. The markers MED1 and H3K27ac cannot be solely relied upon to define SEs. A more robust algorithmic approach, such as the ROSE program that identifies clustered enhancers, should be employed. Given that SEs typically comprise several hundred clusters in a given cell, the analysis of miR-9 enrichment within SEs necessitates further scrutiny. Figures 5 and Supplementary Figure 5 offer detailed views of individual enhancers as exemplars of SEs; however, an encompassing representation of entire SE regions is warranted.
3. LOF experiments with miR-9 have primarily been conducted to elucidate its role in genome organization and transcriptional activation within the nucleus. Nevertheless, the possibility that the observed effects may be attributable to alternative pathways involving miR-9 cannot be discounted. Consequently, additional evidence is required to affirm the specific involvement of miR-9 in G-quadruplex formation and chromatin looping.
4. Figure 1b illustrates a variable distribution of miR-9 between nuclear and cytoplasmic compartments. It would be informative to test whether miR-9 also contributes to transcription and genome organization in cell types where its nuclear presence is minimal, such as human lung fibroblasts (hLF).

Minor Comments:

1. In Figure 1, ChIRP-seq should be complemented with a negative control for validation purposes.
2. Figure 2a should include examples of H3K4me3 ChIP-seq track data for assessment purposes. The confocal microscopy

data presented in Supplementary Figure 2b does not substantiate the related claims; notably, all H3K4me3 signals seem to diminish following miR-9 LOF. The examples depicted in Figure 3a are not sufficiently persuasive.

3. Contrary to assertions made in the text, Figure 2c shows that miR-9 targets in control samples exhibit a lower median expression level compared to non-targets, challenging the notion that miR-9 elevates basal transcriptional activity in target genes.
4. Figure 2d-i should include comparative heatmap data for non-miR-9 target genes that are expressed, for comparison.
5. Figures 3a and Supplementary Figure 3 fail to convincingly demonstrate colocalization between miR-9 ChIRP and G4 ChIP signals, raising questions about their functional interplay. It remains unclear whether miR-9 is essential for G-quadruplex formation and, if so, by what mechanism.
6. Analyses across the genome scale (e.g., Figures 3b, 6c, 8) should be presented rather than relying on select examples to substantiate claims.
7. On page 12, there appears to be a typographical error where 'cluster 1 and 3' should be corrected to 'cluster 1 and 4'.
8. The claim that miR-9 is necessary for chromatin alterations induced by TGF- β 1, as suggested by Figure 6c, requires genome-wide evidence. Additionally, HiChIP-seq analysis should be integrated with comprehensive miR-9 ChIRP-seq data to conclusively demonstrate miR-9's direct role in chromatin looping. It is also pertinent to ascertain whether TGF- β 1 induces the recruitment of miR-9 to its targets.

Reviewer #3

(Remarks to the Author)

This study explores a novel function of miRNA in modulating genome organization within the nucleus. The authors report that miR-9, traditionally associated with cytoplasmic functions, exhibits nuclear localization in specific cells, potentially influencing genome interactions. Utilizing miR-9-based ChIRP-seq, the authors propose an interaction between miR-9 and a subset of super enhancers and promoters containing G-quadruplex motifs. Through various high-throughput techniques such as RNA-seq and ChIP-seq, the authors conclude that miR-9 depletion alters chromatin features at these loci, resulting in decreased gene expression. Additionally, they demonstrate the role of miR-9 in TGFB1-mediated promoter-enhancer looping.

The premise of the study is intriguing, particularly in light of emerging evidence suggesting nuclear localization of Argonaute proteins under certain physiological conditions. If validated, the proposed model would offer valuable insights into the role of nuclear miRNAs, thereby significantly contributing to the field.

However, despite the extensive data provided, I find the evidence to be preliminary and the proposed model unconvincing. Firstly, the manuscript lacks a mechanistic understanding of key aspects. Questions regarding the nuclear translocation of miR-9, the potential involvement of Ago proteins in its nuclear function, the specificity of miR-9 in targeting G-quadruplex motifs, and its selective role in TGFB1 regulation remain unanswered. Without elucidating these mechanisms, establishing such a controversial model requires robust evidence, which is currently lacking. In fact, most conclusions are based on associations rather than causal relationships, with limited support from approaches such as mutagenesis studies. Furthermore, the presentation of analyses is unclear and fails to convincingly support many of the conclusions drawn. For example, how are targets of nuclear miR-9 defined/selected? Lastly, the method used for miR-9 LOF through antagomir treatment raises concerns about potential off-target effects, which may confound the interpretation of results. This study can clearly benefit from including additional controls.

Given these shortcomings, I cannot recommend the publication of this manuscript in its current form.

Specific points:

1. Figure 1a: The observation that miR-9 shares the same nuclear localization signal as miR-126 raises questions about the potential involvement of Mex3a, known to mediate miR-126 nuclear transportation, in the nuclear localization of miR-9. It is important to test whether Mex3a is also responsible for shuttling miR-9 into the nucleus. Experimentally manipulating Mex3a to modulate miR-9 shuttling and subsequently validating the functional consequences of nuclear miR-9 would significantly strengthen the conclusions of this study.
2. Figure 1b. It is striking that miR-9 exclusively resides inside nucleus in MLg cells, which authors opted to focus on in the rest of the study. Nonetheless, I would strongly suggest that authors include cells such as hLF where nuclear miR-9 is limited in the following functional characterization. Incorporating hLF cells as a negative control would serve to validate that the observations are indeed specific to miR-9.
3. Figure 1c: Authors should consider performing RNA FISH on hLF cells as well to confirm that the signal detected corresponds to mature miR-9.
4. Figure 1d: Only showing enrichment over background might be misleading. How can we be certain that it is specific to miR-9? What is the enrichment relative to the control (pulldown with control LNA oligos)?
5. Figure 1e: The definition of targets (genome regions) of nuclear miR-9 requires clarification. Specifically, the authors should elaborate on how the targets, indicated as "n=2637," were selected. It is essential to clarify whether these targets

were chosen based on relative enrichment over control and emphasize the significance of this data, as subsequent analyses in the study heavily rely on it.

6. The results presented in Figure 2 would benefit from the inclusion of additional controls. Specifically, authors should consider using cells where miR-9 is not expressed or absent in the nucleus to provide a comparative context for the observed effects.

7. Figure 2c: This is confusing. While the text (lines 188-191) suggests a comparison between miR-9 targets and non-targets defined by genome interaction (ChIRP, Figure 1), the figure legend indicates "n=2439" and "n=881," which represent the number of genes whose expression was or wasn't influenced by miR-9 antagomir treatment. The discrepancy should be addressed and clarified.

Version 1:

Reviewer comments:

Reviewer #1

(Remarks to the Author)

The study by Cordero et al. have significantly improved in the revision stage. In particular, all my critical issues have been satisfactorily addressed in the revised version. My only minor suggestion is to remove the mention to G-quadruplexes in the title, as this is not the main focus of the paper, while specifically mentioning miR-9.

Reviewer #2

(Remarks to the Author)

The authors have adequately addressed this reviewer's comments. Publication of the revised manuscript is recommended.

Reviewer #3

(Remarks to the Author)

I appreciate the substantial effort the authors have put into addressing the concerns raised in the initial review. The new data and controls have significantly improved the manuscript, making many of the previous conclusions more convincing.

However, I remain unconvinced that the proposed model is entirely correct. There are still aspects that do not fully make sense to me, and crucial mechanistic insights are still missing. These gaps are especially challenging given how controversial the model is. It is perhaps understandable that establishing such a model in a single study might be difficult, if not impossible.

Despite these lingering concerns, I believe that this study should be published with the understanding that time will ultimately determine whether this represents a breakthrough in understanding miRNA function in the nucleus or if it is an artifact. The field should have the opportunity to engage with these interesting findings and test their validity.

Point-by-point response to the Reviewers – Manuscript with the number NCOMMS-23-61140-T and the title “3D genome organization during TGFB-induced transcription requires nuclear microRNA and G-quadruplexes” submitted as original scientific article to Nature Communications by Cordero J et al.

REVIEWER COMMENTS

We would like to thank all the Reviewers for taking the time to review our manuscript and providing valuable comments on how to improve it. The amendments introduced in response to their feedback are highlighted in green in the revised manuscript file. The constructive comments from all three Reviewers helped us to improve our manuscript by leading us to new experiments providing new results that further strengthen our model and support our interpretation of the data. These new experiments are listed in the table below:

New Experiments	Figures
miR-9 - and Mirlet7d -specific ChIRP-seq in MLg and MLE-12 cells	Fig 1d-e Supp Fig 2a-f, 6b-c
H3K4me3-specific CUT&Tag in MLg and MLE-12 cells after Ctrl transfection or miR-9 -LOF	Fig 2a-e Supp Fig 3a-d
miR-9 -specific FISH in Ctrl hLF and IPF hLF	Supp Fig 1d
G4-specific CUT&Tag in MLg and MLE-12 cells after Ctrl transfection or miR-9 -LOF	Fig 3g, 4a, 5h-i, 6a, 8e, Supp Fig 4a-c, 5a, 7a
miR-9 - and Mirlet7f -specific TaqMan after G4-specific Ch-RIP in MLg cells	Fig 3h
high-resolution mass spectrometry based proteomic approach after miR-Pd using the nuclear fraction of MLg or MLE-12 cells and biotinylated control miRNA (mirctrl) or miR-9 as baits	Fig 3i-j Supp Fig 4d-f
H327ac-specific CUT&Tag in MLg and MLE-12 cells after Ctrl transfection or miR-9 -LOF	Fig 5e-h, 6a, Supp Fig 6a, 7a
miR-9 -specific FISH in MLg cells that were non-treated or treated with TGFB1, and Ctrl transfected or after miR-9 -LOF	Fig 7c-d
G4- and H3K4me3-specific immunostaining in MLg cells that were non-treated or treated with TGFB1, and Ctrl transfected or after miR-9 -LOF	Fig 7c-d
qPCR after miR-9 -specific ChIRP	Fig 7e
RNA-seq in MLg cells that were non-treated or treated with TGFB1, and Ctrl transfected or after miR-9 -LOF	Fig 9c

Reviewer #1 (Remarks to the Author):

The study by Cordero et al. investigated the role of miR-9 in transcription regulation. Following the finding that miR-9 is preferentially localized in the nucleus, the authors mapped the genomic binding of miR-9 by ChIRP-seq and assessed the impact of miR-9 depletion on gene expression and chromatin structure. They found that miR-9 binds to a subset of regulatory regions associated with expressed genes and its binding is required for H3K4me3 broad domains, high basal transcriptional activity, and, to a lesser extent G-quadruplex formation at promoters. The authors further investigated the role of miR-9 on distal interaction between promoters and super-enhancers (SE) upon TGF β stimulation. Interestingly, they found a striking interplay between miR-9 binding, 3D interactions and G4 formation.

Overall, the study is well performed and combines state-of-the-art approaches to unravel the nuclear function of miR-9. However, as detailed below, some conclusions are overstated and will need either further experimental validation or changes in the text.

We thank Reviewer 1 for the positive comment related to the novelty of our work and for the time and efforts implemented during the peer review of our manuscript.

Major remarks

1. Figure 1b shows the preferential localization of miR-9 in the nucleus. This figure should be complemented by control transcripts preferentially localized in the nucleus or the cytoplasm.

We appreciate this constructive comment from Reviewer 1, which aims to demonstrate the specificity of our *miR-9* FISH. Due to resource limitations, we were not able to purchase an RNA FISH probe that is specific to another miRNA. Our financial resources were invested in the experiments listed in the table above.

However, with the aim to demonstrate the specificity of the *miR-9* FISH, we followed the comment 4 from Reviewers 2 and comment 3 from Reviewer 3:

We performed a *miR-9*-specific FISH in primary human lung fibroblast (hLF) from control donors (Ctrl hLF) as cells with relatively low levels of *miR-9* in the cell nucleus (see Figure 1b). Confirming the results from the TaqMan assay-based, *miR-9*-specific expression analysis after cellular fractionation presented in Figure 1b, we observed by RNA FISH the majority of *miR-9* in the cytosol of Ctrl hLF (Supplementary Figure 1d, top).

In addition, we performed *miR-9*-specific FISH also in hLF isolated from the lung of patients with idiopathic pulmonary fibrosis (IPF hLF), a lethal interstitial lung disease involving TGFB1 signaling. To our surprise, we observed by RNA FISH that the majority of *miR-9* is in the cell nucleus of IPF hLF, suggesting a translocation mechanism of *miR-9* into the cell nucleus potentially related to IPF (Supplementary Figure 1d, middle). Further, *miR-9*-LOF in IPF hLF reduced the levels of *miR-9* (Supplementary Figure 1d, bottom).

The results of these new RNA FISH experiments are presented in a new figure panel (Supplementary Figure 1c) and described in the manuscript as follows:

Change in the manuscript:

(Lines 145-155)

“Further, the nuclear localization of *miR-9* was confirmed by RNA fluorescence in situ hybridization (FISH) in MLg cells (Figure 1c and Supplementary Figure 1c) and hLF from control donors (Ctrl hLF) or patients with idiopathic pulmonary fibrosis (IPF hLF; Supplementary Figure 1d), a lethal interstitial lung disease involving TGFB1 signaling³⁷. In MLg cells, we detected *miR-9* in specific regions of the nuclei, whereas the levels of *miR-9* were reduced after loss-of-function (LOF) experiments using unlabeled *miR-9*-specific antagomiR probes. In Ctrl hLF, the intensity of *miR-9* FISH was higher in the cytosol than in the nucleus, whereas in IPF hLF the majority of *miR-9* was detected in the cell nucleus, pointing

to a translocation mechanism of *miR-9* into the cell nucleus potentially related to IPF. Further, *miR-9*-LOF in IPF hLF reduced the levels of *miR-9*.”

Further, we performed RNA FISH in MLg cells that were non-treated or treated with TGFB1, and transfected with a control (Ctrl) or *miR-9*-specific antagomir to induce a loss-of-function (LOF). The results of these new RNA FISH experiments are presented in new figure panels (Figure 7c-d) and described in the manuscript as follows:

Change in the manuscript:

(Lines 356-360)

“Supporting these observations, RNA FISH and immunostaining in MLg cells showed that TGFB1 treatment increased the levels of *miR-9*, G4 and H3K4me3 in *miR-9*-dependent manner (Figure 7c-d). Similarly as in IPF hLF (Supplementary Figure 1d), the majority of *miR-9* was detected in the cell nucleus of MLg cells after TGFB1 treatment.”

The new results by RNA FISH in Ctrl hLF, IPF hLF and MLg cells confirm the specificity of our results.

2. How the authors assessed the specificity of the *miR-9* ChIRP? If feasible, the a control ChIRP should be performed after depletion of *miR-9* using the specific antagomiR.

We followed this suggestion from Reviewer 1 and performed a *miR-9* ChIRP-seq after *miR-9*-LOF. However, the library preparation was not possible, as expected, since the *miR-9*-LOF did not allow any enriched chromatin that could be sequenced.

In any case, we understand this comment from Reviewer 1 as a suggestion to demonstrate the specificity of the *miR-9* ChIRP-seq. With the aim of demonstrating the specificity of our *miR-9* ChIRP-seq, and following also the

comment 1 from Reviewer 2 and comment 4 from Reviewer 3, we implemented two strategies: (1) we performed *miR-9* ChIRP seq in another cell line (MLE-12 cells) and (2) we did a ChIRP-seq using a probe specific for another miRNA that has been characterized in the cell nucleus (*Mirlet7d*). The results of these strategies are presented in new figure panels (Figures 1d-e, and Supplementary Figures 2a-d) and described in the manuscript as follows:

Change in the manuscript:

(Lines 155-169)

“To investigate the role of *miR-9* in the cell nucleus we performed a sequencing experiment after chromatin isolation by miRNA purification (ChIRP-seq) using chromatin from MLg cells and control (Ctrl) or *miR-9*-specific biotinylated antisense oligonucleotides for the precipitation of endogenous mature *miR-9* along with the chromatin bound to it (Figures 1d-e and Supplementary Figures 2a-f). To demonstrate the specificity of our ChIRP-seq experiment, we also used a probe specific for another miRNA characterized in the cell nucleus (miRNA lethal 7d, *Mirlet7d*, also known as *let-7d*)^{8,53}, and chromatin from MLE-12 cells. We detected specific enrichment of *miR-9* at loci without *Mirlet7d* in MLg and MLE-12 cells (Figure 1d, left, and Supplementary Figure 2b, left). Further, genome-wide binding profile analysis of *miR-9* in MLg cells revealed an increase in the number of *miR-9* peaks at promoters, transcription termination sides (TTS) and intronic regions compared to the negative control (Figure 1d, right), whereas in MLE-12 cells *miR-9* was enriched at TTS and intronic regions (Supplementary Figure 2b, right). Interestingly, the loci with *miR-9* enrichment were different in MLg and MLE-12 cells (Supplementary Figure 2c), suggesting that *miR-9* regulates different genes in different cells.”

Our new results not only demonstrate the specificity of our *miR-9* ChIRP-seq, but also show that both nuclear miRNAs (*miR-9* and *Mirlet7d*) regulate different loci in the same cell line (Figure 1d and Supplementary Figure 2b), and that *miR-9* regulates different loci in different cell lines (MLg and MLE-12 cells; Supplementary Figure 2c).

3. The association of miR-9 with Broad H3K4me3 domains is very interesting but needs further investigation. 1) Is miR-9 preferentially targeting (binding) Broad-associated genes? 2) To better appreciate the impact of miR-9 on Broad H3K4me3 domains, it will be interesting to compare the size distribution of H3K4me3 peaks in wild-type and miR-9 depleted conditions. In addition, the comparison of H3K4me3 signals (e.g., average profiles) for the three H3K4me3 categories (broad, medium and narrow) will be useful. 3) Finally, the impact of miR-9 on TGFb-dependent broad domains should be investigated.

We appreciate the constructive comment by Reviewer 1, which has also contributed to the improvement of our manuscript. To fully address the suggestions from Reviewer 1, we performed a new sequencing experiment following Cleavage Under Targets and Tagmentation (CUT&Tag) for genome-wide profiling of H3K4me3 in MLg cells that were transiently transfected with Ctrl or *miR9*-specific antagomiR to induce a *miR-9*-LOF. In parallel, we performed the same experiment in MLE-12 cells to investigate differences between cell lines. The results of these experiments are presented in new figure panels (Figures 2a-d and Supplementary Figures 3a-d) and described in the manuscript as follows:

Change in the manuscript:

(Lines 184-209)

“To further investigate a potential role of nuclear *miR-9* in transcription regulation, we performed a sequencing experiment following Cleavage Under Targets and Tagmentation (CUT&Tag) for high-resolution, genome-wide profiling of tri-methylated lysine 4 of histone 3 (H3K4me3) in MLg and MLE-12 cells that were transiently transfected with Ctrl or *miR9*-specific antagomiR to induce a *miR-9*-LOF (Figure 2a-e, Supplementary Figure 3a-d). Peak distribution analysis of the H3K4me3 CUT&Tag showed that 60.8% ($P = 0.01$) of the H3K4me3 broad domains in Ctrl transfected MLg cells were enriched with *miR-9* (Figure 2a), whereas 63% ($P < 0.01$) of the H3K4me3 broad domains were enriched with *miR-9* in Ctrl transfected MLE-12 cells (Supplementary Figure 3b). Interestingly, H3K4me3 levels at broad

domains were significantly reduced in MLg cells from a median of 1.6 RPKM (IQR = 3.3) in Ctrl transfected cells to a median of 0.8 RPKM (IQR = 1.8; $P = 0.002$) following *miR-9*-LOF (Figure 2b, top), whereas the effects of *miR-9*-LOF in MLE-12 cells were not significant (Figure 2b, bottom). In addition, we observed that the loci of the H3K4me3 broad domains with *miR-9* enrichment were different in MLg and MLE-12 cells (Supplementary Figure 3c-d), confirming that *miR-9* regulates different genes in these two cell lines. Due to these results and the higher levels of nuclear *miR-9* (Figure 1b), we focused on MLg cells. Further peak distribution analysis showed that H3K4me3 broad domains were reduced from 27.6% in Ctrl transfected MLg cells to 22.1% after *miR-9*-LOF, whereas medium and narrow H3K4me3 domains increased (Figure 2c). Interestingly, the shift from H3K4me3 broad domains to medium and narrow domains following *miR-9*-LOF was significant at promoters but not at gene body and intergenic regions (Figure 2d). However, enrichment plots showed that H3K4me3 levels were reduced following *miR-9*-LOF in H3K4me3 broad and medium domains at promoter, gene body and intergenic regions (Figure 2e). The reduction of H3K4me3 levels after *miR-9*-LOF was confirmed by confocal microscopy after H3K4me3-specific immunostaining in Ctrl- and *miR-9*-antagomiR transfected MLg cells (Supplementary Figure 3e).”

These new results address the first two questions raised from Reviewer 1 in this specific comment. To answer the third question, we accordingly analyzed our H3K4me3 ChIP-seq experiment in MLg cells that were non-treated or treated with TGFB1 and Ctrl transfected or transfected with *miR-9*-specific antagomir to induce *miR-9*-LOF. The results of this new analysis of the H3K4me3 ChIP-seq experiment are presented in new figure panels (Figures 7f-g and Supplementary Figures 8a-b) and described in the manuscript as follows:

Change in the manuscript:

(Lines 364-374)

“These results were complemented by H3K4me3 ChIP-seq in MLg cells that were transiently transfected with Ctrl or *miR9*-specific antagomiR probes, and non-treated or treated with TGFB1 (Figures 7f-h and Supplementary Figure 8a-c). H3K4me3 levels significantly increased after TGFB1 treatment at promoters of TGFB-responsive genes in *miR-9*-dependent manner (Figure 7f, left). These effects were not observed at the promoters of genes that did not respond to TGFB1 treatment (Figure 7f, right) supporting the specificity of the effects observed. By checking on H3K4me3 levels at loci, in which we detected G4s by CUT&Tag, we observed that TGFB1 did not significantly affect H3K4me3 levels, whereas the combination of TGFB1 and *miR-9*-LOF reduced H3K4me3 levels at these loci (Supplementary Figure 8b). Interestingly, the breadth of H3K4me3 peaks increased after TGFB1 treatment also in *miR-9*-dependent manner (Figure 7g).”

4. To claim that miR-9 is required for G4 formation will require a genome-wide assessment of G4 formation in wild-type and miR-9 depleted cells.

We completely agree with this comment from Reviewer 1. Thus, we performed a G4-specific CUT&Tag experiment in MLg cells that were transiently transfected with Ctrl or *miR9*-specific antagomiR to induce a *miR-9*-LOF. In addition, we performed the same experiment in MLE-12 cells to investigate differences between cell lines. The results of these experiments are presented in new figure panels (Figures 3g, 4a, 5a, Supplementary Figures 4a-c and Supplementary Figure 5a) and described in the manuscript as follows:

Change in the manuscript:

(Lines 253-263)

“To demonstrate that *miR-9* is required for G4 formation, we analyzed by CUT&Tag using G4-specific antibodies chromatin from MLg and MLE-12 cells transiently transfected with Ctrl or *miR9*-specific antagomiR (Figure 3g and Supplementary Figure 4a-c). Analysis of the G4 CUT&Tag data without or with filtering based on G4Hunter scores⁴¹ showed genome-wide

reduction of G4s after *miR-9*-LOF in MLg and MLE-12 cells (Figure 3g and Supplementary Figure 4b, both left), thereby demonstrating the requirement of *miR-9* for G4 formation in both cell lines. However, correlating with the levels of nuclear *miR-9* in both cell lines (Figure 1b), the reducing effect of *miR-9*-LOF on G4 levels was more pronounced in MLg cells at TSS of *miR-9* target genes (Figure 3g and Supplementary Figure 4b, both right). Interestingly, the majority of the loci with G4s were different in MLg and MLE-12 cells (Supplementary Figure 4c), supporting that G4s are involved in the regulation of different genes in both cell lines.”

To further confirm the role of miR-9 in G4 formation, we performed additional experiments:

On one hand we did TaqMan-based miRNA enrichment analysis following chromatin-RNA immunoprecipitation (Ch-RIP) using G4 specific antibodies (Figure 3h) showing that G4s significantly bound mature *miR-9* and *miR-9*-LOF abolished this interaction.

On the other hand, we performed a high-resolution mass spectrometry based proteomic approach after miRNA pulldown (miR-Pd) using the nuclear fraction of MLg or MLE-12 cells and biotinylated control miRNA (*mirctrl*) or *miR-9* as baits (Figure 3i, Supplementary Figures 4c-e and Source Data file 01) that demonstrates the interaction of mature miR-9 with proteins that have been reported to interact with G4s [PMID 34188089] in both cell lines. Interestingly, *miR-9* did not significantly bind AGO1, AGO2, MEX3D and MEX3A. However, gene Set Enrichment Analysis (GSEA) of the *miR-9* binding proteins (Figure 3j) showed that nuclear miR-9 interacted with proteins involved in G4s, Chromatin, RNA processing, Gene expression and Chromatin organization. All these results support an important role of miR-9 in G4 formation and transcription regulation.

These new figure panels are described in the manuscript as follows:

Change in the manuscript:

(Lines 263-282)

“To verify the interaction between mature *miR-9* and G4s, we performed TaqMan-based miRNA enrichment analysis following chromatin-RNA immunoprecipitation (Ch-RIP) using

G4 specific antibodies (Figure 3h). G4s significantly bound mature *miR-9* and *miR-9-LOF* abolished this interaction, whereas G4s did not bind mature *Mirlet7f*, a miRNA used as negative control, thereby showing the specificity of the interaction between mature *miR-9* and G4s. To further investigate this interaction and identify protein-binding partners of *miR-9* in the nucleus, we performed a high-resolution mass spectrometry based proteomic approach after miRNA pull-down (miR-Pd) using the nuclear fraction of MLg or MLE-12 cells and biotinylated control miRNA (*mirctrl*) or *miR-9* as baits (Figure 3i, Supplementary Figures 4d-f and Source Data file 01). Results from three independent experiments identified 169 proteins in the nuclear fraction of MLg cells and 233 proteins in the nuclear fraction of MLE-12 cells that were significantly enriched after *miR-9*-Pd. From the *miR-9* binding proteins, 20 proteins in the nuclear fraction of MLg cells and 58 proteins in the nuclear fraction of MLE-12 cells have been reported to interact with G4s⁴². Interestingly, *miR-9* did not significantly bind AGO1, AGO2 and MEX3D. However, Gene Set Enrichment Analysis (GSEA)⁴³ of the *miR-9* binding proteins (Figure 3j) showed that nuclear *miR-9* interacted with proteins involved in G4s ($P = 1.23E-4$), Chromatin ($P = 1.5E-3$), RNA processing ($P = 1.65E-49$), Gene expression ($P = 1.56E-54$) and Chromatin organization ($P = 9.3E-3$). All these results support an important role of *miR-9* in G4 formation and transcription regulation.”

5. Figure 4d: to claim that miR-9 preferentially associates with SE requires a statistical assessment of the overlap with SE and typical enhancers (TE), taking into account the relative genomic size of these elements.

To address this constructive comment from Reviewer 1, we performed a new H3K27ac CUT&Tag experiment in Ctrl or miR-9 antagomiR-transfected MLg cells (Supplementary Figure 6a) and analyzed the data using the rank-ordering of super-enhancers (ROSE) algorithm [PMID 23582322] to separate super-enhancer (SE) from typical enhancer (TYE). The results of this new experiment are presented in new figure panels (Figure 5e and Supplementary Figure 6a) and described in the manuscript as follows:

Change in the manuscript:

(Lines 304-310)

“To investigate a potential role of *miR-9* in enhancers, we performed H3K27ac CUT&Tag in Ctrl or *miR-9* antagomiR-transfected MLg cells (Supplementary Figure 6a) and analyzed the data using the rank-ordering of super-enhancers (ROSE) algorithm ¹¹ to separate SE from typical enhancer (TYE) (Figure 5d). We detected 1,649 SE in Ctrl transfected MLg cells that were significantly reduced to 1,084 ($P = 9.9E-142$) after *miR-9*-LOF supporting the requirement of *miR-9* for SE formation.”

In addition, to make our manuscript more interesting and increase the impact of our findings, we followed a suggestion from a colleague after an oral presentation of this project during a chromatin-related congress in France. To link the reducing effect of *miR-9*-LOF on G4 formation to SE or TYE, we decided to analyze our new G4-specific CUT&Tag experiment in MLg cells that were transiently transfected with Ctrl or *miR9*-specific antagomiR to induce a *miR-9*-LOF (Figures 3g, 4a, 5a, Supplementary Figures 4a-c and Supplementary Figure 5a) using the ROSE algorithm [PMID 23582322]. The results from this analysis are presented in new figure panels (Figure 5h-i) and described in the manuscript as follows:

Change in the manuscript:

(Lines 327-332)

“In addition, the levels of H3K27ac and G4 were higher at SE than at TYE (Figure 5h). However, *miR-9*-LOF reduced the levels of H3K27ac and G4 at both enhancer types. Remarkably, analysis of the G4 CUT&Tag using the ROSE algorithm showed that the number of SE containing G4s (1,753) was significantly reduced (937, $P = 2.5E-35$) after *miR-9*-LOF (Figure 5i) suggesting a role of G4s in SE formation in a *miR-9*-dependent manner.”

6. Figures 4e and 4f compare the average profiles between SE bound by miR-9 and TE not bound by miR-9. Therefore, it is not possible to assess whether the

observed differences are due to the binding of miR-9 or the type of regulatory element. A fairer comparison should show the average profiles of the same type of regulatory element (SE or TE) with or without miR-9 binding.

We thank Reviewer 1 for this relevant comment. We followed the suggestion from Reviewer 1 and placed the enrichment plots comparing the enrichment of *miR-9* and enhancer-related proteins at SE and TYE in the Supplementary Figures 6b-c. In addition, we generated a heat map summarizing the results from these enrichment plots and the statistical significance. We placed this heat map into the manuscript in the Figure 5f showing that the enrichment of all these proteins was significantly higher at SE that also contained *miR-9* as compared to SE without *miR-9* or TYE with/without *miR-9*. These figure panels are described in the manuscript as follows:

Change in the manuscript:

(Lines 313-320)

“These results were confirmed by further analysis of the *miR-9* ChIRP-seq data together with publicly available ChIP-seq data of proteins that have been related to enhancers. Aggregate plots (Supplementary Figure 6b-c) and a heat map (Figure 5f) showed significant enrichment of *miR-9* at SE together with H3K27ac, MED1, KLF4, HDAC1, SMAD3, SMARCA4, EP300, SMARCA1, RAD21, MYC, MEIS1, BRG1, HDAC2, and EST1. Interestingly, we also detected significant enrichment of *miR-9* and all the analyzed proteins at TYE. However, the enrichment of all these proteins was significantly higher at SE that also contained *miR-9* as compared to SE without *miR-9* or TYE with/without *miR-9*.”

7. Figures 3 and 5: To demonstrate the specificity of miR-9 impact on G4 formation, a control G4-region not affected by miR-9 depletion should be shown.

We have included in the revised version of our manuscript results on *Prap1* as negative control in the Figures 1e-f, Supplementary Figures 2d and 5a. In addition, we have selected another two loci without miR-9 enrichment as examples (shown

below), in which we observe that *miR-9*-LOF does not significantly affect G4 levels.

Minor remarks

8. Some rationale should be provided to justify the study of *miR-9* in lung cells. Is *miR-9* expressed at a high level in lung tissues or deregulated in lung cancers?

Following this suggestion from Reviewer 1, we added two sentences to the revised manuscript:

Changes in the manuscript:

(Lines 142-143)

“We were interested in lung cells since *miR-9* levels are increased in hyperproliferative lung diseases.”

(Lines 198-199)

“Due to these results and the higher levels of nuclear *miR-9* (Figure 1b), we focused on MLg cells.”

9. The subtitle “Mature *miR-9* is detected in the cell nucleus enriched at promoters and introns” could be replaced by “Mature *miR-9* is detected in the cell nucleus and is enriched at promoters and introns”

Subtitle was corrected according to the comment from Reviewer 1.

10. Figure 2b: the legend “non target” is not useful here as this category is not presented in this panel.

Removed from the previous Figure panel 2b that is now Figure 2e in the revised version of the manuscript.

11. Figure 2c: the blue color is misleading as the same color is used in panel 2b to represent downregulated genes only.

The blue color was changed to green. In any case, the box plot represents the reduction of the levels of the 2,439 transcripts that are shown in the Venn diagram.

12. Figure 2h: while motifs 1 and 3 show some level of similarities between mir-9 targets and G4 peaks, motifs 2 are not similar. This should be better explained in the text.

The previous Figure 2h became Figure 3e in the revised version of the manuscript. Further, we corrected the previous motif 2 and became motif 1 in the revised version of the manuscript. With the correction, this comment from Reviewer 1 is answered.

13. The authors claim to use HiChIP-seq with H3K4me3-specific antibodies to precipitate active SE (lines 291-293). However, the use of H3K4me3 will primarily precipitate promoters. Maybe the authors could modify the sentence to indicate that HiChIP-seq will allow to precipitate SEs interacting with active promoters.

We thank Reviewer 1 for this observation. We change the text of the revised manuscript according to this suggestion.

Changes in the manuscript:

(Lines 386-390)

“For this HiChIP-seq we used H3K4me3-specific antibodies to precipitate SE that physically interact with active promoters and chromatin from MLg cells that were transiently transfected with Ctrl or *miR9*-specific antagomiR probes, and non-treated or treated with TGFB1 (Figure 8b-e and Supplementary Figure 9a-d).”

14. Figure 6d: the number of interactions should be shown for each cluster.

The previous Figure 6d became Figure 7d in the revised version of the manuscript. We added to the Figure 7d the number of interactions.

15. Line 296: clusters 1 and 3 should be replaced by 1 and 4.

We thank Reviewer 1 for this observation. We corrected the text in the revised version of the manuscript.

16. Typo line 575: H3K4me3.

Thanks once again. Typo was corrected.

17. Line 348: “a so called a G-loop structure is generated” should be “a so called G-loop structure is generated”

The text was corrected.

We would like to thank Reviewer 1 for the constructive comments and the time invested in the peer-review of our manuscript. We respectfully hope that Reviewer 1 shares our positive opinion about the improved version of our manuscript and recommend it for publication at Nature Communications.

Reviewer #2 (Remarks to the Author):

The manuscript delineates a new regulatory axis mediated by the nuclear-localized microRNA, miR-9, and G-quadruplex formation that facilitates three-dimensional chromatin looping between the promoter regions and super-enhancers (SEs) of genes pertinent to TGF- β 1 signaling. The insights presented by the authors could potentially enrich our understanding of microRNA-mediated regulatory mechanisms in the nucleus. Nonetheless, the conclusions drawn require more thorough validation prior to consideration for publication in Nature Communications.

We appreciate the time and effort invested by Reviewer 2 in the peer-review of our manuscript, as well as the constructive suggestions from Reviewer 2.

Major Comments:

1. The investigation is limited to a single fibroblast cell type, within which the enrichment of miR-9 targets associated with TGF- β 1 response and cell proliferation has been identified. To thoroughly comprehend the nuclear functions of miR-9, it is imperative to extend the analysis of miR-9 targets and G-quadruplex formations across different cell types. The specificity of miR-9/G-quadruplex target interactions raises several questions: Is miR-9 ubiquitously associated with all transcriptionally active genes across cell types? Does it regulate a distinct subset of genes involved in specific cellular pathways? Or does it selectively modulate genes requiring high expression levels?

We thank Reviewer 2 for these comments that led us to new experiments that significantly improved our revised manuscript.

(1) We performed *miR-9* ChIRP seq in another cell line (MLE-12 cells). In addition, we did a ChIRP-seq using a probe specific for another miRNA that has been characterized in the cell nucleus (*Mirlet7d*). The results of these new

experiments are presented in new figure panels (Figures 1d-e, and Supplementary Figures 2a-d) and described in the manuscript as follows:

Change in the manuscript:

(Lines 155-169)

“To investigate the role of *miR-9* in the cell nucleus we performed a sequencing experiment after chromatin isolation by miRNA purification (ChIRP-seq) using chromatin from MLg cells and control (Ctrl) or *miR-9*-specific biotinylated antisense oligonucleotides for the precipitation of endogenous mature *miR-9* along with the chromatin bound to it (Figures 1d-e and Supplementary Figures 2a-f). To demonstrate the specificity of our ChIRP-seq experiment, we also used a probe specific for another miRNA characterized in the cell nucleus (miRNA lethal 7d, *Mirlet7d*, also known as *let-7d*)^{8,53}, and chromatin from MLE-12 cells. We detected specific enrichment of *miR-9* at loci without *Mirlet7d* in MLg and MLE-12 cells (Figure 1d, left, and Supplementary Figure 2b, left). Further, genome-wide binding profile analysis of *miR-9* in MLg cells revealed an increase in the number of *miR-9* peaks at promoters, transcription termination sides (TTS) and intronic regions compared to the negative control (Figure 1d, right), whereas in MLE-12 cells *miR-9* was enriched at TTS and intronic regions (Supplementary Figure 2b, right). Interestingly, the loci with *miR-9* enrichment were different in MLg and MLE-12 cells (Supplementary Figure 2c), suggesting that *miR-9* regulates different genes in different cells.”

Our new results not only demonstrate the specificity of our *miR-9* ChIRP-seq, but also show that both nuclear miRNAs (*miR-9* and *Mirlet7d*) regulate different loci in the same cell line (Figure 1d and Supplementary Figure 2b), and that *miR-9* regulates different loci in different cell lines (MLg and MLE-12 cells; Supplementary Figure 2c).

(2) In addition, we performed a new sequencing experiment following Cleavage Under Targets and Tagmentation (CUT&Tag) for genome-wide profiling of H3K4me3 in MLg cells that were transiently transfected with Ctrl or *miR9*-specific antagomiR to induce a *miR-9*-LOF. In parallel, we performed the same experiment in MLE-12 cells to investigate differences between cell lines. The results of these experiments are presented in new figure panels (Figures 2a-d and Supplementary Figures 3a-d) and described in the manuscript as follows:

Change in the manuscript:

(Lines 184-209)

“To further investigate a potential role of nuclear *miR-9* in transcription regulation, we performed a sequencing experiment following Cleavage Under Targets and Tagmentation (CUT&Tag) for high-resolution, genome-wide profiling of tri-methylated lysine 4 of histone 3 (H3K4me3) in MLg and MLE-12 cells that were transiently transfected with Ctrl or *miR9*-specific antagomiR to induce a *miR-9*-LOF (Figure 2a-e, Supplementary Figure 3a-d). Peak distribution analysis of the H3K4me3 CUT&Tag showed that 60.8% ($P = 0.01$) of the H3K4me3 broad domains in Ctrl transfected MLg cells were enriched with *miR-9* (Figure 2a), whereas 63% ($P < 0.01$) of the H3K4me3 broad domains were enriched with *miR-9* in Ctrl transfected MLE-12 cells (Supplementary Figure 3b). Interestingly, H3K4me3 levels at broad domains were significantly reduced in MLg cells from a median of 1.6 RPKM (IQR = 3.3) in Ctrl transfected cells to a median of 0.8 RPKM (IQR = 1.8; $P = 0.002$) following *miR-9*-LOF (Figure 2b, top), whereas the effects of *miR-9*-LOF in MLE-12 cells were not significant (Figure 2b, bottom). In addition, we observed that the loci of the H3K4me3 broad domains with *miR-9* enrichment were different in MLg and MLE-12 cells (Supplementary Figure 3c-d), confirming that *miR-9* regulates different genes in these two cell lines. Due to these results and the higher levels of nuclear *miR-9* (Figure 1b), we focused on MLg cells. Further peak distribution analysis showed that H3K4me3 broad domains were reduced from 27.6% in Ctrl transfected MLg cells to 22.1% after *miR-9*-LOF, whereas medium and narrow H3K4me3 domains increased (Figure

2c). Interestingly, the shift from H3K4me3 broad domains to medium and narrow domains following *miR-9*-LOF was significant at promoters but not at gene body and intergenic regions (Figure 2d). However, enrichment plots showed that H3K4me3 levels were reduced following *miR-9*-LOF in H3K4me3 broad and medium domains at promoter, gene body and intergenic regions (Figure 2e). The reduction of H3K4me3 levels after *miR-9*-LOF was confirmed by confocal microscopy after H3K4me3-specific immunostaining in Ctrl- and *miR-9*-antagomiR transfected MLg cells (Supplementary Figure 3e).”

Coming back to the questions raised by Reviewer 2 in this specific comment:

Is miR-9 ubiquitously associated with all transcriptionally active genes across cell types?

Mir-9 regulates different loci in different cell lines (MLg and MLE-12 cells; Supplementary Figure 2c).

Does it regulate a distinct subset of genes involved in specific cellular pathways?

In MLg cells, *miR-9* regulates genes that are related to fibroblasts, lung development, VEGF signaling and EGFR inhibitor resistance (Supplementary Figure 3d)

Or does it selectively modulate genes requiring high expression levels?

Mir-9 is required to maintain the basal transcriptional activity of its target genes (Figure 2f), as well as for the inducibility of its target genes by TGFB1 (Figures 7-9).

2. The association between miR-9 and super enhancers, as depicted in Figure 4a, is not immediately evident. The markers MED1 and H3K27ac cannot be solely relied upon to define SEs. A more robust algorithmic approach, such as the ROSE program that identifies clustered enhancers, should be employed. Given that SEs typically comprise several hundred clusters in a given cell, the analysis of miR-9 enrichment within SEs necessitates further scrutiny. Figures 5 and Supplementary

Figure 5 offer detailed views of individual enhancers as exemplars of SEs; however, an encompassing representation of entire SE regions is warranted.

To address this constructive comment from Reviewer 2, we performed a new H3K27ac CUT&Tag experiment in Ctrl or miR-9 antagomiR-transfected MLg cells (Supplementary Figure 6a) and analyzed the data using the rank-ordering of super-enhancers (ROSE) algorithm [PMID 23582322] to separate super-enhancer (SE) from typical enhancer (TYE). The results of this new experiment are presented in new figure panels (Figure 5e and Supplementary Figure 6a) and described in the manuscript as follows:

Change in the manuscript:

(Lines 304-310)

“To investigate a potential role of *miR-9* in enhancers, we performed H3K27ac CUT&Tag in Ctrl or *miR-9* antagomiR-transfected MLg cells (Supplementary Figure 6a) and analyzed the data using the rank-ordering of super-enhancers (ROSE) algorithm ¹¹ to separate SE from typical enhancer (TYE) (Figure 5d). We detected 1,649 SE in Ctrl transfected MLg cells that were significantly reduced to 1,084 ($P = 9.9E-142$) after *miR-9*-LOF supporting the requirement of *miR-9* for SE formation.”

In addition, to make our manuscript more interesting and increase the impact of our findings, we followed a suggestion from a colleague after an oral presentation of this project during a chromatin-related congress in France. To link the reducing effect of *miR-9*-LOF on G4 formation to SE or TYE, we decided to analyze our new G4-specific CUT&Tag experiment in MLg cells that were transiently transfected with Ctrl or *miR9*-specific antagomiR to induce a *miR-9*-LOF (Figures 3g, 4a, 5a, Supplementary Figures 4a-c and Supplementary Figure 5a) using the ROSE algorithm [PMID 23582322]. The results from this analysis are presented in new figure panels (Figure 5h-i) and described in the manuscript as follows:

Change in the manuscript:

(Lines 327-332)

“In addition, the levels of H3K27ac and G4 were higher at SE than at TYE (Figure 5h). However, *miR-9*-LOF reduced the levels of H3K27ac and G4 at both enhancer types. Remarkably, analysis of the G4 CUT&Tag using the ROSE algorithm showed that the number of SE containing G4s (1,753) was significantly reduced (937, $P = 2.5E-35$) after *miR-9*-LOF (Figure 5i) suggesting a role of G4s in SE formation in a *miR-9*-dependent manner.”

We believe that with these new results we were able to satisfactorily address this comment from Reviewer 2.

3. LOF experiments with *miR-9* have primarily been conducted to elucidate its role in genome organization and transcriptional activation within the nucleus. Nevertheless, the possibility that the observed effects may be attributable to alternative pathways involving *miR-9* cannot be discounted. Consequently, additional evidence is required to affirm the specific involvement of *miR-9* in G-quadruplex formation and chromatin looping.

We agree with this comment from Reviewer 2. To present further evidence demonstrating the requirement of *miR-9* in the cell nucleus for G4 formation we performed various experiments:

(1) We performed a G4-specific CUT&Tag experiment in MLg cells that were transiently transfected with Ctrl or *miR9*-specific antagomiR to induce a *miR-9*-LOF. In addition, we performed the same experiment in MLE-12 cells to investigate differences between cell lines. The results of these experiments are presented in new figure panels (Figures 3g, 4a, 5a, Supplementary Figures 4a-c and Supplementary Figure 5a) and described in the manuscript as follows:

Change in the manuscript:

(Lines 253-259)

“To demonstrate that *miR-9* is required for G4 formation, we analyzed by CUT&Tag using G4-specific antibodies chromatin from MLg and MLE-12 cells transiently transfected with Ctrl or

miR9-specific antagomiR (Figure 3g and Supplementary Figure 4a-c). Analysis of the G4 CUT&Tag data without or with filtering based on G4Hunter scores⁴¹ showed genome-wide reduction of G4s after *miR-9*-LOF in MLg and MLE-12 cells (Figure 3g and Supplementary Figure 4b, both left), thereby demonstrating the requirement of *miR-9* for G4 formation in both cell lines.”

(2) We did TaqMan-based miRNA enrichment analysis following chromatin-RNA immunoprecipitation (Ch-RIP) using G4 specific antibodies (Figure 3h) showing that G4s significantly bound mature *miR-9* and *miR-9*-LOF abolished this interaction.

(3) We performed a high-resolution mass spectrometry based proteomic approach after miRNA pulldown (miR-Pd) using the nuclear fraction of MLg or MLE-12 cells and biotinylated control miRNA (*mirctrl*) or *miR-9* as baits (Figure 3i, Supplementary Figures 4c-e and Source Data file 01) that demonstrates the interaction of mature miR-9 with proteins that have been reported to interact with G4s [PMID 34188089] in both cell lines. Interestingly, miR-9 did not significantly bind AGO1, AGO2 and MEX3D. However, gene Set Enrichment Analysis (GSEA) of the *miR-9* binding proteins (Figure 3j) showed that nuclear miR-9 interacted with proteins involved in G4s, Chromatin, RNA processing, Gene expression and Chromatin organization. All these results support an important role of miR-9 in G4 formation and transcription regulation.

These new figure panels are described in the manuscript as follows:

Change in the manuscript:

(Lines 263-282)

“To verify the interaction between mature *miR-9* and G4s, we performed TaqMan-based miRNA enrichment analysis following chromatin-RNA immunoprecipitation (Ch-RIP) using G4 specific antibodies (Figure 3h). G4s significantly bound mature *miR-9* and *miR-9*-LOF abolished this interaction, whereas G4s did not bind mature *Mirlet7f*, a miRNA used as negative

control, thereby showing the specificity of the interaction between mature *miR-9* and G4s. To further investigate this interaction and identify protein-binding partners of *miR-9* in the nucleus, we performed a high-resolution mass spectrometry based proteomic approach after miRNA pulldown (miR-Pd) using the nuclear fraction of MLg or MLE-12 cells and biotinylated control miRNA (*mirctrl*) or *miR-9* as baits (Figure 3i, Supplementary Figures 4d-f and Source Data file 01). Results from three independent experiments identified 169 proteins in the nuclear fraction of MLg cells and 233 proteins in the nuclear fraction of MLE-12 cells that were significantly enriched after *miR-9*-Pd. From the *miR-9* binding proteins, 20 proteins in the nuclear fraction of MLg cells and 58 proteins in the nuclear fraction of MLE-12 cells have been reported to interact with G4s⁴². Interestingly, *miR-9* did not significantly bind AGO1, AGO2 and MEX3D. However, Gene Set Enrichment Analysis (GSEA)⁴³ of the *miR-9* binding proteins (Figure 3j) showed that nuclear *miR-9* interacted with proteins involved in G4s ($P = 1.23E-4$), Chromatin ($P = 1.5E-3$), RNA processing ($P = 1.65E-49$), Gene expression ($P = 1.56E-54$) and Chromatin organization ($P = 9.3E-3$). All these results support an important role of *miR-9* in G4 formation and transcription regulation.”

We hope that Reviewer 2 acknowledges that all these results together support a role of *miR-9* in G4 formation. It will be the scope of future manuscripts to further elucidate the mechanism.

4. Figure 1b illustrates a variable distribution of miR-9 between nuclear and cytoplasmic compartments. It would be informative to test whether miR-9 also contributes to transcription and genome organization in cell types where its nuclear presence is minimal, such as human lung fibroblasts (hLF).

We would like to refer Reviewer 2 to our answer to her/his comment 1, in which we already described new experiments (*miR-9* ChIRP-seq and H3K4me3 CUT&Tag) that were performed in MLg and MLE-12 cells to demonstrate differences in the role of *miR-9* in fibroblasts (MLg cells) and epithelial cells

(MLE-12 cells). The levels of nuclear *miR-9* are relatively low in MLE-12 cells. Nevertheless, as described above, *miR-9* is enriched in different loci in MLE-12 cells as compared to MLg cells (Supplementary Figure 2d) and is required for H3K4me3 broad domains that are also located in different loci regulating different genes (Supplementary Figure 3c-d)

In addition, we performed a *miR-9*-specific FISH in primary human lung fibroblast (hLF) from control donors (Ctrl hLF) as cells with relatively low levels of *miR-9* in the cell nucleus (see Figure 1b). Confirming the results from the TaqMan assay-based, *miR-9*-specific expression analysis after cellular fractionation presented in Figure 1b, we observed by RNA FISH the majority of *miR-9* in the cytosol of Ctrl hLF (Supplementary Figure 1d, top).

Moreover, we performed *miR-9*-specific FISH also in hLF isolated from the lung of patients with idiopathic pulmonary fibrosis (IPF hLF), a lethal interstitial lung disease involving TGFB1 signaling. To our surprise, we observed by RNA FISH that the majority of *miR-9* is in the cell nucleus of IPF hLF, suggesting a translocation mechanism of *miR-9* into the cell nucleus potentially related to IPF (Supplementary Figure 1d, middle). Further, *miR-9*-LOF in IPF hLF reduced the levels of *miR-9* (Supplementary Figure 1d, bottom).

The results of these new RNA FISH experiments are presented in a new figure panel (Supplementary Figure 1c) and described in the manuscript as follows:

Change in the manuscript:

(Lines 145-155)

“Further, the nuclear localization of *miR-9* was confirmed by RNA fluorescence in situ hybridization (FISH) in MLg cells (Figure 1c and Supplementary Figure 1c) and hLF from control donors (Ctrl hLF) or patients with idiopathic pulmonary fibrosis (IPF hLF; Supplementary Figure 1d), a lethal interstitial lung disease involving TGFB1 signaling³⁷. In MLg cells, we detected *miR-9* in specific regions of the nuclei, whereas the levels of *miR-9* were reduced after loss-of-function (LOF) experiments using unlabeled *miR-9*-specific

antagomiR probes. In Ctrl hLF, the intensity of *miR-9* FISH was higher in the cytosol than in the nucleus, whereas in IPF hLF the majority of *miR-9* was detected in the cell nucleus, pointing to a translocation mechanism of *miR-9* into the cell nucleus potentially related to IPF. Further, *miR-9*-LOF in IPF hLF reduced the levels of *miR-9*.”

Additional experiments investigating the role of *miR-9* in transcription regulation and 3D genome organization during IPF are currently ongoing in our laboratory. The results of these experiments will be the scope of the follow up manuscript further elucidating the mechanism presented in the present manuscript at Nature Communications.

Minor Comments:

5. In Figure 1, ChIRP-seq should be complemented with a negative control for validation purposes.

We would like to refer Reviewer 2 to the first part of our answer to her/his comment 1 (please see above). Briefly, we performed *miR-9* ChIRP seq in another cell line (MLE-12 cells). In addition, we did a ChIRP-seq using a probe specific for another miRNA that has been characterized in the cell nucleus (*Mirlet7d*). The results of these new experiments are presented in new figure panels (Figures 1d-e, and Supplementary Figures 2a-d).

Our new results not only demonstrate the specificity of our *miR-9* ChIRP-seq, but also show that both nuclear miRNAs (*miR-9* and *Mirlet7d*) regulate different loci in the same cell line (Figure 1d and Supplementary Figure 2b), and that *miR-9* regulates different loci in different cell lines (MLg and MLE-12 cells; Supplementary Figure 2c).

6. Figure 2a should include examples of H3K4me3 ChIP-seq track data for assessment purposes. The confocal microscopy data presented in Supplementary

Figure 2b does not substantiate the related claims; notably, all H3K4me3 signals seem to diminish following miR-9 LOF. The examples depicted in Figure 3a are not sufficiently persuasive.

We would like to refer Reviewer 2 to the second part of our answer to her/his comment 1 (please see above). Briefly, we performed a new sequencing experiment following Cleavage Under Targets and Tagmentation (CUT&Tag) for genome-wide profiling of H3K4me3 in MLg cells that were transiently transfected with Ctrl or *miR9*-specific antagomiR to induce a *miR-9*-LOF. In parallel, we performed the same experiment in MLE-12 cells to investigate differences between cell lines. The results of these experiments are presented in new figure panels (Figures 2a-d and Supplementary Figures 3a-d).

The results of H3K4me3 ChIP-seq from the previous Figure 2a were confirmed by a new H3K4me3 CUT&Tag experiment that is shown Figure 2b of the revised version of our manuscript. Moreover, the new figure panels generated using the new data from the H3K4me3 CUT&Tag (Figures 2a, 2c-d and Supplementary Figures 3a-d) further strengthen our interpretation of the role of nuclear miR-9 on H3K4me3 broad domains.

The previous Supplementary Figure 2b became Supplementary Figure 3e in the revised version of our manuscript. The supposed discrepancy commented by Reviewer 2, between the data using Next Generation Sequencing technologies and the results obtained by microscopy after H3K4me3-specific immunostaining can be explained by the results presented in the new Figure 2d and described in the text as follows:

Change in the manuscript:

(Lines 202-209)

“Interestingly, the shift from H3K4me3 broad domains to medium and narrow domains following *miR-9*-LOF was significant at promoters but not at gene body and intergenic regions (Figure 2d). However, enrichment plots showed that H3K4me3 levels were reduced following *miR-9*-LOF in H3K4me3 broad and medium domains at promoter, gene body and intergenic regions (Figure 2e). The reduction of H3K4me3 levels after *miR-9*-LOF was confirmed by

confocal microscopy after H3K4me3-specific immunostaining in Ctrl- and *miR-9*-antagomiR transfected MLg cells (Supplementary Figure 3e).”

Another possible explanation for the supposed discrepancy commented by Reviewer 2, between the data using Next Generation Sequencing technologies and the results obtained by microscopy after H3K4me3-specific immunostaining may be that the microscopy method will be more sensitive to the *miR-9*-LOF due to the characteristic of the antibodies used.

The previous Figure 3a was substituted by the new Figure 5a, in which we exchanged the previous H3K4me3 ChIP-seq data by the newly performed H3K4me3 CUT&Tag data. (1) The fact that the new H3K4me3 CUT&Tag results confirmed the previous H3K4me3 ChIP-seq results, (2) together with the sensitivity and specificity of the CUT&Tag-based genome-wide profiling of H3K4me3 are two arguments that should make the presented results sufficiently persuasive. We respectfully hope that Review 2 shares our perspective.

7. Contrary to assertions made in the text, Figure 2c shows that *miR-9* targets in control samples exhibit a lower median expression level compared to non-targets, challenging the notion that *miR-9* elevates basal transcriptional activity in target genes.

We appreciate this comment from Reviewer 2 that attracted our attention to potential misunderstandings. To avoid these misunderstandings, we have accordingly changed the text of the revise manuscript.

8. Figure 2d-i should include comparative heatmap data for non-*miR-9* target genes that are expressed, for comparison.

The previous Figures 2d-i became Figures 3a-f in the revised version of our manuscript. Due to the newly generated results addressing the comments of the 3 Reviewers together with the limitations in the number of Figures, we did not include the requested heat maps into the manuscript. However, we generated a Figure including these heat maps for the perusal of Reviewer 2 (please see below).

As expected, we also observed enrichment at TSS of the non-*miR-9* target genes, since the heat maps show enrichment of factors that are general features of open chromatin and the non-*miR-9* target genes are also expressed.

9. Figures 3a and Supplementary Figure 3 fail to convincingly demonstrate colocalization between *miR-9* ChIRP and G4 ChIP signals, raising questions about their functional interplay. It remains unclear whether *miR-9* is essential for G-quadruplex formation and, if so, by what mechanism.

The previous Figure 3a and Supplementary Figure 3 became Figure 4a and Supplementary Figure 5a, respectively, in the revised version of our manuscript. Specifically in these two figure panels we have added results from the newly performed G4 CUT&Tag in Ctrl or *miR-9*-LOF transfected MLg cells. The results presented clearly show a reduction of G4 after *miR*-LOF.

Complementing the results from the G4 CUT&Tag and supporting a role of *miR-9* in G4 formation, we performed further experiments that were already mentioned in our answer to the comment 3 from Reviewer 2 (please see above). Briefly:

(2) We did TaqMan-based miRNA enrichment analysis following chromatin-RNA immunoprecipitation (Ch-RIP) using G4 specific antibodies (Figure 3h) showing that G4s significantly bound mature *miR-9* and *miR-9*-LOF abolished this interaction.

(3) We performed a high-resolution mass spectrometry based proteomic approach after miRNA pulldown (miR-Pd) using the nuclear fraction of MLg or MLE-12 cells and biotinylated control miRNA (*mirctrl*) or *miR-9* as baits (Figure 3i, Supplementary Figures 4c-e and Source Data file 01) that demonstrates the interaction of mature miR-9 with proteins that have been reported to interact with G4s [PMID 34188089] in both cell lines.

We hope that Reviewer 2 recognizes that all these results together support a role of *miR-9* in G4 formation. It will be the scope of future manuscripts to further elucidate the mechanism.

10. Analyses across the genome scale (e.g., Figures 3b, 6c, 8) should be presented rather than relying on select examples to substantiate claims.

We agree with this comment from Reviewer 2. In fact, significant amount of the results presented in our manuscript are based on “omics” approaches, not only on NGS-based methods, but also on high-resolution mass spectrometry based proteomic approach after miRNA pulldown (miR-Pd, please see the new Figure 3i and Supplementary Figures 4d-f). Please, see the table at the beginning of this document with a list of new experiments added during the revision. However, the relevance of experiments carried out with classical techniques of molecular biology or biochemistry to confirm the results obtained by new technologies should not be diminished neither underestimated. Beside the fact that the new technologies may be not accessible to laboratories with limited resources. Moreover, the results that we present in the figure panels, in which we used classical molecular biology methods, confirm the results obtained by “omics” approaches.

11. On page 12, there appears to be a typographical error where 'cluster 1 and 3' should be corrected to 'cluster 1 and 4'.

Thanks for this observation. The typo was corrected.

12. The claim that miR-9 is necessary for chromatin alterations induced by TGF- β 1, as suggested by Figure 6c, requires genome-wide evidence. Additionally,

HiChIP-seq analysis should be integrated with comprehensive miR-9 ChIRP-seq data to conclusively demonstrate miR-9's direct role in chromatin looping. It is also pertinent to ascertain whether TGF- β 1 induces the recruitment of miR-9 to its targets.

The previous Figure 6c showing snapshots of H3K4me3 ChIP-seq of selected *miR-9* target genes became Figure 7h in the revised version of our manuscript.

To answer this comment from Reviewer 2, we have performed new analysis of our previous H3K4me3 ChIP-seq experiment, and we have performed new experiments that should strengthen our interpretation of our data:

(1) Figure 7f shows in TGFB responsive promoters, increased enrichment of H3K4me3 after TGFB1 treatment in *miR-9*-dependent manner, supporting the results presented in the snapshots of Figure 7h and the chromatin analysis by ChIP-qPCR presented in the Figure 9a.

(2) Figure 7g shows an increase of the breadth of H3K4me3 peaks after TGFB1 treatment also in *miR-9*-dependent manner. These results correlate with the results presented in the new Figures 2a-d showing that *miR-9*-LOF reduced the width of H3K4me3 domains genome wide, but also at promoters.

(3) Figures 7c-d show by H3K4me3 immunostainings in MLg cells supporting that TGFB1 treatment increased the global levels of H3K4me3 in a *miR-9*-dependent manner.

(4) Figure 7e shows by qPCR after ChIRP using *miR-9*-specific biotinylated antisense oligonucleotides the enrichment of *miR-9* at the promoters of *miR-9* target genes after TGFB1 treatment, thereby supporting our interpretation of the results.

These new figure panels are described in the manuscript as follows:

Change in the manuscript:

(Lines 357-381)

“Supporting these observations, RNA FISH and immunostaining in MLg cells showed that TGFB1 treatment increased the levels of *miR-9*, G4 and H3K4me3 in *miR-9*-dependent manner (Figure 7c-d). Similarly as in IPF hLF (Supplementary Figure 1d), the majority of *miR-9* was detected in the cell nucleus of MLg cells after TGFB1 treatment. Interestingly, we detected significantly increased enrichment of *miR-9* at promoters of *miR-9* target genes after TGFB1 treatment in MLg cells that were analyzed by qPCR after ChIRP using *miR-9*-specific biotinylated antisense oligonucleotides (Figure 7e). These results were complemented by H3K4me3 ChIP-seq in MLg cells that were transiently transfected with Ctrl or *miR9*-specific antagomiR probes, and non-treated or treated with TGFB1 (Figures 7f-h and Supplementary Figure 8a-c). H3K4me3 levels significantly increased after TGFB1 treatment at promoters of TGFB-responsive genes in *miR-9*-dependent manner (Figure 7f, left). These effects were not observed at the promoters of genes that did not respond to TGFB1 treatment (Figure 7f, right) supporting the specificity of the effects observed. By checking on H3K4me3 levels at loci, in which we detected G4s by CUT&Tag, we observed that TGFB1 did not significantly affect H3K4me3 levels, whereas the combination of TGFB1 and *miR-9*-LOF reduced H3K4me3 levels at these loci (Supplementary Figure 8b). Interestingly, the breadth of H3K4me3 peaks increased after TGFB1 treatment also in *miR-9*-dependent manner (Figure 7g). Further, loci visualization of the selected *miR-9* target genes (*Zdhhc5*, *Ncl*, *Lzts2*, *Hdac7* and *Ep300*) using the IGV genome browser showed H3K4me3 enrichment at the promoters in non-treated, and Ctrl antagomiR transfected MLg cells that increased after TGFB1 treatment (Figure 7h and Supplementary Figure 8c). However, a combination of TGFB1 treatment and *miR9*-specific antagomiR transfection showed that *miR-9*-LOF counteracted the effect caused by TGFB1 demonstrating the requirement of *miR-9* for the chromatin changes induced by TGFB1 and suggesting its requirement for TGFB1-inducibility of the analyzed genes, as shown below.”

Once again, we would like to thank Reviewer 2 for the constructive comments that led us to an improved version of our manuscript. We respectfully hope that Reviewer 2 shares our positive opinion about the quality and novelty of our results and recommend our manuscript for publication at Nature Communications.

Reviewer #3 (Remarks to the Author):

This study explores a novel function of miRNA in modulating genome organization within the nucleus. The authors report that miR-9, traditionally associated with cytoplasmic functions, exhibits nuclear localization in specific cells, potentially influencing genome interactions. Utilizing miR-9-based ChIRP-seq, the authors propose an interaction between miR-9 and a subset of super enhancers and promoters containing G-quadruplex motifs. Through various high-throughput techniques such as RNA-seq and ChIP-seq, the authors conclude that miR-9 depletion alters chromatin features at these loci, resulting in decreased gene expression. Additionally, they demonstrate the role of miR-9 in TGFB1-mediated promoter-enhancer looping.

The premise of the study is intriguing, particularly in light of emerging evidence suggesting nuclear localization of Argonaute proteins under certain physiological conditions. If validated, the proposed model would offer valuable insights into the role of nuclear miRNAs, thereby significantly contributing to the field.

However, despite the extensive data provided, I find the evidence to be preliminary and the proposed model unconvincing. Firstly, the manuscript lacks a mechanistic understanding of key aspects. Questions regarding the nuclear translocation of miR-9, the potential involvement of Ago proteins in its nuclear function, the specificity of miR-9 in targeting G-quadruplex motifs, and its selective role in TGFB1 regulation remain unanswered. Without elucidating these mechanisms, establishing such a controversial model requires robust evidence, which is currently lacking. In fact, most conclusions are based on associations rather than causal relationships, with limited support from approaches such as mutagenesis studies. Furthermore, the presentation of analyses is unclear and fails to convincingly support many of the conclusions drawn. For example, how are targets of nuclear miR-9 defined/selected? Lastly, the method used for miR-9 LOF through antagomir treatment raises concerns about potential off-target effects, which may confound the interpretation of results. This study can clearly benefit from including additional controls.

Given these shortcomings, I cannot recommend the publication of this manuscript in its current form.

We thank Reviewer 3 for the time and efforts invested during the peer-review of our manuscript. We also appreciate the comments from Reviewer 3 related to the originality of our model, the novelty of the methods that we implemented and the extensive amount of data that we presented in the previous version of our manuscript. Nevertheless, we also recognize the skepticism expressed by Reviewer 3, which we respectfully believe to have addressed by answering to her/his comments supported by a set of newly performed experiments that further strengthen our model and support our interpretation of the data. We appreciate the constructive comments from Reviewer 3 that helped us to improve our manuscript.

Specific points:

1. Figure 1a: The observation that miR-9 shares the same nuclear localization signal as miR-126 raises questions about the potential involvement of Mex3a, known to mediate miR-126 nuclear transportation, in the nuclear localization of miR-9. It is important to test whether Mex3a is also responsible for shuttling miR-9 into the nucleus. Experimentally manipulating Mex3a to modulate miR-9 shuttling and subsequently validating the functional consequences of nuclear miR-9 would significantly strengthen the conclusions of this study.

We thank Reviewer 3 for this specific comment. Before starting experiments targeting Mex3a, as suggested by Reviewer 3, we decided to perform a high-resolution mass spectrometry based proteomic approach after miRNA pulldown (miR-Pd) using the nuclear fraction of MLg or MLE-12 cells and biotinylated control miRNA (*mirctrl*) or *miR-9* as baits (Figure 3i, Supplementary Figures 4c-e and Source Data file 01). One aim of this experiment was to determine whether *miR-9* interacts with any of the Mex-3 RNA Binding Family proteins. However, we found that only MEX3D was pulled down by *miR-9*, but not in statistically significant manner as compared to the negative control (*mirctrl*). Similar results were obtained with the members of the Argonaute family of proteins AGO1 and AGO2. The results of our mass spectrometry based proteomic approach led us to desist from performing experiments targeting proteins of the Mex-3 RNA Binding Family nor the Argonaute family.

We interpreted this comment from Reviewer 3 as suggestion to elucidate mechanistic aspects of the model of transcriptional regulation that we are proposing. Thus, further analysis of our high-resolution mass spectrometry based proteomic approach after miR-Pd demonstrate the interaction of mature *miR-9* with proteins that have been reported to interact with G4s [PMID 34188089]. Moreover, Gene Set Enrichment Analysis (GSEA) of the *miR-9* binding proteins (Figure 3j) showed that nuclear *miR-9* interacted with proteins involved in G4s, Chromatin, RNA processing, Gene expression and Chromatin organization. All these results support an important role of *miR-9* in G4 formation and transcription regulation.

A second experiment confirming the interaction of *miR-9* with G4s is a TaqMan-based miRNA enrichment analysis following chromatin-RNA immunoprecipitation (Ch-RIP) using G4 specific antibodies (Figure 3h) showing that G4s significantly bound mature *miR-9* and *miR-9*-LOF abolished this interaction.

A third experiment providing further mechanistic insight was a new sequencing experiment following Cleavage Under Targets and Tagmentation (CUT&Tag) for genome-wide profiling G4 in MLg cells that were transiently transfected with Ctrl or *miR9*-specific antagomiR to induce a *miR-9*-LOF. In addition, we performed the same experiment in MLE-12 cells to investigate differences between cell lines. The results of these experiments are presented in new figure panels (Figures 3g, 4a, 5a, Supplementary Figures 4a-c and Supplementary Figure 5a).

These new figure panels are described in the manuscript as follows:

Change in the manuscript:

(Lines 253-282)

“To demonstrate that *miR-9* is required for G4 formation, we analyzed by CUT&Tag using G4-specific antibodies chromatin from MLg and MLE-12 cells transiently transfected with Ctrl or *miR9*-specific antagomiR (Figure 3g and Supplementary Figure 4a-c). Analysis of the G4 CUT&Tag data without or with filtering based on G4Hunter scores⁴¹ showed genome-wide

reduction of G4s after *miR-9*-LOF in MLg and MLE-12 cells (Figure 3g and Supplementary Figure 4b, both left), thereby demonstrating the requirement of *miR-9* for G4 formation in both cell lines. However, correlating with the levels of nuclear *miR-9* in both cell lines (Figure 1b), the reducing effect of *miR-9*-LOF on G4 levels was more pronounced in MLg cells at TSS of *miR-9* target genes (Figure 3g and Supplementary Figure 4b, both right). Interestingly, the majority of the loci with G4s were different in MLg and MLE-12 cells (Supplementary Figure 4c), supporting that G4s are involved in the regulation of different genes in both cell lines. To verify the interaction between mature *miR-9* and G4s, we performed TaqMan-based miRNA enrichment analysis following chromatin-RNA immunoprecipitation (Ch-RIP) using G4 specific antibodies (Figure 3h). G4s significantly bound mature *miR-9* and *miR-9*-LOF abolished this interaction, whereas G4s did not bind mature *Mirlet7f*, a miRNA used as negative control, thereby showing the specificity of the interaction between mature *miR-9* and G4s. To further investigate this interaction and identify protein-binding partners of *miR-9* in the nucleus, we performed a high-resolution mass spectrometry based proteomic approach after miRNA pulldown (miR-Pd) using the nuclear fraction of MLg or MLE-12 cells and biotinylated control miRNA (*mirctrl*) or *miR-9* as baits (Figure 3i, Supplementary Figures 4d-f and Source Data file 01). Results from three independent experiments identified 169 proteins in the nuclear fraction of MLg cells and 233 proteins in the nuclear fraction of MLE-12 cells that were significantly enriched after *miR-9*-Pd. From the *miR-9* binding proteins, 20 proteins in the nuclear fraction of MLg cells and 58 proteins in the nuclear fraction of MLE-12 cells have been reported to interact with G4s⁴². Interestingly, *miR-9* did not significantly bind AGO1, AGO2 and MEX3D. However, Gene Set Enrichment Analysis (GSEA)⁴³ of the *miR-9* binding proteins (Figure 3j) showed that nuclear *miR-9* interacted with proteins involved in G4s ($P = 1.23E-4$), Chromatin ($P = 1.5E-3$), RNA processing ($P = 1.65E-49$), Gene expression ($P = 1.56E-54$) and Chromatin organization ($P = 9.3E-3$). All these results support an important role of *miR-9* in G4 formation and transcription regulation.”

We hope that Reviewer 3 acknowledges that all these new results added to the revised version of our manuscript together with the results that were included in the previous version of the manuscript (e.g. current Figures 1f, 4a-b, 6a-b, 7e, 9a-b) demonstrate a role of *miR-9* in G4 formation and transcriptional regulation. It will be the scope of future manuscripts to further elucidate the mechanism.

2. Figure 1b. It is striking that miR-9 exclusively resides inside nucleus in MLg cells, which authors opted to focus on in the rest of the study. Nonetheless, I would strongly suggest that authors include cells such as hLF where nuclear miR-9 is limited in the following functional characterization. Incorporating hLF cells as a negative control would serve to validate that the observations are indeed specific to miR-9.

To address this constructive comment from Reviewer 3, we performed various experiments in MLE-12 cells and primary human lung fibroblasts (hLF) as cells, in which the relative amount of *miR-9* is low as compared to the amount of *miR-9* in the cytosol of the respective cells, or with the amount of nuclear *miR-9* in the nucleus of MLg cells (Figure 1b):

(1) As described in our answer to the comment 4 from Reviewer 3, we performed *miR-9* ChIRP seq in MLg and MLE-12 cells. In addition, we did a ChIRP-seq using a probe specific for another miRNA that has been characterized in the cell nucleus (*Mirlet7d*). The results of these new experiments are presented in new figure panels (Figures 1d-e, and Supplementary Figures 2a-d). We would like to refer Reviewer 3 to our answer to comment 4 below containing also the text with the description in the revised version of the manuscript.

(2) As described in our answer to the comment 6 from Reviewer 3, we did CUT&Tag for genome-wide profiling of H3K4me3 in MLg cells that were transiently transfected with Ctrl or *miR9*-specific antagomiR to induce a *miR-9*-LOF. In parallel, we performed the same experiment in MLE-12 cells to investigate differences between cell lines. The results of these experiments are presented in new figure panels (Figures 2a-d and Supplementary Figures 3a-d). We would like to refer Reviewer 3 to our answer to comment 6 below containing also the text with the description in the revised version of the manuscript.

(3) As described in our answer to the comment 1 from Reviewer 3, we did CUT&Tag for genome-wide profiling of G4 in MLg cells that were transiently transfected with Ctrl or *miR9*-specific antagomiR to induce a *miR-9*-LOF. In addition, we performed the same experiment in MLE-12 cells to investigate differences between cell lines. The results of these experiments are presented in new figure panels (Figures 3g, 4a, 5a, Supplementary Figures 4a-c and Supplementary Figure 5a). We would like to refer Reviewer 3 to our answer to comment 1 above containing also the text with the description in the revised version of the manuscript.

(4) As described in our answer to the comment 1 from Reviewer 3, we performed a high-resolution mass spectrometry based proteomic approach after miR-Pd in MLg and MLE-12 cells, which demonstrate in both cell lines the interaction of mature *miR-9* with proteins that have been reported to interact with G4s [PMID 34188089]. We would like to refer Reviewer 3 to our answer to comment 1 above containing also the text with the description in the revised version of the manuscript.

(5) We performed a *miR-9*-specific FISH in primary human lung fibroblast (hLF) from control donors (Ctrl hLF) as other cells with relatively low levels of *miR-9* in the cell nucleus. Confirming the results from the TaqMan assay-based, *miR-9*-specific expression analysis after cellular fractionation presented in Figure 1b, we observed by RNA FISH the majority of *miR-9* in the cytosol of Ctrl hLF (Supplementary Figure 1d, top). We would like to refer Reviewer 3 to our answer to comment 3 from Reviewer 3 below containing further details of this experiment, as well as also the text with the description in the revised version of the manuscript.

3. Figure 1c: Authors should consider performing RNA FISH on hLF cells as well to confirm that the signal detected corresponds to mature miR-9.

Following this suggestion from reviewer 3, we performed a *miR-9*-specific FISH in primary human lung fibroblast (hLF) from control donors (Ctrl hLF) as other cells with relatively low levels of *miR-9* in the cell nucleus. Confirming the results from the TaqMan assay-based, *miR-9*-specific expression analysis after cellular

fractionation presented in Figure 1b, we observed by RNA FISH the majority of *miR-9* in the cytosol of Ctrl hLF (Supplementary Figure 1d, top).

In addition, we performed *miR-9*-specific FISH also in hLF isolated from the lung of patients with idiopathic pulmonary fibrosis (IPF hLF), a lethal interstitial lung disease involving TGFB1 signaling. To our surprise, we observed by RNA FISH that the majority of *miR-9* is in the cell nucleus of IPF hLF, suggesting a translocation mechanism of *miR-9* into the cell nucleus potentially related to IPF (Supplementary Figure 1d, middle). Further, *miR-9*-LOF in IPF hLF reduced the levels of *miR-9* (Supplementary Figure 1d, bottom).

The results of these new RNA FISH experiments are presented in a new figure panel (Supplementary Figure 1c) and described in the manuscript as follows:

Change in the manuscript:

(Lines 145-155)

“Further, the nuclear localization of *miR-9* was confirmed by RNA fluorescence in situ hybridization (FISH) in MLg cells (Figure 1c and Supplementary Figure 1c) and hLF from control donors (Ctrl hLF) or patients with idiopathic pulmonary fibrosis (IPF hLF; Supplementary Figure 1d), a lethal interstitial lung disease involving TGFB1 signaling³⁷. In MLg cells, we detected *miR-9* in specific regions of the nuclei, whereas the levels of *miR-9* were reduced after loss-of-function (LOF) experiments using unlabeled *miR-9*-specific antagomiR probes. In Ctrl hLF, the intensity of *miR-9* FISH was higher in the cytosol than in the nucleus, whereas in IPF hLF the majority of *miR-9* was detected in the cell nucleus, pointing to a translocation mechanism of *miR-9* into the cell nucleus potentially related to IPF. Further, *miR-9*-LOF in IPF hLF reduced the levels of *miR-9*.”

4. Figure 1d: Only showing enrichment over background might be misleading. How can we be certain that it is specific to *miR-9*? What is the enrichment relative to the control (pulldown with control LNA oligos)?

We appreciate this comment from Reviewer 3 as suggestion to demonstrate the specificity of the *miR-9* ChIRP-seq. With the aim of demonstrating the specificity of our *miR-9* ChIRP-seq, and following also the comment 1 from Reviewer 1 and comment 1 from Reviewer 2, we implemented two strategies: (1) we performed *miR-9* ChIRP seq in another cell line (MLE-12 cells) and (2) we did a ChIRP-seq using a probe specific for another miRNA that has been characterized in the cell nucleus (*Mirlet7d*). The results of these strategies are presented in new figure panels (Figures 1d-e, and Supplementary Figures 2a-d) and described in the manuscript as follows:

Change in the manuscript:

(Lines 155-169)

“To investigate the role of *miR-9* in the cell nucleus we performed a sequencing experiment after chromatin isolation by miRNA purification (ChIRP-seq) using chromatin from MLg cells and control (Ctrl) or *miR-9*-specific biotinylated antisense oligonucleotides for the precipitation of endogenous mature *miR-9* along with the chromatin bound to it (Figures 1d-e and Supplementary Figures 2a-f). To demonstrate the specificity of our ChIRP-seq experiment, we also used a probe specific for another miRNA characterized in the cell nucleus (miRNA lethal 7d, *Mirlet7d*, also known as *let-7d*)^{8,53}, and chromatin from MLE-12 cells. We detected specific enrichment of *miR-9* at loci without *Mirlet7d* in MLg and MLE-12 cells (Figure 1d, left, and Supplementary Figure 2b, left). Further, genome-wide binding profile analysis of *miR-9* in MLg cells revealed an increase in the number of *miR-9* peaks at promoters, transcription termination sides (TTS) and intronic regions compared to the negative control (Figure 1d, right), whereas in MLE-12 cells *miR-9* was enriched at TTS and intronic regions (Supplementary Figure 2b, right). Interestingly, the loci with *miR-9* enrichment were different in MLg and MLE-12 cells (Supplementary Figure 2c), suggesting that *miR-9* regulates different genes in different cells.”

Our new results not only demonstrate the specificity of our *miR-9* ChIRP-seq, but also show that both nuclear miRNAs (*miR-9* and *Mirlet7d*) regulate different loci in the same cell line (Figure 1d and Supplementary Figure 2b), and that *miR-9* regulates different loci in different cell lines (MLg and MLE-12 cells; Supplementary Figure 2c).

5. Figure 1e: The definition of targets (genome regions) of nuclear miR-9 requires clarification. Specifically, the authors should elaborate on how the targets, indicated as "n=2637," were selected. It is essential to clarify whether these targets were chosen based on relative enrichment over control and emphasize the significance of this data, as subsequent analyses in the study heavily rely on it.

We thank Reviewer 3 for attracting our attention to this point and take care in revised manuscript to avoid misunderstandings.

In main text of our revised manuscript, we refer to the loci with *miR-9* enrichment determined by ChIRP-seq (Figures 1d-e and Supplementary Figures 2a-f) as “putative *miR-9* target genes”.

Following confirmation by RNA-seq that the expression from these genes is altered by *miR-9*-LOF (Figure 2f-g and Supplementary Figure 3f), we refer to the loci coding for the transcripts significantly changed after *miR-9*-LOF as “*miR-9* target genes”.

For the sake of clarity, we have simplified the corresponding text in the revised manuscript as follows:

Change in the manuscript:

(Lines 218-221)

“The genes coding for the transcripts significantly affected by *miR-9*-LOF will be further referred to as *miR-9* target genes. We also observed genes coding for transcripts that were not significantly affected by *miR-9*-LOF (Figure 2g, bottom, non-targets).”

6. The results presented in Figure 2 would benefit from the inclusion of additional controls. Specifically, authors should consider using cells where miR-9 is not expressed or absent in the nucleus to provide a comparative context for the observed effects.

As listed in our answer to comment 2 from Reviewer 3, we have performed several experiments in MLE-12 cells and hLF as cells with relatively low amount of nuclear *miR-9* as compared to MLg cells. One of these experiments is a H3K4me3 CUT&Tag for genome-wide profiling of H3K4me3 in MLg or MLE-12 cells that were transiently transfected with Ctrl or *miR9*-specific antagomiR to induce a *miR-9*-LOF to investigate differences between cell lines. The results of these experiments are presented in new figure panels (Figures 2a-d and Supplementary Figures 3a-d) and described in the manuscript as follows:

Change in the manuscript:

(Lines 184-209)

“To further investigate a potential role of nuclear *miR-9* in transcription regulation, we performed a sequencing experiment following Cleavage Under Targets and Tagmentation (CUT&Tag) for high-resolution, genome-wide profiling of tri-methylated lysine 4 of histone 3 (H3K4me3) in MLg and MLE-12 cells that were transiently transfected with Ctrl or *miR9*-specific antagomiR to induce a *miR-9*-LOF (Figure 2a-e, Supplementary Figure 3a-d). Peak distribution analysis of the H3K4me3 CUT&Tag showed that 60.8% ($P = 0.01$) of the H3K4me3 broad domains in Ctrl transfected MLg cells were enriched with *miR-9* (Figure 2a), whereas 63% ($P < 0.01$) of the H3K4me3 broad domains were enriched with *miR-9* in Ctrl transfected MLE-12 cells (Supplementary Figure 3b). Interestingly, H3K4me3 levels at broad domains were significantly reduced in MLg cells from a median of 1.6 RPKM (IQR = 3.3) in Ctrl transfected cells to a median of 0.8 RPKM (IQR = 1.8; $P = 0.002$) following *miR-9*-LOF (Figure 2b, top), whereas the effects of *miR-9*-LOF in MLE-12 cells were not significant (Figure 2b, bottom). In addition, we observed that the loci of the H3K4me3 broad domains with *miR-9*

enrichment were different in MLg and MLE-12 cells (Supplementary Figure 3c-d), confirming that *miR-9* regulates different genes in these two cell lines. Due to these results and the higher levels of nuclear *miR-9* (Figure 1b), we focused on MLg cells. Further peak distribution analysis showed that H3K4me3 broad domains were reduced from 27.6% in Ctrl transfected MLg cells to 22.1% after *miR-9*-LOF, whereas medium and narrow H3K4me3 domains increased (Figure 2c). Interestingly, the shift from H3K4me3 broad domains to medium and narrow domains following *miR-9*-LOF was significant at promoters but not at gene body and intergenic regions (Figure 2d). However, enrichment plots showed that H3K4me3 levels were reduced following *miR-9*-LOF in H3K4me3 broad and medium domains at promoter, gene body and intergenic regions (Figure 2e). The reduction of H3K4me3 levels after *miR-9*-LOF was confirmed by confocal microscopy after H3K4me3-specific immunostaining in Ctrl- and *miR-9*-antagomiR transfected MLg cells (Supplementary Figure 3e).”

The new results included in the revised version of our manuscript demonstrate that even in cells with relatively low levels of *miR-9* in the cell nucleus (as MLE-12 cells), *miR-9* is required for the breadth of H3K4me3 domains, however in different loci as compared to MLg cells. Please, see the newly added Supplementary Figure 3c. Accordingly, we observed by ChIRP-seq that *miR-9* is enriched at different loci in MLg and MLE-12 cells (see Supplementary Figure 2c).

7. Figure 2c: This is confusing. While the text (lines 188-191) suggests a comparison between miR-9 targets and non-targets defined by genome interaction (ChIRP, Figure 1), the figure legend indicates "n=2439" and "n=881," which represent the number of genes whose expression was or wasn't influenced by miR-9 antagomir treatment. The discrepancy should be addressed and clarified.

We appreciate this comment from Reviewer 3. We would like to refer Reviewer 3 to our answer to her/his comment 5 above.

Once again, we would like to thank Reviewer 3 for the constructive comments that led us to an improved version of our manuscript. We respectfully hope that Reviewer 2 shares our positive opinion about the quality and novelty of our results and recommend our manuscript for publication at Nature Communications.

Point-by-point response to the Reviewers – Manuscript with the number NCOMMS-23-61140-A (now NCOMMS-23-61140-B) and the new title “Nuclear microRNA 9 is required for G-quadruplexes and 3D genome organization during TGFB-induced transcription” submitted as original scientific article to Nature Communications by Cordero J et al.

REVIEWER COMMENTS

Reviewer #1 (Remarks to the Author):

The study by Cordero et al. have significantly improved in the revision stage. In particular, all my critical issues have been satisfactorily addressed in the revised version. My only minor suggestion is to remove the mention to G-quadruplexes in the title, as this is not the main focus of the paper, while specifically mentioning miR-9.

We thank Reviewer 1 for the positive comments about our work and for the time and efforts implemented during the peer review of our manuscript providing valuable comments on how to improve it.

We understand the minor suggestion from Reviewer 1 as placing microRNA 9 in the spotlight, whereas G-quadruplexes and 3D genome organization are of secondary relevance. We have changed the title of the manuscript mentioning microRNA 9. However, we leaved G-quadruplexes in the title since it is a fundamental part of the mechanisms of transcriptional regulation that we are proposing as it is the 3D genome organization, which is also mentioned in the title. Following this line of ideas, we changed the title to:

“Nuclear microRNA 9 is required for G-quadruplexes and 3D genome organization during TGFB-induced transcription”

In addition, including the word G-quadruplexes in the title will allow the colleagues working on secondary nucleic acid structures to detect easier our manuscript.

Last but not least, the G4-specific CUT&Tag sequencing experiment performed during the revision of the manuscript is fundamental part of the work performed during the revision and the main reason for the delay of the re-submission of the manuscript, supporting the relevance of these results for our work and for the model of transcriptional regulation that we are proposing.

We respectfully hope that you share our opinion and agree with the title that we are suggesting.

Reviewer #2 (Remarks to the Author):

The authors have adequately addressed this reviewer's comments. Publication of the revised manuscript is recommended.

We appreciate the time and effort invested by Reviewer 2 in the peer-review of our manuscript, as well as the constructive suggestions from Reviewer 2. We also appreciate the recommendation for publication of our revised manuscript.

Reviewer #3 (Remarks to the Author):

I appreciate the substantial effort the authors have put into addressing the concerns raised in the initial review. The new data and controls have significantly improved the manuscript, making many of the previous conclusions more convincing.

However, I remain unconvinced that the proposed model is entirely correct. There are still aspects that do not fully make sense to me, and crucial mechanistic insights are still missing. These gaps are especially challenging given how controversial the model is. It is perhaps understandable that establishing such a model in a single study might be difficult, if not impossible.

Despite these lingering concerns, I believe that this study should be published with the understanding that time will ultimately determine whether this represents a breakthrough in understanding miRNA function in the nucleus or if it is an artifact. The field should have the opportunity to engage with these interesting findings and test their validity.

We thank Reviewer 3 for the time and efforts invested during the peer-review of our manuscript, as well as for the constructive comments that led us to an improved version of our manuscript during the revision.

We also appreciate the fair and professional recommendation from Reviewer 3 to publish the revised version of our manuscript even though she/he expresses a trace of skepticism about the model of transcriptional regulation that we propose based on the data presented. In any case, we completely agree with statement from Reviewer 3: "...The field should have the opportunity to engage with these interesting findings..."